# Distributional Robustness Bounds Generalization Errors

## Abstract

Bayesian methods, distributionally robust optimization methods, and regularization methods are three pillars of machine learning under distributional uncertainty, e.g., the uncertainty of an empirical distribution compared to the true underlying distribution. This paper investigates the connections among the three frameworks and, in particular, explores why these frameworks tend to have smaller generalization errors. Specifically, first, we suggest a quantitative definition for "distributional robustness", propose the concept of "robustness measure", and formalize several philosophical concepts in distributionally robust optimization. Second, we show that Bayesian methods are distributionally robust in the probably approximately correct (PAC) sense; in addition, by constructing a Dirichlet-process-like prior in Bayesian nonparametrics, it can be proven that any regularized empirical risk minimization method is equivalent to a Bayesian method. Third, we show that generalization errors of machine learning models can be characterized using the distributional uncertainty of the nominal distribution and the robustness measures of these machine learning models; this explains the reason why distributionally robust optimization models, Bayesian models, and regularization models tend to have smaller generalization errors in a unified manner.

## 1 Introduction

### 1.1 Background

A great number of statistical machine learning problems can be modeled by the optimization problem

$$\min_{\boldsymbol{x} \in \mathcal{X}} \; \mathbb{E}_{\xi \sim \mathbb{P}_0} h(\boldsymbol{x}, \xi), \tag{1}$$

where $\boldsymbol{x}$ is the decision vector, $\mathcal{X} \subseteq \mathbb{R}^l$ is the feasible region, and $\xi$ is the parameter supported on $\Xi \subseteq \mathbb{R}^m$; the parameter $\xi$ is a random vector and $\mathbb{P}_0$ is its true underlying distribution; $h : \mathcal{X} \times \Xi \to \mathbb{R}$ (typically $\mathbb{R}_+$) is the cost function. For example, see Vapnik (2000, Sec. 1.2), Bishop & Nasrabadi (2006, Sec. 1.5), James et al. (2021, Sec. 2.1), Kuhn et al. (2019). In data-driven supervised machine learning, $\xi$ can represent a data pair $(\boldsymbol{I}, \boldsymbol{O})$, i.e., $\xi := (\boldsymbol{I}, \boldsymbol{O})$, where $\boldsymbol{I}$ is the feature vector, $\boldsymbol{O}$ is the response, and $\boldsymbol{x}$ is the parameter of the hypothesis which is a function from $\boldsymbol{I}$ to $\boldsymbol{O}$. The optimization problem (1) is also popular in several other areas than statistical machine learning, e.g., applied statistics (Wang et al., 2022b), operations research (Shapiro et al., 2009), system simulations (Kouri et al., 2022), and statistical signal processing (Wang, 2022b), where the specific meanings that it conveys vary from one to another; for extensive reading on this aspect, see Appendix A.

### 1.2 Problem Statement

In the practice of machine learning, the underlying true distribution $\mathbb{P}_0$ might be unknown, and the nominal distribution $\bar{\mathbb{P}}$ (e.g., an estimate of $\mathbb{P}_0$) is used to construct the **nominal model**

$$\min_{\boldsymbol{x} \in \mathcal{X}} \mathbb{E}_{\xi \sim \bar{\mathbb{P}}} h(\boldsymbol{x}, \xi), \tag{2}$$

which serves as a surrogate for the **true model** (1). In the data-driven case, $\bar{\mathbb{P}}$ can be the empirical distribution $\hat{\mathbb{P}}_n$ constructed using $n$ training samples from $\mathbb{P}_0$, and (2) becomes

$$\min_{\boldsymbol{x} \in \mathcal{X}} \mathbb{E}_{\xi \sim \hat{\mathbb{P}}_n} h(\boldsymbol{x}, \xi), \tag{3}$$

which is a sample-average approximation (SAA) model known in the operations research community (Anderson & Nguyen, 2020) or an empirical risk minimization (ERM) model known in the machine learning community (Murphy, 2012; Mohri et al., 2018).[1] However, $\bar{\mathbb{P}}$ might be uncertain, that is, there might exist a discrepancy between $\bar{\mathbb{P}}$ and $\mathbb{P}_0$. For instance, in the data-driven setting, the uncertainty of $\hat{\mathbb{P}}_n$ comes from the scarcity of data. Usually, neglecting such **distributional uncertainty** in $\bar{\mathbb{P}}$ and directly using the nominal model (2) may cause significant performance degradation because the **empirical error** (i.e., empirical cost) $\mathbb{E}_{\bar{\mathbb{P}}} h(\boldsymbol{x}, \xi)$ is not guaranteed to provide an upper bound for the **generalization error** (i.e., generalization cost) $\mathbb{E}_{\mathbb{P}_0} h(\boldsymbol{x}, \xi)$, for every $\boldsymbol{x}$. As a result, by approaching the minimum empirical error $\mathbb{E}_{\bar{\mathbb{P}}} h(\bar{\boldsymbol{x}}, \xi)$, the generalization error $\mathbb{E}_{\mathbb{P}_0} h(\bar{\boldsymbol{x}}, \xi)$ at $\bar{\boldsymbol{x}}$ is not necessarily tightly controlled, where $\bar{\boldsymbol{x}} \in \mathrm{argmin}_{\boldsymbol{x}} \mathbb{E}_{\bar{\mathbb{P}}} h(\boldsymbol{x}, \xi)$. Therefore, reducing the negative influence caused by the distributional uncertainty in $\bar{\mathbb{P}}$ is the core of trustworthy machine learning. Since $\mathbb{E}_{\mathbb{P}_0} h(\boldsymbol{x}, \xi)$ cannot be directly evaluated, a natural strategy is to find a tight point-wise **generalization error (upper) bound** $\mathrm{UppBnd}(\boldsymbol{x})$ such that

$$\mathbb{E}_{\mathbb{P}_0} h(\boldsymbol{x}, \xi) \le \mathrm{UppBnd}(\boldsymbol{x}), \quad \forall \boldsymbol{x} \in \mathcal{X}. \tag{4}$$

Then by minimizing $\mathrm{UppBnd}(\boldsymbol{x})$ with $\boldsymbol{x}^*$, the generalization error $\mathbb{E}_{\mathbb{P}_0} h(\boldsymbol{x}^*, \xi)$ of $\boldsymbol{x}^*$ is also controlled. A popular realization of $\mathrm{UppBnd}(\boldsymbol{x})$ is through employing the empirical error $\mathbb{E}_{\bar{\mathbb{P}}}(\boldsymbol{x}, \xi)$. To be specific, the **generalization error gap** from the empirical error, which is a measure of overfitting, i.e.,

$$\mathbb{E}_{\mathbb{P}_0} h(\boldsymbol{x}, \xi) - \mathbb{E}_{\bar{\mathbb{P}}} h(\boldsymbol{x}, \xi), \tag{5}$$

should be *minimized or tightly upper bounded*, in expectation or in probability because (5) is a random quantity; note that $\bar{\mathbb{P}}$ might be a random measure such as the empirical measure $\hat{\mathbb{P}}_n$. As an example, in the data-driven setting, the **expected generalization error gap** from the empirical error, i.e.,

$$\mathbb{E}_{\mathbb{P}_0^n} \left[ \mathbb{E}_{\mathbb{P}_0} h(\boldsymbol{x}, \xi) - \mathbb{E}_{\hat{\mathbb{P}}_n} h(\boldsymbol{x}, \xi) \right], \tag{6}$$

is studied, where $\mathbb{P}_0^n$ is the $n$-fold product measure induced by $\mathbb{P}_0$.

Trustworthy machine learning under distributional uncertainty hence aims to design a new model based on, instead of directly using, the nominal/surrogate model (2) such that the generalization error bound in (4) or the generalization error gap in (5) is minimized or tightly controlled in expectation or in probability.

## 1.3 Literature Review

Facing the distributional mismatch between $\bar{\mathbb{P}}$ and $\mathbb{P}_0$, three treatment frameworks exist across the fields of statistics, operations research, and machine learning.

Bayesian methods are the earliest and also the first-hand choice (Ferguson, 1973; Ghosal & Van der Vaart, 2017; Wu et al., 2018). Suppose that we can construct a family $\mathcal{C} \subseteq \mathcal{M}(\Xi)$ of nominal distributions based on the knowledge of $\bar{\mathbb{P}}$ where $\mathcal{M}(\Xi)$ is the set of all distributions on the measurable space $(\Xi, \mathcal{B}_\Xi)$ and $\mathcal{B}_\Xi$ is the Borel $\sigma$-algebra on $\Xi$. For example, $\mathcal{C}$ can be a closed distributional $\epsilon$-ball centered at $\bar{\mathbb{P}}$, i.e., $\mathcal{C} := B_\epsilon(\bar{\mathbb{P}})$. Bayesian methods try to assign a distribution $\mathbb{Q}$ on the measurable space $(\mathcal{C}, \mathcal{B}_\mathcal{C})$ and solve the Bayesian counterpart of the nominal model (2):

$$\min_{\boldsymbol{x} \in \mathcal{X}} \mathbb{E}_{\mathbb{P} \sim \mathbb{Q}} \mathbb{E}_{\xi \sim \mathbb{P}} h(\boldsymbol{x}, \xi), \tag{7}$$

where $\mathbb{P} \in \mathcal{C}$ and $\mathbb{Q} \in \mathcal{M}(\mathcal{C})$. Note that $\mathbb{P}$ and $\mathbb{Q}$ may be non-parametric. In this case, we need to guarantee that $\mathbb{P}_0 \in \mathcal{C}$ and $\mathbb{Q}$ should be such that $\mathbb{P}_0$ is the most probable element to be drawn from $\mathcal{C}$. In fact, under some mild conditions, there exists an element $\mathbb{P}' \in \mathcal{C}$ such that $\mathbb{E}_\mathbb{Q} \mathbb{E}_\mathbb{P} h(\boldsymbol{x}, \xi) = \mathbb{E}_{\mathbb{P}'} h(\boldsymbol{x}, \xi)$ for every $\boldsymbol{x}$; see Theorem 1. Therefore, in nature, Bayesian methods inform us of a possible way to choose the "best"

---

[1]In this paper, we interchangeably use the two terms SAA and ERM.

candidate $\mathbb{P}'$ from the nominal distribution class $\mathcal{C}$. The ideal case is that $\mathbb{P}'$ can coincide with $\mathbb{P}_0$. If $\mathbb{P}'$ is a better surrogate for (i.e., is closer to) $\mathbb{P}_0$ than $\bar{\mathbb{P}}$, Bayesian methods have the potential to reduce generalization errors.

Another promising and popular treatment gives rise of the min-max distributionally robust optimization (DRO) framework (Rahimian & Mehrotra, 2022)

$$\min_{\boldsymbol{x} \in \mathcal{X}} \max_{\mathbb{P} \in \mathcal{C}} \mathbb{E}_{\xi \sim \mathbb{P}} h(\boldsymbol{x}, \xi), \tag{8}$$

if we are interested in controlling the worst-case performance of the nominal model (2). When the underlying true distribution $\mathbb{P}_0$ is contained in $\mathcal{C}$, by minimizing the worst-case cost, the cost at $\mathbb{P}_0$ can be also expected to be put down; cf. (4) where $\mathrm{UppBnd}(\boldsymbol{x}) := \max_{\mathbb{P} \in \mathcal{C}} \mathbb{E}_{\mathbb{P}} h(\boldsymbol{x}, \xi)$; for more justifications on DRO methods, see Kuhn et al. (2019). Under some mild conditions, there exists a distribution $\mathbb{P}'$ such that $\min_{\boldsymbol{x} \in \mathcal{X}} \max_{\mathbb{P} \in \mathcal{C}} \mathbb{E}_{\mathbb{P}} h(\boldsymbol{x}, \xi) = \min_{\boldsymbol{x} \in \mathcal{X}} \mathbb{E}_{\mathbb{P}'} h(\boldsymbol{x}, \xi)$ (Mohajerin Esfahani & Kuhn, 2018; Yue et al., 2021; Zhang et al., 2022; Gao, 2022; Gao & Kleywegt, 2022). Hence, in nature, DRO methods advise us of another way to find the "best" candidate $\mathbb{P}'$ from the nominal distribution class $\mathcal{C}$. However, the size of $\mathcal{C}$, e.g., the radius $\epsilon$ of the associated distributional ball $\mathcal{C} := B_\epsilon(\bar{\mathbb{P}})$, needs to be elegantly specified: neither too small nor too large. Suppose that $\boldsymbol{x}^*$ solves (8). If the size is too small, $\mathbb{P}_0$ may not be included in $\mathcal{C}$, and as a result, the true cost (i.e., generalization error) $\mathbb{E}_{\mathbb{P}_0} h(\boldsymbol{x}^*, \xi)$ cannot be upper bounded by the worst-case cost (8). If, on the other hand, the size of $\mathcal{C}$ is extremely large, the DRO method may be overly conservative: The upper bound (8) may be too loose, far upward away from the true cost $\mathbb{E}_{\mathbb{P}_0} h(\boldsymbol{x}^*, \xi)$. Interested readers may refer to, e.g., Mohajerin Esfahani & Kuhn (2018); Shapiro (2017); Sun & Xu (2016); Gao & Kleywegt (2022); Gao (2022) for more technical discussions on the DRO method. Comprehensive surveys, Rahimian & Mehrotra (2022); Kuhn et al. (2024), are also recommended for the general introduction of the DRO method.

Yet another strategy to hedge against the distributional uncertainty in $\bar{\mathbb{P}}$ is to use a regularizer $f(\boldsymbol{x})$, that is, to solve the regularized problem (Vapnik, 1998, Sec. A1.3) of the nominal model (2):

$$\min_{\boldsymbol{x} \in \mathcal{X}} \mathbb{E}_{\xi \sim \bar{\mathbb{P}}} h(\boldsymbol{x}, \xi) + \lambda f(\boldsymbol{x}), \tag{9}$$

where $\lambda \geq 0$ is a weight coefficient. This is a trending method in data-driven machine learning and applied statistics to reduce "overfitting". A well-known instance is the regularized empirical risk minimization model $\min_{\boldsymbol{x} \in \mathcal{X}} \mathbb{E}_{\hat{\mathbb{P}}_n} h(\boldsymbol{x}, \xi) + \lambda_n f(\boldsymbol{x})$ where $\bar{\mathbb{P}} := \hat{\mathbb{P}}_n$ is the empirical distribution given $n$ i.i.d. samples, $f(\boldsymbol{x})$ is the regularization term (e.g., any norm $\|\boldsymbol{x}\|$ of $\boldsymbol{x}$; see, e.g., Goodfellow et al. (2016, Chap. 7)), and the weight coefficient $\lambda_n$ may depend on $n$. By introducing a bias term $f(\boldsymbol{x})$, the generalization error of the regularized model (9) is possible to be reduced; recall the "bias-variance trade-off" in machine learning (Hastie et al., 2009, Sec. 2.9), (Domingos, 2012). A quantitative justification is that $\mathbb{E}_{\hat{\mathbb{P}}_n} h(\boldsymbol{x}, \xi) + \lambda_n f(\boldsymbol{x})$ can provide a (probabilistic) upper bound on $\mathbb{E}_{\mathbb{P}_0} h(\boldsymbol{x}, \xi)$, while $\mathbb{E}_{\hat{\mathbb{P}}_n} h(\boldsymbol{x}, \xi)$ solely cannot; cf. (4) where $\mathrm{UppBnd}(\boldsymbol{x}) := \mathbb{E}_{\hat{\mathbb{P}}_n} h(\boldsymbol{x}, \xi) + \lambda_n f(\boldsymbol{x})$, $\boldsymbol{x} \in \mathcal{X}$; recall, e.g., concentration properties of empirical measures (Zhang & Chen, 2021). As a result, supposing that $\boldsymbol{x}^*$ solves (9), the true cost (i.e., generalization error) $\mathbb{E}_{\mathbb{P}_0} h(\boldsymbol{x}^*, \xi)$ can be upper bounded.

For a specific but supplementary literature review on the data-driven case where $\bar{\mathbb{P}} = \hat{\mathbb{P}}_n$, interested readers can see Appendix B.

### 1.4 Research Gaps and Motivations

Although Bayesian probabilistic interpretation for some special regularized methods (e.g., LASSO and Ridge regression) exists, and recent results show that there is a close connection between distributional robustness and regularization under some conditions, NONE of the existing literature (profoundly) discusses the three methods together in a unified framework and collectively explains why they can generalize well in a unified sense. Hence, the primary motivation of this work is to investigate the connections among the Bayesian method (7), the DRO method (8), and the regularization method (9) in a unified framework, which can bring new insights to different research communities, such as machine learning and operations research. To this end, we begin by introducing a quantitative definition of "distributional robustness", formalizing the DRO theory from this perspective, and examining the rationale and limitation of the existing min-max DRO model (8).

## 1.5 Contributions

Following the research gaps and motivations aforementioned, the contributions of this paper can be visualized in Figure 1.

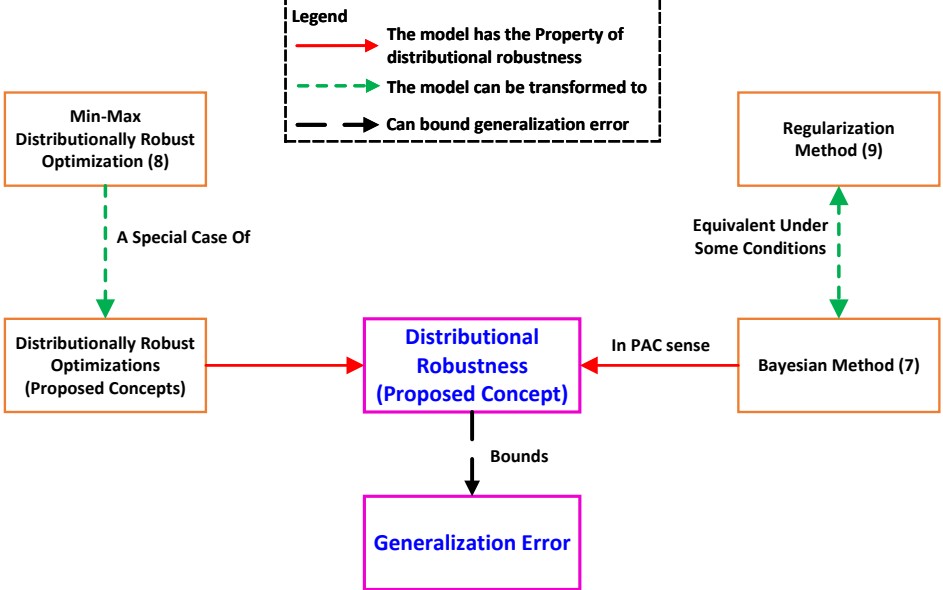

Figure 1: Bayesian method (7), DRO method (8), and regularization method (9) are distributionally robust. Since distributional robustness bounds generalization errors, the three methods can generalize well. (Detailed discussions and illustrating examples are available in Appendix D.7 and Appendix Q, respectively.)

To be specific, the contributions are summarized as follows.

1. Several existing concepts in distributionally robust optimization have been formalized; see Subsection 2.1. For example, a formal and quantitative definition of the concept of "distributional robustness" is suggested. Using the proposed concept system, we explain the rationale behind the popular min-max robust optimization model (8).

2. Bayesian methods (7) are proved to be probably approximately distributionally robust; see Subsection 2.2 and Figure 1.

3. We show that any regularization method (9) can be transformed to a Bayesian method (7), and therefore, interpreted from the Bayesian probabilistic perspective (7). The converse is also true under some conditions: Any Bayesian method (7) is adding a regularization term to the nominal model (2), and therefore, equivalent to a regularized method (9).[2] For details, see Subsection 2.3 and Figure 1.

4. We show that generalization errors can be characterized from the perspective of the distributional uncertainty of the nominal distribution $\bar{\mathbb{P}}$. In addition, it is proved that generalization errors can also be bounded using the distributional robustness of machine learning models. Therefore, by conducting

---

[2]In the existing literature, the regularized problem (9) is interpreted from another Bayesian perspective $\min_{\mathbb{Q}_{\boldsymbol{x}}} \mathbb{E}_{\boldsymbol{x} \sim \mathbb{Q}_{\boldsymbol{x}}} \mathbb{E}_{\xi \sim \bar{\mathbb{P}}} h(\boldsymbol{x}, \xi) + \lambda \operatorname{KL}(\mathbb{Q}_{\boldsymbol{x}} \| \Pi_{\boldsymbol{x}})$, where a prior distribution $\Pi_{\boldsymbol{x}}$ on $\mathcal{X}$, which encodes our prior belief on the hypothesis class $\mathcal{X}$, is used; $\mathbb{Q}_{\boldsymbol{x}}$ is a distribution on $\mathcal{X}$; $\operatorname{KL}(\cdot \| \cdot)$ denotes the Kullback–Leibler divergence. Let $\mathbb{Q}_{\boldsymbol{x}}$ be a point-mass distribution. If $\Pi_{\boldsymbol{x}}$ is a Gaussian (resp. Laplacian) distribution, we have the 2-norm (resp. 1-norm) regularized model. In this paper, we alternatively wonder whether the regularized model (9) can be transformed to, and hence interpreted from, the Bayesian method (7). The difference lies in over which space the prior distribution is applied: the hypothesis space $\mathcal{X}$ or the data distribution space $\mathcal{C}$. For details, see Subsection 2.3.1.

distributionally robust optimization, the upper bounds of generalization errors are reduced. It is in this sense that *distributional robustness bounds generalization errors*, which is a new perspective for the area. For details, see Subsection 2.4.

5. Comparisons between the proposed framework and the existing frameworks are conducted. For details, see Subsection 2.5.

The primary technical contributions of this paper lie in Subsections 2.2 and 2.3: to be specific, 1) Theorem 1 and Proposition 2 are first proved for Bayesian nonparametrics; 2) the connection between the Bayesian nonparametrics model (7) and the regularization model (9) is first established (see, e.g., Proposition 3, Subsection 2.3.1). In addition to these technical results, the remaining contributions are more motivational and conceptual in nature, positioning this paper as a framework-style survey that bridges multiple perspectives.

## 1.6 Notations, Preliminaries, and Organization

The notations are listed in Table 1. Technical preliminaries (e.g., the theory of min-max DRO and the concepts of generalization errors) are put in Appendix C. Section 2 presents the main results; concrete illustrating examples for these theoretical results are available in Appendix Q. Discussions and conclusions in Section 3 complete this paper.

Table 1: Notations

| Notation | Description |
|---|---|
| $\mathcal{M}(\Xi)$ | all probability measures on the measurable space $(\Xi, \mathcal{B}_\Xi)$ where $\mathcal{B}_\Xi$ is the Borel $\sigma$-algebra on $\Xi$ |
| $\mathcal{B}_{\mathcal{M}(\Xi)}$ | the Borel $\sigma$-algebra on $\mathcal{M}(\Xi)$ |
| $\mathbb{P}_0$ | underlying true distribution |
| $\hat{\mathbb{P}}_n$ | empirical distribution constructed by $n$ i.i.d. data from $\mathbb{P}_0$ |
| $\hat{\mathbb{P}}$ | prior distribution which is estimated using prior knowledge of $\mathbb{P}_0$ |
| $\bar{\mathbb{P}}$ | surrogate distribution (i.e., reference distribution) acting as a generic estimate of $\mathbb{P}_0$, which can be $\hat{\mathbb{P}}_n$ and $\hat{\mathbb{P}}$, among others |
| $\mathbb{P}^n$ | $n$-fold product measure (i.e., $n$-fold joint distribution) induced by the probability measure $\mathbb{P}$ |
| $[n]$ | $:= \{1, 2, \ldots, n\}$, the running index set |
| $\Delta(\mathbb{P}, \bar{\mathbb{P}})$ | statistical similarity measure between two distributions $\mathbb{P}$ and $\bar{\mathbb{P}}$; $\Delta$ can be any possible divergence or metric, among others, unless specified in a context |
| $B_\epsilon(\bar{\mathbb{P}})$ | $:= \{\mathbb{P} \in \mathcal{M}(\Xi) \mid \Delta(\mathbb{P}, \bar{\mathbb{P}}) \leq \epsilon\}$, closed distributional $\epsilon$-ball centered at $\bar{\mathbb{P}}$ |
| $\mathbb{E}_\mathbb{P} h(\boldsymbol{x}, \xi)$ | shorthand for $\mathbb{E}_{\xi \sim \mathbb{P}} h(\boldsymbol{x}, \xi)$; the expectation operator under distribution $\mathbb{P}$ |

# 2 Main Results

## 2.1 Concept System of Distributional Robustness

This subsection tries to formalize the concept system of "distributional robustness" and "distributionally robust optimization", and explain the rationale behind the min-max formulation (8). As a result, distributionally robust optimization models, Bayesian models, and regularization models can be discussed in a common concept system.

For conceptual simplicity and without loss of generality, this subsection assumes that the true problem (1) has only one global optimizer $\boldsymbol{x}_0 := \operatorname{argmin}_{\boldsymbol{x}} \mathbb{E}_{\mathbb{P}_0} h(\boldsymbol{x}, \xi)$. We begin with the following intuitive philosophy in uncertainty quantification.

**Philosophy 1** *For a specified system or model, "robustness" means that "small perturbations in parameters or inputs lead to small changes in performances or outputs".* □

This section tries to mathematically formalize the intuition above. Considering an **optimal value functional** (OVF)

$$T : \mathbb{P} \mapsto \min_{\boldsymbol{x} \in \mathcal{X}} \mathbb{E}_{\xi \sim \mathbb{P}} h(\boldsymbol{x}, \xi), \tag{10}$$

induced by true optimization model (1), every specific optimization model $\min_{\boldsymbol{x}} \mathbb{E}_{\mathbb{P}} h(\boldsymbol{x}, \xi)$ is a particularization of the optimal value functional $T$ at $\mathbb{P}$. For example, (1) can be written as $T(\mathbb{P}_0)$. Also, we define a two-argument cost functional

$$M : (\boldsymbol{x}, \mathbb{P}) \mapsto \mathbb{E}_{\xi \sim \mathbb{P}} h(\boldsymbol{x}, \xi). \tag{11}$$

In this subsection, we investigate a set $B_\epsilon(\mathbb{P}_0)$ of nominal distributions centered at the true distribution $\mathbb{P}_0$, because we are interested in studying the consequence of the deviations of the nominal distributions $\mathbb{P} \in B_\epsilon(\mathbb{P}_0)$ from the true distribution $\mathbb{P}_0$. As a result, both $T$ and $M$ can induce a **model set** through $B_\epsilon(\mathbb{P}_0)$, that is, $\{\mathbb{P} \mapsto T(\mathbb{P}) | \mathbb{P} \in B_\epsilon(\mathbb{P}_0)\}$ and $\{\mathbb{P} \mapsto M(\boldsymbol{x}, \mathbb{P}) | \mathbb{P} \in B_\epsilon(\mathbb{P}_0)\}$ (given a solution $\boldsymbol{x}$), respectively.

**Definition 1 (Distributional Robustness of Solution)** *Suppose the solution $\boldsymbol{x}_0$ to $T(\mathbb{P}_0)$ is unique. A point $\boldsymbol{x} \in \mathcal{X}$ is an $(\epsilon, L)$-distributionally robust solution to the cost functional $M$, or equivalently the model set $\{\mathbb{P} \mapsto M(\boldsymbol{x}, \mathbb{P}) | \mathbb{P} \in B_\epsilon(\mathbb{P}_0)\}$, at $\mathbb{P}_0$ if for every $\mathbb{P} \in B_\epsilon(\mathbb{P}_0)$, we have*

$$\left| \mathbb{E}_{\xi \sim \mathbb{P}} h(\boldsymbol{x}, \xi) - \mathbb{E}_{\xi \sim \mathbb{P}_0} h(\boldsymbol{x}_0, \xi) \right| \leq L,$$

*where $\boldsymbol{x}_0 := \operatorname{argmin}_{\boldsymbol{x}} \mathbb{E}_{\mathbb{P}_0} h(\boldsymbol{x}, \xi)$ and the value of $L$, which is the smallest value satisfying the above display, depends on the values of $\epsilon$ and $\boldsymbol{x}$. Here, given $\epsilon$ and $\boldsymbol{x}$, $L$ is the robustness measure of the solution $\boldsymbol{x}$ for the cost functional $M$ at $\mathbb{P}_0$: Given $\epsilon$ and $\boldsymbol{x}$, the smaller the value of $L$, the more robust the solution $\boldsymbol{x}$ for the cost functional $M$ at $\mathbb{P}_0$.* □

Note that a robust solution is with respect to a model set rather than a single model. This is because, when we discuss the distributional robustness of a robust solution $\boldsymbol{x}$ to a model set, all distributions in $B_\epsilon(\mathbb{P}_0)$ share the same solution $\boldsymbol{x}$. Definition 1 intuitively describes the robustness of the solution $\boldsymbol{x}$ when the nominal distribution deviates from the true distribution $\mathbb{P}_0$: i.e., *the objective value remains unchanged as much as possible at the given solution when (reasonably small) model perturbations exist.* The following example offers further motivations for Definition 1.

**Example 1** *We expect a robust solution for a model set because, for example, when the testing performance of a deep learning model is acceptable in multiple testing data sets, we do not want to adjust the parameters of the network (i.e., trained solution) from one to another. For another example, in political policy-making, when the external and internal environments do not significantly change, we prefer a policy to be consistent: "Governing a big country is like cooking small fish"; we should not stir the fish too much in cooking so that they will not fall apart into small pieces.* □

Interested readers are invited to see some additional motivations and discussions on the concept of "distributional robustness" in Appendix D.1; one can ignore it without missing the main points of this paper.

An optimization model that finds a distributionally robust solution to an optimization model set is called the **distributionally robust counterpart** for this model set. The example below exemplifies the concept of distributionally robust counterpart.

**Example 2** *Consider an optimization $\min_{\boldsymbol{x}} \mathbb{E}_{\mathbb{P}_0} h(\boldsymbol{x}, \xi)$ and a set of nominal distributions $B_\epsilon(\mathbb{P}_0)$. Suppose $\boldsymbol{x}'$ solves $\min_{\boldsymbol{x}} \mathbb{E}_{\mathbb{P}'} h(\boldsymbol{x}, \xi)$, for a distribution $\mathbb{P}' \in B_\epsilon(\mathbb{P}_0)$, and $\boldsymbol{x}'$ is also a distributionally robust solution for the nominal model set induced by $\min_{\boldsymbol{x}} \mathbb{E}_{\mathbb{P}_0} h(\boldsymbol{x}, \xi)$ through $B_\epsilon(\mathbb{P}_0)$. Then $\min_{\boldsymbol{x}} \mathbb{E}_{\mathbb{P}'} h(\boldsymbol{x}, \xi)$ is a distributionally robust counterpart for this model set, or simply, for $\min_{\boldsymbol{x}} \mathbb{E}_{\mathbb{P}_0} h(\boldsymbol{x}, \xi)$.* □

### 2.1.1 Formalization of Distributionally Robust Optimization

For a real-world optimization problem, we aim to find a robust solution for the associated model set and the robustness measure of this robust solution. This motivates the definition of the distributionally robust counterpart for the model (1).

**Definition 2 (Distributionally Robust Optimization)** *The distributionally robust counterpart for the model* (1) *is defined as*

$$
\begin{aligned}
\min_{\boldsymbol{x}, L} \quad & L \\
s.t. \quad & |\mathbb{E}_{\mathbb{P}} h(\boldsymbol{x}, \xi) - \mathbb{E}_{\mathbb{P}_0} h(\boldsymbol{x}_0, \xi)| \le L, \quad \forall \mathbb{P} : \Delta(\mathbb{P}, \mathbb{P}_0) \le \epsilon, \\
& \boldsymbol{x} \in \mathcal{X},
\end{aligned}
\tag{12}
$$

*where* $\boldsymbol{x}_0 := \operatorname{argmin}_{\boldsymbol{x}} \mathbb{E}_{\mathbb{P}_0} h(\boldsymbol{x}, \xi)$. *If* (12) *has a finite solution* $(\boldsymbol{x}^*, L^*)$, $\boldsymbol{x}^*$ *is a distributionally robust solution with the **absolute robustness measure** $L^*$; cf. Definition 1.* □

The distributionally robust optimization model in Definition 2 is a two-sided version: It limits the cost deviation $\mathbb{E}_{\mathbb{P}} h(\boldsymbol{x}, \xi) - \mathbb{E}_{\mathbb{P}_0} h(\boldsymbol{x}_0, \xi)$ at the solution $\boldsymbol{x}$ from both above and below. In practice, people may be only interested in the one-sided versions, that is, the upper bound of the cost difference $\mathbb{E}_{\mathbb{P}} h(\boldsymbol{x}, \xi) - \mathbb{E}_{\mathbb{P}_0} h(\boldsymbol{x}_0, \xi)$ at the solution $\boldsymbol{x}$. This is reminiscent of the two-sided and one-sided generalization errors in machine learning.

**Definition 3 (One-Sided Distributionally Robust Optimization)** *The one-sided distributionally robust counterpart for the model* (1) *is defined as*

$$
\begin{aligned}
\min_{\boldsymbol{x}, L} \quad & L \\
s.t. \quad & \mathbb{E}_{\mathbb{P}} h(\boldsymbol{x}, \xi) - \mathbb{E}_{\mathbb{P}_0} h(\boldsymbol{x}_0, \xi) \le L, \quad \forall \mathbb{P} : \Delta(\mathbb{P}, \mathbb{P}_0) \le \epsilon, \\
& L \ge 0, \\
& \boldsymbol{x} \in \mathcal{X},
\end{aligned}
\tag{13}
$$

*where* $\boldsymbol{x}_0 := \operatorname{argmin}_{\boldsymbol{x}} \mathbb{E}_{\mathbb{P}_0} h(\boldsymbol{x}, \xi)$. *If* (13) *has a finite solution* $(\boldsymbol{x}^*, L^*)$, *then* $\boldsymbol{x}^*$ *is a distributionally robust solution with the one-sided absolute robustness measure* $L^*$. □

### 2.1.2 Practical Implementations of Distributionally Robust Optimization

In practice, the true distribution $\mathbb{P}_0$ might be unknown (e.g., in the data-driven setup), and therefore, the optimization problems (12) and (13) cannot be explicitly solved. However, whenever we have a good estimate, e.g., $\bar{\mathbb{P}}$, of $\mathbb{P}_0$, we can alternatively resort to the surrogate distributionally robust optimization counterpart of (1).

**Definition 4 (Surrogate Distributionally Robust Optimization)** *The distributionally robust counterpart for the model* (1) *at the surrogate* $\bar{\mathbb{P}}$ *is defined as*

$$
\begin{aligned}
\min_{\boldsymbol{x}, L} \quad & L \\
s.t. \quad & |\mathbb{E}_{\mathbb{P}} h(\boldsymbol{x}, \xi) - \mathbb{E}_{\bar{\mathbb{P}}} h(\bar{\boldsymbol{x}}, \xi)| \le L, \quad \forall \mathbb{P} : \Delta(\mathbb{P}, \bar{\mathbb{P}}) \le \epsilon, \\
& \boldsymbol{x} \in \mathcal{X},
\end{aligned}
\tag{14}
$$

*where* $\bar{\boldsymbol{x}} := \operatorname{argmin}_{\boldsymbol{x}} \mathbb{E}_{\bar{\mathbb{P}}} h(\boldsymbol{x}, \xi)$. *For a data-driven problem,* $\bar{\mathbb{P}}$ *can be chosen as* $\hat{\mathbb{P}}_n$. □

Problem (14) is therefore the ***distributionally robust counterpart*** to the nominal problem (2). The one-sided version is given below.

**Definition 5 (One-Sided Surrogate Distributionally Robust Optimization)** *The one-sided distributionally robust counterpart for the model* (1) *at the surrogate* $\bar{\mathbb{P}}$ *is defined as*

$$
\begin{aligned}
\min_{\boldsymbol{x}, L} \quad & L \\
s.t. \quad & \mathbb{E}_{\mathbb{P}} h(\boldsymbol{x}, \xi) - \mathbb{E}_{\bar{\mathbb{P}}} h(\bar{\boldsymbol{x}}, \xi) \le L, \quad \forall \mathbb{P} : \Delta(\mathbb{P}, \bar{\mathbb{P}}) \le \epsilon, \\
& L \ge 0, \\
& \boldsymbol{x} \in \mathcal{X},
\end{aligned}
\tag{15}
$$

where $\bar{\boldsymbol{x}} := \operatorname{argmin}_{\boldsymbol{x}} \mathbb{E}_{\bar{\mathbb{P}}} h(\boldsymbol{x}, \xi)$. For a data-driven problem, $\bar{\mathbb{P}}$ can be chosen as $\hat{\mathbb{P}}_n$. □

*Relative Robustness*: In practice, for example, in a data-driven setting, $\epsilon$ is not easy to suitably specify to guarantee that $\mathbb{P}_0$ is contained in the empirical $\epsilon$-ball $B_\epsilon(\hat{\mathbb{P}}_n) := \{\mathbb{P} \in \mathcal{M}(\Xi) | \Delta(\mathbb{P}, \hat{\mathbb{P}}_n) \le \epsilon\}$. When $\mathbb{P}_0$ is not contained in the empirical $\epsilon$-ball, the data-driven surrogate min-max distributionally robust optimizations (14) and (15) at $\hat{\mathbb{P}}_n$ is meaningless because the true cost at $\mathbb{P}_0$ cannot be explicitly upper bounded. To this end, Appendix D.2 discusses a more general case where the nominal distributions are not limited to the subspace $B_\epsilon(\hat{\mathbb{P}}_n)$. The key idea is to study the alternative constraint

$$\mathbb{E}_{\mathbb{P}} h(\boldsymbol{x}, \xi) - \mathbb{E}_{\bar{\mathbb{P}}} h(\bar{\boldsymbol{x}}, \xi) \le L \cdot \Delta(\mathbb{P}, \bar{\mathbb{P}}) \tag{16}$$

instead of

$$\mathbb{E}_{\mathbb{P}} h(\boldsymbol{x}, \xi) - \mathbb{E}_{\bar{\mathbb{P}}} h(\bar{\boldsymbol{x}}, \xi) \le L, \quad \forall \mathbb{P} : \Delta(\mathbb{P}, \bar{\mathbb{P}}) \le \epsilon. \tag{17}$$

The minimum value of $L$ satisfying (16) is termed the **relative robustness measure**; cf. the absolute robustness measure in (17). The concept of relative robustness measure is related to the *Robust Satisficing* model (Long et al., 2022). Interested readers can see Appendix D.2 for details; however, one may ignore it without missing the main points of this paper.

The fact below provides the rationale for the surrogate distributionally robust optimizations.

**Fact 1 (Surrogate Distributionally Robust Optimization)** *Let $(\boldsymbol{x}_1, L_1)$ solve the surrogate distributionally robust counterpart (14) at $\bar{\mathbb{P}}$ for model (1). Then $\boldsymbol{x}_1$ is distributionally robust (in the sense of Definitions 1 and 2) with robustness measure $L_1 + L_2$ where $L_2 := |\mathbb{E}_{\bar{\mathbb{P}}} h(\bar{\boldsymbol{x}}, \xi) - \mathbb{E}_{\mathbb{P}_0} h(\boldsymbol{x}_0, \xi)|$. Likewise, suppose $(\boldsymbol{x}_1, L_1)$ solves the one-sided surrogate distributionally robust counterpart (15) at $\bar{\mathbb{P}}$ for model (1). Then $\boldsymbol{x}_1$ is distributionally robust with one-sided robustness measure $L_1 + L_2$.*

**Proof.** See Appendix E for the proof based on telescoping and triangle inequalities. □

### 2.1.3 Min-Max Distributionally Robust Optimization

In what follows, we explain the rationale behind the min-max distributionally robust optimization (8). That is, we aim to find the relation between the min-max distributionally robust optimization (8) and the proposed concept system of "distributional robustness".

*Rationale Behind The Min-Max Distributional Robustness*: In operations research and machine learning practice, the most popular distributionally robust optimization model is the min-max model, which is formally defined in Definition 6.

**Definition 6 (Min-Max Distributionally Robust Optimization)** *The min-max distributionally robust counterpart for the model (1) is defined as*

$$\begin{aligned} \min_{\boldsymbol{x}} \max_{\mathbb{P}} \quad & \mathbb{E}_{\xi \sim \mathbb{P}} h(\boldsymbol{x}, \xi) \\ s.t. \quad & \Delta(\mathbb{P}, \mathbb{P}_0) \le \epsilon, \\ & \boldsymbol{x} \in \mathcal{X}. \end{aligned} \tag{18}$$

*If (18) has a finite solution $(\boldsymbol{x}^*, \mathbb{P}^*)$, then $\boldsymbol{x}^*$ is the min-max (i.e., worst-case) distributionally robust solution at the least-favorable (i.e., worst-case) distribution $\mathbb{P}^*$.* □

We can show that the min-max distributionally robust optimization model is a practical instance of the one-sided distributionally robust optimization model in Definition 3.

**Proposition 1 (Min-Max Distributionally Robust Optimization)** *Problem (13) is equivalent to Problem (18) in the sense that if $(\boldsymbol{x}^*, L^*)$ solves (13), then $(\boldsymbol{x}^*, \mathbb{P}^*)$ solves (18) where $\mathbb{P}^*$ satisfies*

$$\mathbb{E}_{\mathbb{P}^*} h(\boldsymbol{x}^*, \xi) - \mathbb{E}_{\mathbb{P}_0} h(\boldsymbol{x}_0, \xi) = L^*;$$

*the converse is also true: if $(\boldsymbol{x}^*, \mathbb{P}^*)$ solves (18), then $(\boldsymbol{x}^*, L^*)$ solves (13) where $L^*$ satisfies the above display.*

**Proof.** See Appendix H. The key is to show that Problem (13) can be reformulated to Problem (18), and vice versa. □

Proposition 1 explains why min-max distributionally robust optimization models are valid. However, min-max distributionally robust optimization models are only able to provide one-sided distributional robustness, that is, the upper bound of $\mathbb{E}_{\mathbb{P}}h(\boldsymbol{x},\xi)$ at $\boldsymbol{x}$. For additional discussions on this point, see Appendix D.5, if interested.

*Rationale Behind The Surrogate Min-Max Distributional Robustness*: In what follows, we explore the situation where only the nominal distribution $\bar{\mathbb{P}}$ is available, which is the practical case as in (8).

**Definition 7 (Surrogate Min-Max Distributionally Robust Optimization)** *The surrogate min-max distributionally robust counterpart for the model* (1) *at the surrogate $\bar{\mathbb{P}}$ is defined as*

$$
\begin{aligned}
\min_{\boldsymbol{x}} \max_{\mathbb{P}} \quad & \mathbb{E}_{\xi\sim\mathbb{P}}h(\boldsymbol{x},\xi) \\
s.t. \quad & \Delta(\mathbb{P},\bar{\mathbb{P}}) \leq \epsilon, \\
& \boldsymbol{x} \in \mathcal{X},
\end{aligned}
\tag{19}
$$

*which is an instance of* (8) *when the distributional class $\mathcal{C}$ in* (8) *is specified by a distributional ball. For a data-driven problem, $\bar{\mathbb{P}}$ can be chosen as $\hat{\mathbb{P}}_n$.* □

Problem (19) is therefore termed the ***min-max distributionally robust counterpart*** of the nominal Problem (2).

**Corollary 1 (Surrogate Min-Max Distributionally Robust Optimization)** *Problem* (15) *is equivalent to* (19)*. Therefore, the solutions returned by surrogate min-max distributionally robust optimization models are distributionally robust for the model set centered at the nominal model* (2)*.*

**Proof.** Compare with Fact 1 and Proposition 1. □

Corollary 1 explains why the popular surrogate min-max distributionally robust optimization models (8) are valid in real-world applications, in the sense of distributional robustness.

### 2.1.4 Robustness and Sensitivity

In the data-driven setup, there is a trade-off between the distributional robustness against distributional model perturbations and the sensitivity/specificity to the (training) data (i.e., the nominal model). To be specific, if a model set has a large robustness measure, the model set has a weak ability to hedge against model perturbations because the objective value is not still in this model set; cf. Definition 2. However, this model set tends to distinguish two similar but different data distributions; i.e., the model set has large resolution in identifying different data distributions. In contrast, if the model set has a small robustness measure, the objective value of this model set is insensitive (i.e., still) to model perturbations, however, this model set possibly cannot identify two similar but different data distributions either. In practice, we must balance between the robustness against perturbations and the sensitivity/specificity to the collected training data. This motivates us that generalization error bounds might be specified using the distributional model perturbations and the distributional robustness measures, which will be revisited in detail later in Subsection 2.4.

**Example 3** *The robustness-sensitivity trade-off can be intuitively understood from a life example. The position of a big stone is robust (i.e., still) against a light wind while the position of a piece of cloth is not. However, only this piece of cloth is an indicator of the existence of the light wind; we cannot identify this kind of slight change in air currents through the movement of the big stone.* □

The remark below summarizes a new insight for the uncertainty quantification community.

**Remark 1 (Robustness-Sensitivity Trade-Off)** *There is a trade-off between the robustness (to the distributional uncertainty of $\bar{\mathbb{P}}$ compared to $\mathbb{P}_0$) and the sensitivity/specificity (to the distributional information $\bar{\mathbb{P}}$) for a machine learning model. In the data-driven case, the distributional uncertainty is caused by unseen data, while the distributional information is conveyed in training data.* $\quad\square$

### 2.2 Bayesian Methods

In this subsection, we consider the Bayesian setting (7) to address the modeling uncertainty in $\bar{\mathbb{P}}$ for $\mathbb{P}_0$:

$$\min_{\boldsymbol{x}\in\mathcal{X}}\mathbb{E}_{\mathbb{P}\sim\mathbb{Q}}\mathbb{E}_{\xi\sim\mathbb{P}}h(\boldsymbol{x},\xi),$$

where $\mathbb{Q}$ is a probability measure on $(\mathcal{M}(\Xi),\mathcal{B}_{\mathcal{M}(\Xi)})$ and $\mathcal{B}_{\mathcal{M}(\Xi)}$ is the Borel $\sigma$-algebra on $\mathcal{M}(\Xi)$; $\mathbb{P}$ is a random measure which is distributed according to $\mathbb{Q}$. In other words, although we do not know the true distribution $\mathbb{P}_0$, we know that it is a realization drawn from the distribution $\mathbb{Q}$ rather than simply lies in a distributional ball (i.e., an uncertainty set). This philosophy can be straightforwardly generated in light of the difference between the Frequentists and the Bayesians: Frequentist statistics only assumes that an unknown parameter lies in a space (e.g., a subset of $\mathbb{R}^n$) but Bayesian statistics assumes that there exists a (prior) distribution for the unknown parameter. In Bayesian statistics, $\mathbb{P}$ is called a first-order probability measure and $\mathbb{Q}$ is called a second-order probability measure (Gaudard & Hadwin, 1989). In this case, $\mathbb{Q}$ can also be seen as a stochastic process whose realizations are probability measures and $\mathbb{P}$ is a realization of $\mathbb{Q}$ (Ferguson, 1973), (Ghosal & Van der Vaart, 2017, Chap. 3). To clarify further, suppose $(\Xi_1,\Xi_2,\cdots,\Xi_k)$ is an arbitrary $k$-partition of $\Xi$. The probability vector $(\mathbb{P}(\Xi_1),\mathbb{P}(\Xi_2),\cdots,\mathbb{P}(\Xi_k))$ is a random vector whose distribution is specified by $\mathbb{Q}$. Hence, $(\mathbb{Q}(\Xi_1),\mathbb{Q}(\Xi_2),\cdots,\mathbb{Q}(\Xi_k))$ is a stochastic process indexed by the set $\{1,2,\cdots,k\}$ and $(\mathbb{P}(\Xi_1),\mathbb{P}(\Xi_2),\cdots,\mathbb{P}(\Xi_k))$ is a realization.[3] Note that the true distribution $\mathbb{P}_0$ can also be seen as a realization of $\mathbb{Q}$ if $\mathbb{P}_0$ is in the support set of $\mathbb{Q}$. A good prior $\mathbb{Q}$ should concentrate around $\mathbb{P}_0$: it is most ideal that $\mathbb{Q}$ is a Dirac measure at $\mathbb{P}_0$ and it is ideal if $\mathbb{P}_0$ is the mean distribution under $\mathbb{Q}$.[4]

**Example 4** *A concrete example of the Bayesian setting* (7) *is the parametric Bayesian method:*

$$\min_{\boldsymbol{x}}\mathbb{E}_{\boldsymbol{\theta}\sim\mathbb{Q}_{\boldsymbol{\theta}}}\mathbb{E}_{\xi\sim\mathbb{P}_{\xi;\boldsymbol{\theta}}}h(\boldsymbol{x},\xi), \tag{20}$$

*where $\mathbb{P}_{\xi;\boldsymbol{\theta}}$ is the distribution of $\xi$, parameterized by $\boldsymbol{\theta}\in\mathbb{R}^q$, and $\mathbb{Q}_{\boldsymbol{\theta}}$ is the distribution of $\boldsymbol{\theta}$. In this parametric setting, the true parameter $\boldsymbol{\theta}_0$ of the true distribution $\mathbb{P}_{\xi;\boldsymbol{\theta}_0}$ is a realization from $\mathbb{Q}_{\boldsymbol{\theta}}$. For a specific parametric example, see Wu et al. (2018). In this paper, we focus on the general non-parametric case as in* (7). $\quad\square$

The theorem below gives a reformulation of the Bayesian model (7), which is new to the area of Bayesian nonparametrics.

**Theorem 1** *If $\mathbb{P}'$ is the mean of $\mathbb{P}$ under $\mathbb{Q}$ and $\mathbb{E}_{\mathbb{Q}}\mathbb{E}_{\mathbb{P}}|h(\boldsymbol{x},\xi)| < \infty$, for every $\boldsymbol{x}$, we have $\mathbb{E}_{\mathbb{Q}}\mathbb{E}_{\mathbb{P}}h(\boldsymbol{x},\xi) = \mathbb{E}_{\mathbb{P}'}h(\boldsymbol{x},\xi)$.*

**Proof.** See Appendix I for the proof based on measure-theoretic Funibi's theorem. $\quad\square$

Theorem 1 generalizes Ferguson (1973, Thm. 3), where only the Dirichlet process is investigated. Using a Dirichlet-process prior $\mathbb{Q}$ means that the probability vector $(\mathbb{P}(\Xi_1),\mathbb{P}(\Xi_2),\cdots,\mathbb{P}(\Xi_k))$ follows a Dirichlet distribution. However, Theorem 1 is more a theoretical justification than a useful instruction because, in practice, the most widely used priors on the space of non-parametric probability measures are Dirichlet-process priors and tail-free process priors, attributed to their conjugacy; see Ghosal & Van der Vaart (2017, Chap. 3) and Xie et al. (2021).

The proposition below shows that the solution returned by the Bayesian counterpart (7) for the surrogate model (2) is Probably Approximately Distributionally Robust (PADR).

---

[3] More generally, the stochastic process can be indexed by the Borel $\sigma$-algebra on $\Xi$, i.e., $\{\mathbb{Q}(E)\}$, $\forall E\in\mathcal{B}(\Xi)$.

[4] The mean distribution under $\mathbb{Q}$ is $\mathbb{P}'$ such that $\mathbb{P}'(E) = \int_{\mathbb{R}}\mathbb{P}(E)\mathbb{Q}(\mathrm{d}\mathbb{P}(E))$, $\forall E\in\mathcal{B}(\Xi)$. Note that, for a given Borel set $E\in\mathcal{B}(\Xi)$, $\mathbb{P}(E)$ is a random variable on $\mathbb{R}$.

**Proposition 2** *Suppose $h$ is a non-negative cost function.[5] The solution $\boldsymbol{x}^*$ of the Bayesian counterpart* (7) *for the surrogate model* (2) *is probably approximately distributionally robust with absolute robustness measure $L$ (cf. Definition 2) with probability at least $\max\left\{0, \quad 1 - \frac{\mathbb{E}_{\mathbb{Q}}\mathbb{E}_{\mathbb{P}}h(\boldsymbol{x}^*,\xi)+\mathbb{E}_{\mathbb{P}_0}h(\boldsymbol{x}_0,\xi)}{L}\right\}$. To be specific, we have*

$$\mathbb{Q}\Big[\big|\mathbb{E}_{\mathbb{P}}h(\boldsymbol{x}^*,\xi) - \mathbb{E}_{\mathbb{P}_0}h(\boldsymbol{x}_0,\xi)\big| \leq L\Big] = 1 - \frac{\mathbb{E}_{\mathbb{Q}}\mathbb{E}_{\mathbb{P}}h(\boldsymbol{x}^*,\xi) + \mathbb{E}_{\mathbb{P}_0}h(\boldsymbol{x}_0,\xi)}{L}.$$

**Proof.** See Appendix J for the proof based on Markov's inequality. □

Note that, in Proposition 2, $L$ needs to be specified to a sufficiently large value. Otherwise, the probability on the right side cannot be sufficiently large. However, for a fixed probability level, minimizing $\mathbb{E}_{\mathbb{Q}}\mathbb{E}_{\mathbb{P}}h(\boldsymbol{x},\xi)$ implies reducing the robustness measure $L$.

**Remark 2** *Proposition 2 can be alternatively stated as follows. Suppose $h$ is non-negative. We have*

$$\mathbb{Q}[\mathbb{E}_{\mathbb{P}}h(\boldsymbol{x},\xi) \leq L] \geq 1 - \frac{\mathbb{E}_{\mathbb{Q}}\mathbb{E}_{\mathbb{P}}h(\boldsymbol{x},\xi)}{L}, \quad \forall\boldsymbol{x}, \tag{21}$$

*which also justifies the Bayesian model* (7)*, in the sense of bounding generalization error. However,* (21) *does not specify how close $\mathbb{E}_{\mathbb{P}}h(\boldsymbol{x},\xi)$ is to $\mathbb{E}_{\mathbb{P}_0}h(\boldsymbol{x}_0,\xi)$.* □

*Data-Driven Case:* In the data-driven setting, the most popular non-parametric Bayesian prior for $\mathbb{Q}$ is the Dirichlet-process prior, whose posterior mean distribution, after observing $n$ i.i.d. samples, is given by (Ferguson, 1973), (Ghosal & Van der Vaart, 2017, Chap. 3)

$$\frac{\alpha}{\alpha+n}\hat{\mathbb{P}} + \frac{n}{\alpha+n}\hat{\mathbb{P}}_n,$$

where $\hat{\mathbb{P}}$ is a prior estimate of $\mathbb{P}_0$ based on our prior belief and $\alpha$ is a non-negative scalar used to adjust our trust level of $\hat{\mathbb{P}}$: the larger the value of $\alpha$, the more trust we have towards $\hat{\mathbb{P}}$. This can be seen as a mixture distribution of $\hat{\mathbb{P}}$ and $\hat{\mathbb{P}}_n$ with mixing weights $\frac{\alpha}{\alpha+n}$ and $\frac{n}{\alpha+n}$, respectively: i.e., a weighted combination of prior knowledge and data evidence. As $n \to \infty$, the weight of the prior belief $\hat{\mathbb{P}}$ decays quickly, which is consistent with our intuition that as the sample size gets larger, we should trust more on the empirical distribution $\hat{\mathbb{P}}_n$ due to the concentration property of $\hat{\mathbb{P}}_n$, i.e., the weak convergence of $\hat{\mathbb{P}}_n$ to $\mathbb{P}_0$ (e.g., recall the Portmanteau theorem). One may also reminisce about the concentration property of the empirical distribution $\hat{\mathbb{P}}_n$ defined by Wasserstein distance;[6] note that the Wasserstein distance metrizes this weak convergence (Weed & Bach, 2019).

As a result of Theorem 1, if we use the Dirichlet-process prior, the Bayesian model (7) is particularized into

$$\min_{\boldsymbol{x}} \frac{\alpha}{\alpha+n}\mathbb{E}_{\hat{\mathbb{P}}}h(\boldsymbol{x},\xi) + \frac{n}{\alpha+n}\mathbb{E}_{\hat{\mathbb{P}}_n}h(\boldsymbol{x},\xi). \tag{22}$$

We can further generalize (22) into

$$\min_{\boldsymbol{x}} \beta_n\mathbb{E}_{\hat{\mathbb{P}}}h(\boldsymbol{x},\xi) + (1-\beta_n)\mathbb{E}_{\hat{\mathbb{P}}_n}h(\boldsymbol{x},\xi), \tag{23}$$

where $\beta_n \in [0,1]$ is not necessarily equal to $\frac{\alpha}{\alpha+n}$ but we still require that $\beta_n \to 0$, as $n \to \infty$. The example below shows an application of the learning model (23) in data augmentation (Cui et al., 2015; Shorten & Khoshgftaar, 2019; Shorten et al., 2021) and adversarial learning.

**Example 5 (Data Augmentation)** *In data-driven machine learning, $\hat{\mathbb{P}}$ can be seen as a perturbation distribution used to augment the training data set $\{\xi_i\}_{i\in[n]}$; for a specific example, recall the panda-gibbon example in the adversarial learning framework (Goodfellow et al., 2015). To clarify further, instead of directly*

---

[5]This non-negativity condition is standard in machine learning; see, e.g., Wang et al. (2022a).

[6]Compare with (32) in Appendix C.1.

using the empirical distribution $\hat{\mathbb{P}}_n$ to solve a learning problem, we can use data augment techniques (e.g., Gaussian perturbations for images, geometric transformations for images, word shuffling/insertion/deletion for texts, noise injection for audios) to construct a surrogate distribution $\beta_n\hat{\mathbb{P}} + (1 - \beta_n)\hat{\mathbb{P}}_n$ to solve the problem. The practical benefit of data augmentation has been widely reported in, e.g., Cui et al. (2015); Shorten & Khoshgoftaar (2019); Shorten et al. (2021). $\square$

Based on (23), a new learning framework is introduced in Wang et al. (2023) as follows:

$$\min_{\boldsymbol{x}} \beta_n \max_{\mathbb{P}\in B_\epsilon(\hat{\mathbb{P}})} \mathbb{E}_{\mathbb{P}}h(\boldsymbol{x}, \xi) + (1 - \beta_n)\mathbb{E}_{\hat{\mathbb{P}}_n}h(\boldsymbol{x}, \xi),$$

which aims to achieve robustness against the uncertainty in the user-specified prior distribution $\hat{\mathbb{P}}$. The highlights of this new framework are three-fold: 1) By letting $\hat{\mathbb{P}} \coloneqq \hat{\mathbb{P}}_n$, it is a unification of DRO and ERM, which introduces a new algorithmic freedom (i.e., $\beta_n \in [0, 1]$) for better testing performance than DRO and ERM; 2) It suggests how to specify the regularizer $f(\boldsymbol{x})$ in a regularized ERM method (9); 3) It helps reduce the conservativeness of DRO and improves the robustness of ERM (recall the robustness-sensitivity trade-off in Remark 1).

## 2.3 Regularized Sample-Average Approximation

We re-arrange (23) to see

$$(1 - \beta_n)\min_{\boldsymbol{x}} \mathbb{E}_{\xi\sim\hat{\mathbb{P}}_n}h(\boldsymbol{x}, \xi) + \lambda_n \cdot f(\boldsymbol{x}), \tag{24}$$

which is equivalent to solve

$$\min_{\boldsymbol{x}} \mathbb{E}_{\xi\sim\hat{\mathbb{P}}_n}h(\boldsymbol{x}, \xi) + \lambda_n \cdot f(\boldsymbol{x}), \tag{25}$$

where

$$\lambda_n \coloneqq \frac{\beta_n}{1 - \beta_n} \qquad \text{and} \qquad f(\boldsymbol{x}) \coloneqq \mathbb{E}_{\xi\sim\hat{\mathbb{P}}}h(\boldsymbol{x}, \xi).$$

Obviously, (25) is a regularized SAA (i.e., regularized ERM) model, which is popular in applied statistics and, especially, machine learning, where $f(\boldsymbol{x})$ is a regularizer (e.g., any norm of $\boldsymbol{x}$). One may recall the Ridge regression where $f(\boldsymbol{x}) \coloneqq \|\boldsymbol{x}\|_2$ and LASSO (least absolute shrinkage and selection operator) regression where $f(\boldsymbol{x}) \coloneqq \|\boldsymbol{x}\|_1$. The example below reveals the connection between the data augmentation techniques and the regularization techniques.

**Example 6 (Data Augmentation; *Continued from Example 5*)** *Any data augmentation technique* (23) *is associated with a data perturbation distribution $\hat{\mathbb{P}}$, and therefore, any data augmentation technique is equivalent to a regularization method* (25) *where the regularizer $f(\boldsymbol{x}) \coloneqq \mathbb{E}_{\hat{\mathbb{P}}}h(\boldsymbol{x}, \xi)$. Note that $\hat{\mathbb{P}}$ can depend on $\hat{\mathbb{P}}_n$ (e.g., geometric transformations for images) and $\hat{\mathbb{P}}$ can also be independent of $\hat{\mathbb{P}}_n$ (e.g., Gaussian perturbations for images).* $\square$

Example 6 serves as a theoretical justification of the equivalence between the data augmentation techniques and the regularization effect; for existing empirical justifications, see, e.g., the success of convolutional neural networks using perturbed training data sets LeCun et al. (1998); Santos & Papa (2022).

The proposition below suggests the condition under which a regularization method (25) can be transformed to a Bayesian method (7).

**Proposition 3** *Let $\mathbb{Q}$ be a Dirichlet-process prior. For every specified regularizer $f(\boldsymbol{x})$, if there exists a probability measure $\hat{\mathbb{P}}$ such that*

$$f(\boldsymbol{x}) = \mathbb{E}_{\xi\sim\hat{\mathbb{P}}}h(\boldsymbol{x}, \xi), \quad \forall\boldsymbol{x}, \tag{26}$$

*then the regularized SAA model* (25) *is equivalent to the Bayesian model* (7). *Therefore, any Bayesian model* (7) *is adding a regularizer $f(\boldsymbol{x})$ (induced by $\hat{\mathbb{P}}$) to the empirical model $\min_{\boldsymbol{x}} \mathbb{E}_{\hat{\mathbb{P}}_n}h(\boldsymbol{x}, \xi)$. Also, any regularized SAA optimization method is a Bayesian method whose solution $\boldsymbol{x}^*$ is probably approximately distributionally robust with absolute robustness measure L with probability at least*

$$\max\left\{0, \quad 1 - \frac{\mathbb{E}_{\mathbb{Q}}|\mathbb{E}_{\mathbb{P}}h(\boldsymbol{x}^*, \xi) - \mathbb{E}_{\mathbb{P}_0}h(\boldsymbol{x}_0, \xi)|}{L}\right\},$$

where $\mathbb{Q}$ is a Dirichlet-process-like prior with posterior mean $\beta_n\hat{\mathbb{P}} + (1-\beta_n)\hat{\mathbb{P}}_n$. Further, if $h$ is non-negative, which is usually the case in supervised machine learning (Wang et al., 2022a), $\boldsymbol{x}^*$ is probably approximately distributionally robust with absolute robustness measure $L$ with probability at least

$$1 - \frac{\mathbb{E}_{\mathbb{Q}}\mathbb{E}_{\mathbb{P}}h(\boldsymbol{x}^*,\xi) + \mathbb{E}_{\mathbb{P}_0}h(\boldsymbol{x}_0,\xi)}{L}$$
$$= 1 - \frac{\beta_n\mathbb{E}_{\hat{\mathbb{P}}}h(\boldsymbol{x}^*,\xi) + (1-\beta_n)\mathbb{E}_{\hat{\mathbb{P}}_n}h(\boldsymbol{x}^*,\xi) + \mathbb{E}_{\mathbb{P}_0}h(\boldsymbol{x}_0,\xi)}{L},$$

*if it is non-negative.*

**Proof.** See Appendix K. Given a regularizer $f(\boldsymbol{x})$, we can construct a Bayesian non-parametric prior $\hat{\mathbb{P}}$, and vice versa. $\qquad\square$

**Remark 3** *In Proposition 3, we require $\mathbb{Q}$ to be a Dirichlet-process prior so that the equivalence between the regularization model (9) and the Bayesian model (7) holds. In general, the equivalence is no longer true but an inclusion relationship holds. To be specific, a regularization model (9) can be transformed into a Bayesian model (7) by constructing $\hat{\mathbb{P}}$ satisfying (26). But a Bayesian model (7) cannot be necessarily transformed into a regularization model (9) if $\mathbb{Q}$ is not a Dirichlet process prior.* $\qquad\square$

The condition in (26) is not restrictive, and $\hat{\mathbb{P}}$ can be constructed using $f(\boldsymbol{x})$ and $h(\boldsymbol{x},\xi)$. An example is as below.

**Example 7** *When $\Xi$ is bounded, $\hat{\mathbb{P}}$ can be chosen as the $\boldsymbol{x}$-parametric uniform distribution on $\Xi$ with density function*

$$\frac{\mathrm{d}\hat{\mathbb{P}}}{\mathrm{d}\mathcal{L}} := \frac{f(\boldsymbol{x})}{\int_{\Xi} h(\boldsymbol{x},\xi)\mathrm{d}\xi}, \quad \forall\boldsymbol{x},$$

*where $\mathcal{L}$ is the Lebesgue measure on $(\Xi, \mathcal{B}_{\Xi})$ and the left side of the above display is the Radon–Nikodym derivative of $\hat{\mathbb{P}}$ with respect to $\mathcal{L}$.* $\qquad\square$

For another example when $\Xi$ is unbounded, see Appendix D.6.

### 2.3.1 Probabilistic Interpretations of Regularized SAA Models

Different from the deterministic learning problem (1), the randomized learning counterpart

$$\min_{\mathbb{Q}_{\boldsymbol{x}}} \mathbb{E}_{\boldsymbol{x}\sim\mathbb{Q}_{\boldsymbol{x}}}\mathbb{E}_{\xi\sim\mathbb{P}_0}h(\boldsymbol{x},\xi), \tag{27}$$

i.e., Gibbs algorithm (Germain et al., 2009), where $\mathbb{Q}_{\boldsymbol{x}}$ is a distribution on $\mathcal{X}$, is also standard in its own right.

Usually, the regularized problem (25) is interpreted from the randomized learning perspective (27) and a prior distribution $\Pi_{\boldsymbol{x}}$ on $\mathcal{X}$, which encodes our prior belief on the hypothesis class $\mathcal{X}$, is used. For example, one may reminisce about the PAC-Bayesian theory, that is, the information empirical risk minimization model (Germain et al., 2016, Sec. 2)

$$\min_{\mathbb{Q}_{\boldsymbol{x}}} \mathbb{E}_{\boldsymbol{x}\sim\mathbb{Q}_{\boldsymbol{x}}}\mathbb{E}_{\xi\sim\hat{\mathbb{P}}_n}h(\boldsymbol{x},\xi) + \lambda\,\mathrm{KL}(\mathbb{Q}_{\boldsymbol{x}}||\Pi_{\boldsymbol{x}}), \tag{28}$$

where $\mathrm{KL}(\mathbb{Q}_{\boldsymbol{x}}||\Pi_{\boldsymbol{x}})$ denotes the Kullback–Leibler divergence of $\mathbb{Q}_{\boldsymbol{x}}$ from $\Pi_{\boldsymbol{x}}$. Note that when $\mathbb{Q}_{\boldsymbol{x}}$ is a point-mass distribution, the information empirical risk minimization model reduces to (25) and $f(\boldsymbol{x}) := -\log\pi(\boldsymbol{x})$, in which $\pi(\boldsymbol{x})$ is the density function of $\Pi_{\boldsymbol{x}}$ with respect to the Lebesgue measure. In this case, if $\Pi_{\boldsymbol{x}}$ is a Gaussian (resp. Laplacian) distribution, we have the 2-norm (resp. 1-norm) regularized model.

In contrast to the PAC-Bayesian viewpoint (28), this paper provides a new Bayesian probabilistic interpretation for regularized SAA methods if the condition (26) holds; see Proposition 3. The difference between

the two Bayesian perspectives lies in over which space we assign a prior distribution: The prior distribution $\mathbb{Q}_{\boldsymbol{x}}$ on the hypothesis class $\mathcal{X}$ or the prior distribution $\mathbb{Q}$ of the data distribution $\mathbb{P}$ in the distribution class $\mathcal{C}$; cf. (28) and (7). Hence, when we face distributional uncertainty in the nominal distribution $\bar{\mathbb{P}}$ for the true distribution $\mathbb{P}_0$, we have ***two*** possible ***Bayesian*** philosophies to hedge against it: The first one is to introduce a prior belief on the hypothesis to reduce the uncertainty (i.e., which hypotheses are relatively more important than others); the second one is to assign a prior belief on the nominal data distributions (i.e., which nominal data distributions are relatively more important than others). The interesting result is that the two ideas can lead to the same technical methodology—the regularized empirical risk minimization model (25).

The remark below summarizes a new insight for the statistical machine learning community.

**Remark 4** *When we combat the distributional uncertainty in the nominal distribution, considering prior knowledge of the nominal distribution class $\mathcal{C}$ can be as effective as considering prior knowledge of the hypothesis class $\mathcal{X}$.* □

This philosophy can also be supported by Bertsimas & Copenhaver (2018) through examining the (conditional) equivalence between regularization and robustness in linear and matrix regression; see, e.g., Corollary 1 therein. The difference is that the equivalence between regularization and robustness in Bertsimas & Copenhaver (2018) is not in the distributional/probabilistic but in the deterministic sense: There is no probability distribution involved and only norm-based uncertainty sets for real-valued parameters are discussed.

### 2.3.2   Understand Bias-Variance Trade-Off From Bayesian Perspective

Although regularized SAA methods are originally invented from the motivation of bias-variance trade-off (i.e., penalizing the complexity of the hypothesis class), we have shown that solving them is equivalent to solving Bayesian models, and therefore, regularized SAA methods are probably approximately distributionally robust optimization models. The bias-variance trade-off shows that by introducing a bias term $\lambda_n f(\boldsymbol{x})$ for an estimator, it is possible to reduce the variance of the estimator. This can be explicitly understood from (23): By introducing the bias term $\beta_n \mathbb{E}_{\bar{\mathbb{P}}} h(\boldsymbol{x}, \xi)$, the variance of $(1 - \beta_n)\mathbb{E}_{\hat{\mathbb{P}}_n} h(\boldsymbol{x}, \xi)$ reduces $(1 - \beta_n)^2$ times compared with $\mathbb{E}_{\hat{\mathbb{P}}_n} h(\boldsymbol{x}, \xi)$. To clarify further, for every $\boldsymbol{x}$, the mean of the objective of (23) is

$$
\begin{aligned}
\mathbb{E}_{\mathbb{P}_0^n}[\beta_n \mathbb{E}_{\bar{\mathbb{P}}} h(\boldsymbol{x}, \xi) + (1 - \beta_n)\mathbb{E}_{\hat{\mathbb{P}}_n} h(\boldsymbol{x}, \xi)] \quad &= \beta_n \mathbb{E}_{\bar{\mathbb{P}}} h(\boldsymbol{x}, \xi) + (1 - \beta_n)\mathbb{E}_{\mathbb{P}_0^n}\mathbb{E}_{\hat{\mathbb{P}}_n} h(\boldsymbol{x}, \xi) \\
&= \beta_n \mathbb{E}_{\bar{\mathbb{P}}} h(\boldsymbol{x}, \xi) + (1 - \beta_n)\mathbb{E}_{\mathbb{P}_0} h(\boldsymbol{x}, \xi),
\end{aligned}
$$

which is biased from the true mean $\mathbb{E}_{\mathbb{P}_0} h(\boldsymbol{x}, \xi)$. But the objective function of (23) has the variance that is $(1 - \beta_n)^2$ times lower than that of $\mathbb{E}_{\hat{\mathbb{P}}_n} h(\boldsymbol{x}, \xi)$. Note that if $\bar{\mathbb{P}}$ could be elegantly given, the performance gap of the model (23) would be smaller than that of the SAA model $\min_{\boldsymbol{x}} \mathbb{E}_{\hat{\mathbb{P}}_n} h(\boldsymbol{x}, \xi)$ because for every $\boldsymbol{x}$

$$
\begin{aligned}
&|\beta_n \mathbb{E}_{\bar{\mathbb{P}}} h(\boldsymbol{x}, \xi) + (1 - \beta_n)\mathbb{E}_{\hat{\mathbb{P}}_n} h(\boldsymbol{x}, \xi) - \mathbb{E}_{\mathbb{P}_0} h(\boldsymbol{x}, \xi)| \\
&\leq \beta_n |\mathbb{E}_{\bar{\mathbb{P}}} h(\boldsymbol{x}, \xi) - \mathbb{E}_{\mathbb{P}_0} h(\boldsymbol{x}, \xi)| + (1 - \beta_n)|\mathbb{E}_{\hat{\mathbb{P}}_n} h(\boldsymbol{x}, \xi) - \mathbb{E}_{\mathbb{P}_0} h(\boldsymbol{x}, \xi)| \\
&\leq |\mathbb{E}_{\hat{\mathbb{P}}_n} h(\boldsymbol{x}, \xi) - \mathbb{E}_{\mathbb{P}_0} h(\boldsymbol{x}, \xi)|,
\end{aligned}
\tag{29}
$$

if $|\mathbb{E}_{\bar{\mathbb{P}}} h(\boldsymbol{x}, \xi) - \mathbb{E}_{\mathbb{P}_0} h(\boldsymbol{x}, \xi)| \leq |\mathbb{E}_{\hat{\mathbb{P}}_n} h(\boldsymbol{x}, \xi) - \mathbb{E}_{\mathbb{P}_0} h(\boldsymbol{x}, \xi)|$ (i.e., $\bar{\mathbb{P}}$ is a good estimate of $\mathbb{P}_0$; $\bar{\mathbb{P}}$ is better than $\hat{\mathbb{P}}_n$). The remark below summarizes the main points above.

**Remark 5** *If $f(\boldsymbol{x})$ is a good regularizer and $\lambda_n$ is a good regularization coefficient (in the sense that they can induce a better $(\beta_n, \bar{\mathbb{P}})$ through $\lambda_n := \frac{\beta_n}{1 - \beta_n}$ and $f(\boldsymbol{x}) := \mathbb{E}_{\bar{\mathbb{P}}} h(\boldsymbol{x}, \xi))$, the regularized SAA model (25) has the potential to give lower generalization error than the standard SAA method because the former has a smaller robustness measure than that of the latter; cf. (29). Note that in the Bayesian setting (7), the probably approximately distributional robustness measure L in Proposition 2 and (46) can be seen as a type of upper bounds of generalization error gaps. (For details, see Subsection 2.4.)* □

### 2.4   Distributional Robustness and Generalization Error

In this subsection, we suppose that the empirical distribution $\hat{\mathbb{P}}_n$ is involved. Let $\bar{\mathbb{P}}$ be a surrogate of $\mathbb{P}_0$ constructed from data. For example, $\bar{\mathbb{P}}$ can be $\hat{\mathbb{P}}_n$ in an SAA model. For another example, $\bar{\mathbb{P}}$ can be

$\beta_n\hat{\mathbb{P}} + (1 - \beta_n)\hat{\mathbb{P}}_n$ in a regularized SAA (i.e., a Bayesian) model where $\hat{\mathbb{P}}$ is a prior belief of $\mathbb{P}_0$. Recall from Appendix C.2 that in the machine learning literature, the probably approximately correct (PAC) upper bound $L_{\boldsymbol{x},\eta}$ of the generalization error gap with probability at least $1 - \eta$, at a solution $\boldsymbol{x}$, is defined as

$$\mathbb{P}_0^n[\mathbb{E}_{\mathbb{P}_0}h(\boldsymbol{x},\xi) - \mathbb{E}_{\bar{\mathbb{P}}}h(\boldsymbol{x},\xi) \le L_{\boldsymbol{x},\eta}] \ge 1 - \eta$$

for the one-sided case and as

$$\mathbb{P}_0^n[|\mathbb{E}_{\mathbb{P}_0}h(\boldsymbol{x},\xi) - \mathbb{E}_{\bar{\mathbb{P}}}h(\boldsymbol{x},\xi)| \le L_{\boldsymbol{x},\eta}] \ge 1 - \eta$$

for the two-sided case. In addition, the upper bound $L_{\boldsymbol{x}}$ of the expected generalization error gap, at a solution $\boldsymbol{x}$, is defined as

$$\mathbb{E}_{\mathbb{P}_0^n}[\mathbb{E}_{\mathbb{P}_0}h(\boldsymbol{x},\xi) - \mathbb{E}_{\bar{\mathbb{P}}}h(\boldsymbol{x},\xi)] \le L_{\boldsymbol{x}}$$

or

$$\mathbb{E}_{\mathbb{P}_0^n}[|\mathbb{E}_{\mathbb{P}_0}h(\boldsymbol{x},\xi) - \mathbb{E}_{\bar{\mathbb{P}}}h(\boldsymbol{x},\xi)|] \le L_{\boldsymbol{x}}.$$

In this subsection, we are concerned with the uniform generalization error gap $|\mathbb{E}_{\mathbb{P}_0}h(\boldsymbol{x},\xi) - \mathbb{E}_{\bar{\mathbb{P}}}h(\boldsymbol{x},\xi)|$, for every $\boldsymbol{x}$, and the ad-hoc generalization error gap $|\mathbb{E}_{\mathbb{P}_0}h(\bar{\boldsymbol{x}},\xi) - \mathbb{E}_{\bar{\mathbb{P}}}h(\bar{\boldsymbol{x}},\xi)|$ of the nominal model $\min_{\boldsymbol{x}} \mathbb{E}_{\bar{\mathbb{P}}}h(\boldsymbol{x},\xi)$ at the solution $\bar{\boldsymbol{x}}$, where $\bar{\boldsymbol{x}} \in \operatorname{argmin}_{\boldsymbol{x}} \mathbb{E}_{\bar{\mathbb{P}}}h(\boldsymbol{x},\xi)$.

Usually, the generalization error bounds of machine learning models are specified by

1. Measure concentration inequalities such as Hoeffding's and Bernstein inequalities (Wainwright, 2019, Chaps. 2-3), McDiarmid's inequality (Zhang & Chen, 2021), Variation-Based Concentration (Gao, 2022, Thm. 1 and Cors. 1-2), among many others (Zhang & Chen, 2021). In this case, the properties of the cost function $h(\boldsymbol{x},\xi)$ such as the boundedness and Lipschitz-norm are leveraged;

2. Richness of hypothesis classes such as VC dimension and Rademacher complexity (Wainwright, 2019, Chap. 4);

3. PAC-Bayesian arguments through the KL-Divergence of the posterior hypothesis distribution from the prior hypothesis distribution (Germain et al., 2016, Sec. 2), etc.;

4. Information-theoretic methods through mutual information (Xu & Raginsky, 2017) or Wasserstein distance (Wang et al., 2019; Rodríguez Gálvez et al., 2021) between the posterior hypothesis distribution and the (training) data distribution, etc.

The PAC-Bayesian arguments describe the generalization error bounds of a machine learning model by the (statistical) difference between the prior distribution $\Pi_{\boldsymbol{x}}$ of the hypothesis class $\mathcal{X}$, as used in (28), and the optimal randomized decision $\mathbb{Q}_{\boldsymbol{x}}$ that solves the randomized-learning model $\min_{\mathbb{Q}_{\boldsymbol{x}}} \mathbb{E}_{\mathbb{Q}_{\boldsymbol{x}}}\mathbb{E}_{\mathbb{P}_0}h(\boldsymbol{x},\xi)$. In contrast, information-theoretic methods depict the generalization error bounds of a machine learning model by the (statistical) difference between the ***posterior*** distribution $\mathbb{Q}_{\boldsymbol{x}}$ of hypothesis and the training data distribution $\hat{\mathbb{P}}_n$; i.e., how much does the posterior distribution $\mathbb{Q}_{\boldsymbol{x}}$ of hypothesis depend on the training data $\hat{\mathbb{P}}_n$? In this subsection, we establish the generalization error bounds of a machine learning model using the (statistical) difference between the training data distribution $\hat{\mathbb{P}}_n$ (or the transformed training data distribution $\bar{\mathbb{P}}$) and the population data distribution $\mathbb{P}_0$, and using its distributional robustness measures. This is a new perspective to characterize and bound generalization errors, which justifies the practical benefits of distributional robust optimizations [i.e., (8), (14) and (15)] and further explains the generalization abilities of Bayesian methods (7) and regularized methods (9). The main motivation here is that the generalization errors of a machine learning model can be specified in several yet extremely different ways.

### 2.4.1 Uniform Generalization Error Through Distributional Uncertainty

The uniform generalization error bound is straightforward to establish due to the weak convergence of $\bar{\mathbb{P}}$ to $\mathbb{P}_0$, that is, $\Delta(\mathbb{P}_0, \bar{\mathbb{P}}) \to 0$, $\mathbb{P}_0^n$-almost surely (Weed & Bach, 2019), and the continuity of the linear functional $\mathbb{P} \mapsto \mathbb{E}_{\mathbb{P}}h(\boldsymbol{x},\xi)$, where $\Delta$ denotes the Wasserstein distance. (Note that the Wasserstein distance metrizes the weak topology on $\mathcal{M}(\Xi)$.) The fact below is standard and well-established in the mathematical statistics

literature. We borrow it to describe and characterize generalization errors from a new perspective, that is, the perspective of distributional uncertainty.[7]

**Fact 2 (Uniform Generalization Error Through Distributional Uncertainty)** *For every $\boldsymbol{x}$, if the cost function $h(\boldsymbol{x}, \cdot)$ is $L_{\boldsymbol{x}}$-Lipschitz continuous in $\xi$ on $\Xi$, we have*

$$|\mathbb{E}_{\mathbb{P}_0} h(\boldsymbol{x}, \xi) - \mathbb{E}_{\bar{\mathbb{P}}} h(\boldsymbol{x}, \xi)| \leq L_{\boldsymbol{x}} \Delta(\mathbb{P}_0, \bar{\mathbb{P}}), \quad \forall \boldsymbol{x},$$

$\mathbb{P}_0^n$*-almost surely, where $\Delta$ is the order-$1$ Wasserstein distance.*

**Proof.** The statement is adapted from Chen & Paschalidis (2020, Thm. 3.1.1); see Appendix L for the detailed proof based on the definition of Wasserstein distance. □

The remark below reveals the usefulness of Fact 2 in data-driven machine learning.

**Remark 6** *The smaller the distributional uncertainty in $\bar{\mathbb{P}}$ is, the smaller the generalization error gap is. In addition, the generalization error is also affected by the Lipschitz constant of the cost function, and therefore, a cost function that has a smaller Lipschitz constant is preferable; this explains why, for example, Huber's loss function is better than the square loss function because the former has a smaller Lipschitz constant.* □

Note that the variable $L$ (i.e., robustness measure) in Definition 2 and Definition 4 depends on $\boldsymbol{x}$ because they are simultaneously involved in a common optimization problem (12) and (14), respectively. However, the Lipschitz constant $L_{\boldsymbol{x}}$ in Fact 2 depends on $\boldsymbol{x}$ just due to the innate property of the cost function $h(\boldsymbol{x}, \xi)$.

**Remark 7** *The order-$1$ Wasserstein distance in Fact 2 can be replaced with any order-$p$ Wasserstein distance where $p \geq 2$ because for any two distributions $\mathbb{P}_1$ and $\mathbb{P}_2$, we have $W_p(\mathbb{P}_1, \mathbb{P}_2) \leq W_q(\mathbb{P}_1, \mathbb{P}_2)$ where $1 \leq p \leq q \leq \infty$. This fact is attributed to Givens & Shortt (1984, Prop. 3); see also Römisch & Schultz (1991, Rem. 2.2).* □

**Corollary 2** *For every $\boldsymbol{x}$, if the cost function $h(\boldsymbol{x}, \cdot)$ is $L_{\boldsymbol{x}}$-Lipschitz continuous in $\xi$ on $\Xi$, we have the expected uniform bound of the generalization error gap $\mathbb{E}_{\mathbb{P}_0^n} |\mathbb{E}_{\mathbb{P}_0} h(\boldsymbol{x}, \xi) - \mathbb{E}_{\bar{\mathbb{P}}} h(\boldsymbol{x}, \xi)| \leq L_{\boldsymbol{x}} \mathbb{E}_{\mathbb{P}_0^n} \Delta(\mathbb{P}_0, \bar{\mathbb{P}})$, for every $\boldsymbol{x}$.* □

The one-sided versions of Fact 2 and Corollary 2 are straightforward to claim by requiring (the one-sided Lipschitz continuity) $h(\boldsymbol{x}, \xi) - h(\boldsymbol{x}, \bar{\xi}) \leq L_{\boldsymbol{x}} \|\xi - \bar{\xi}\|$ for every $\xi, \bar{\xi} \in \Xi$. We do not give details in this paper.

A concrete example of Fact 2 is given as

$$|\mathbb{E}_{\mathbb{P}_0} h(\boldsymbol{x}, \xi) - \mathbb{E}_{\hat{\mathbb{P}}_n} h(\boldsymbol{x}, \xi)| \leq L_{\boldsymbol{x}} \Delta(\mathbb{P}_0, \hat{\mathbb{P}}_n), \quad \forall \boldsymbol{x},$$

$\mathbb{P}_0^n$-almost surely. The $L_{\boldsymbol{x}}$-Lipschitz continuity is not restrictive for a machine learning model because the cost function $h$ can be elegantly designed; cf. Wang et al. (2022a).

### 2.4.2 Generalization Error Through Distributional Robustness

In what follows, we focus on the generalization error of the nominal model $\min_{\boldsymbol{x}} \mathbb{E}_{\bar{\mathbb{P}}} h(\boldsymbol{x}, \xi)$ at its optimal solution $\bar{\boldsymbol{x}}$. Note that, in practical implementation of a machine learning model, we are more interested in the generalization capability of a specified hypothesis, for example, a regression model with already-trained coefficients.

**Proposition 4 (Generalization Error Through Absolute Robustness Measure)** *Suppose $\mathbb{P}_0$ is contained in $B_\epsilon(\bar{\mathbb{P}}) \coloneqq \{\mathbb{P} \in \mathcal{M}(\Xi) | \Delta(\mathbb{P}, \bar{\mathbb{P}}) \leq \epsilon\}$ and $(\boldsymbol{x}^*, L^*)$ solves the surrogate distributionally robust*

---

[7]Recall that the distributional uncertainty refers to the difference between $\bar{\mathbb{P}}$ and $\mathbb{P}_0$, i.e., the deviation of the nominal distribution from the underlying true distribution.

*optimization model* (14), *i.e.,*

$$\min_{\boldsymbol{x}, L} \quad L$$
$$s.t. \quad |\mathbb{E}_{\mathbb{P}}h(\boldsymbol{x}, \xi) - \mathbb{E}_{\bar{\mathbb{P}}}h(\bar{\boldsymbol{x}}, \xi)| \leq L, \quad \forall \mathbb{P} : \Delta(\mathbb{P}, \bar{\mathbb{P}}) \leq \epsilon,$$
$$\boldsymbol{x} \in \mathcal{X}.$$

*If $h(\boldsymbol{x}, \xi)$ is $L(\xi)$-Lipschitz continuous in $\boldsymbol{x}$ on $\mathcal{X}$, for every $\xi \in \Xi$, then the generalization error gap $|\mathbb{E}_{\mathbb{P}_0}h(\bar{\boldsymbol{x}}, \xi) - \mathbb{E}_{\bar{\mathbb{P}}}h(\bar{\boldsymbol{x}}, \xi)|$ of the nominal model $\min_{\boldsymbol{x}} \mathbb{E}_{\bar{\mathbb{P}}}h(\boldsymbol{x}, \xi)$ is upper bounded by*

$$\|\bar{\boldsymbol{x}} - \boldsymbol{x}^*\| \cdot \mathbb{E}_{\mathbb{P}_0}L(\xi) + L^*,$$

*$\mathbb{P}_0^n$-almost surely. In addition, the generalization error gap $|\mathbb{E}_{\mathbb{P}_0}h(\boldsymbol{x}^*, \xi) - \mathbb{E}_{\mathbb{P}^*}h(\boldsymbol{x}^*, \xi)|$ of the surrogate distributionally robust optimization model* (14) *is upper bounded by*

$$2L^*,$$

*$\mathbb{P}_0^n$-almost surely, where*

$$\mathbb{P}^* \in \underset{\mathbb{P}: \ \Delta(\mathbb{P}, \bar{\mathbb{P}}) \leq \epsilon}{\operatorname{argmax}} |\mathbb{E}_{\mathbb{P}}h(\boldsymbol{x}^*, \xi) - \mathbb{E}_{\bar{\mathbb{P}}}h(\bar{\boldsymbol{x}}, \xi)|.$$

*Note that the equality $|\mathbb{E}_{\mathbb{P}^*}h(\boldsymbol{x}^*, \xi) - \mathbb{E}_{\bar{\mathbb{P}}}h(\bar{\boldsymbol{x}}, \xi)| = L^*$ holds.*

**Proof.** See Appendix M for the proof based on telescoping and triangle inequalities. $\square$

Note that Proposition 4 implies another generalization error bound through absolute robustness measure $L^*$: i.e., $|\mathbb{E}_{\mathbb{P}_0}h(\boldsymbol{x}^*, \xi) - \mathbb{E}_{\bar{\mathbb{P}}}h(\bar{\boldsymbol{x}}, \xi)| \leq L^*$. Note also that Proposition 4 holds almost surely, not in the PAC sense, and therefore, generalization error bounds specified by absolute robustness measures are stronger than those specified in the PAC sense.

**Remark 8** *If $\mathbb{P}_0$ is contained in $B_\epsilon(\bar{\mathbb{P}}) := \{\mathbb{P} \in \mathcal{M}(\Xi) | \Delta(\mathbb{P}, \bar{\mathbb{P}}) \leq \epsilon\}$ in $\mathbb{P}_0^n$-probability, then Proposition 4 holds in $\mathbb{P}_0^n$-probability; e.g., see* (32) *in Appendix C.1.* $\square$

**Corollary 3 (Generalization Error Through Absolute Robustness Measure)** *Under the settings of Proposition 4, the expected generalization error gaps are given by*

$$\mathbb{E}_{\mathbb{P}_0^{n+1}}L(\xi)\|\bar{\boldsymbol{x}} - \boldsymbol{x}^*\| + \mathbb{E}_{\mathbb{P}_0^n}L^*,$$

*and $2\mathbb{E}_{\mathbb{P}_0^n}L^*$, respectively; $\bar{\boldsymbol{x}}$, $\boldsymbol{x}^*$, and $L^*$ depend on the specific choice of the training data.*

**Proof.** See Appendix N. $\square$

The term $|\mathbb{E}_{\mathbb{P}_0}h(\bar{\boldsymbol{x}}, \xi) - \mathbb{E}_{\mathbb{P}_0}h(\boldsymbol{x}^*, \xi)|$ can be upper bounded by other possible ways rather than the Lipschitz continuity of $h$. We use Lipschitz continuity just as an example. For a machine learning model, if $h$ is continuous in $\boldsymbol{x}$ and bounded on a compact feasible region $\mathcal{X}$, the Lipschitz continuity is naturally guaranteed. The continuity and boundedness condition is not practically restrictive; see, e.g., Wang et al. (2022a).

Proposition 4 and Corollary 3 reveal that whenever the DRO model (14) is solved, the (expected) generalization error gaps of the nominal model $\min_{\boldsymbol{x}} \mathbb{E}_{\bar{\mathbb{P}}}h(\boldsymbol{x}, \xi)$ and the surrogate distributionally robust optimization model (14) are also controlled by the absolute distributional robustness measure. The one-sided versions of Proposition 4 and Corollary 3 are straightforward to be developed and therefore omitted in this paper. Just note that

$$
\begin{aligned}
\mathbb{E}_{\mathbb{P}_0}h(\bar{\boldsymbol{x}}, \xi) - \mathbb{E}_{\bar{\mathbb{P}}}h(\bar{\boldsymbol{x}}, \xi) \quad &= \mathbb{E}_{\mathbb{P}_0}h(\bar{\boldsymbol{x}}, \xi) - \mathbb{E}_{\mathbb{P}_0}h(\boldsymbol{x}^*, \xi) + \mathbb{E}_{\mathbb{P}_0}h(\boldsymbol{x}^*, \xi) - \mathbb{E}_{\bar{\mathbb{P}}}h(\bar{\boldsymbol{x}}, \xi) \\
&\leq |\mathbb{E}_{\mathbb{P}_0}h(\bar{\boldsymbol{x}}, \xi) - \mathbb{E}_{\mathbb{P}_0}h(\boldsymbol{x}^*, \xi)| + \mathbb{E}_{\mathbb{P}_0}h(\boldsymbol{x}^*, \xi) - \mathbb{E}_{\bar{\mathbb{P}}}h(\bar{\boldsymbol{x}}, \xi) \\
&\leq |\mathbb{E}_{\mathbb{P}_0}h(\bar{\boldsymbol{x}}, \xi) - \mathbb{E}_{\mathbb{P}_0}h(\boldsymbol{x}^*, \xi)| + L^*,
\end{aligned}
$$

where $L^*$ is the one-sided absolute distributional robustness measure returned by (15).

### 2.4.3 Benefits of Bounding Generalization Errors By Robustness Measures

We summarize the power of the proposed surrogate DRO framework, i.e., (14) [and (37) in Appendix D.2], in reducing the generalization error of a data-driven machine learning model: By conducting distributionally robust optimization, the robustness measures, which build the upper bounds of generalization errors (see Proposition 4 in the main body and Proposition 6 in Appendix D.2), are reduced. To be short,

> **distributionally robust optimization reduces generalization errors**.

In addition, the remark below explains the rationale of the Bayesian method (7), which is another benefit of the proposed distributionally optimization framework in (14).

**Remark 9** *In the literature, the generalization errors and the rationale of the regularization method* (9) *are well studied through, e.g., the measure concentration inequalities, the richness of hypothesis classes, the PAC-Bayesian arguments, and the information-theoretic methods; for details and references, see the beginning of Subsection 2.4. The **one-sided** generalization error of the (min-max) DRO method* (8) *is also well established in, e.g., Kuhn et al. (2019); Shafieezadeh-Abadeh et al. (2019). However, the generalization errors of the Bayesian method* (7) *have not been systematically investigated and the rationale of the Bayesian method (i.e., why can it generalize well?) has not been rigorously explained. In this paper, we put the Bayesian method* (7)*, the (min-max) DRO method* (8)*, and the regularization method* (9) *in a unified and generalized DRO framework [i.e, (14)] and show that the three methods are distributionally robust in the sense of Definition 1. Since distributional robustness bounds generalization errors (cf. Proposition 4), the Bayesian method* (7)*, the (min-max) DRO method* (8)*, and the regularization method* (9) *are shown to generalize well in the unified distributional robustness sense. This explains the power and the rationale of the Bayesian method* (7)*; cf. Proposition 2 and Remark 2.* □

### 2.5 Comparisons With Existing Literature

### 2.5.1 Comparisons With Existing DRO Literature

In the existing DRO literature, only the min-max distributionally robust optimization model (8) is studied. The main motivation of (8) is that by minimizing the worst-case cost, the true cost is also expected to be put down (Kuhn et al., 2019). For details, see Remark 10 below, which is well-established in, e.g., Mohajerin Esfahani & Kuhn (2018); Kuhn et al. (2019); Shafieezadeh-Abadeh et al. (2019); Chen & Paschalidis (2020); it characterizes the one-sided generalization error using Wasserstein min-max distributionally robust optimization.

**Remark 10 (cf. Appendix C.1)** *Recall the measure concentration property in the Wasserstein sense: we have, for every $\boldsymbol{x}$,*

$$\mathbb{P}_0^n\left[\mathbb{E}_{\mathbb{P}_0}h(\boldsymbol{x},\xi) \leq \max_{\mathbb{P}\in B_{\epsilon_n,p}(\hat{\mathbb{P}}_n)} \mathbb{E}_{\mathbb{P}}h(\boldsymbol{x},\xi)\right] \geq 1-\eta.$$

*This is because with probability at least $1-\eta$, $\mathbb{P}_0$ is in $B_{\epsilon_n,p}(\hat{\mathbb{P}}_n)$, and therefore, $\mathbb{E}_{\mathbb{P}_0}h(\boldsymbol{x},\xi) \leq \max_{\mathbb{P}\in B_{\epsilon_n,p}(\hat{\mathbb{P}}_n)} \mathbb{E}_{\mathbb{P}}h(\boldsymbol{x},\xi)$, for every $\boldsymbol{x}$. Let $\boldsymbol{x}^* \in \arg\min_{\boldsymbol{x}} \max_{\mathbb{P}\in B_{\epsilon_n,p}(\hat{\mathbb{P}}_n)} \mathbb{E}_{\mathbb{P}}h(\boldsymbol{x},\xi)$ denote the min-max distributionally robust solution. We have*

$$\mathbb{P}_0^n\left[\mathbb{E}_{\mathbb{P}_0}h(\boldsymbol{x}^*,\xi) \leq \min_{\boldsymbol{x}} \max_{\mathbb{P}\in B_{\epsilon_n,p}(\hat{\mathbb{P}}_n)} \mathbb{E}_{\mathbb{P}}h(\boldsymbol{x},\xi)\right] \geq 1-\eta,$$

*which is the one-sided generalization error bound at $\boldsymbol{x}^*$.* □

According to Appendix C.3, there exists conditional equivalence between the min-max DRO model (8) and the regularization model (9); see Mohajerin Esfahani & Kuhn (2018); Shafieezadeh-Abadeh et al. (2019); Blanchet et al. (2019); Gao (2022) for more technical details. Therefore, by Remark 10, we have another uniform generalization error bound in the PAC sense:

$$\mathbb{E}_{\mathbb{P}_0}h(\boldsymbol{x},\xi) \leq \mathbb{E}_{\hat{\mathbb{P}}_n}h(\boldsymbol{x},\xi) + \epsilon_n f(\boldsymbol{x}), \quad \forall \boldsymbol{x}$$

where $f(\boldsymbol{x})$ is a proper regularizer constructed by the cost function $h(\boldsymbol{x}, \xi)$; see, e.g., Gao (2022, Cors. 1-2).

In contrast, this paper handles the problems from a new perspective: We generalize the concept system of "distributionally robust optimization" (from the existing "min-max distributionally robust optimization") and then propose to use "robustness measures" to bound generalization errors (rather than employ the worst-case costs as in Remark 10). The benefits are three-fold:

1. The concept of "robustness" (i.e., small model perturbations render small cost changes) is not explicitly conveyed by min-max distributionally robust optimization models (8). However, the proposed concept system of "robustness", e.g., Definition 1, addresses this issue.

2. The proposed concept system of "distributional robustness" allows to build connections among Bayesian models (7), (min-max) distributionally robust optimization models (8), and regularization models (9). As a result, the rationales of Bayesian models (7) and regularization models (9) can also be justified from the perspective of distributional robustness.

3. The min-max distributionally robust optimization models (8) can only provide one-sided robustness; similarly, min-max distributionally robust optimization models (8) can only provide one-sided generalization error bounds (cf. Remark 10). However, the proposed concept system of robustness is able to offer two-sided robustness, and generalization error bounds specified by robustness measures have both one-sided and two-sided versions.

### 2.5.2 Comparisons With Algorithmic-Stability Literature

In statistical machine learning literature, the generalization errors can also be bounded by the stability of learning algorithms (Bousquet & Elisseeff, 2002), in addition to the measure concentration inequalities, the richness of hypothesis classes, the PAC-Bayesian arguments, and the information-theoretic methods. The notion of "distributional deviation" is also employed in the definition of "stability" of a learning algorithm. However, the following differences should be highlighted:

1. In stability analyses of learning algorithms, distributional deviations (of the training data) are only limited to single sample deletion and single sample replacement (Bousquet & Elisseeff, 2002, Sec. 3). However, in the proposed distributionally robust learning framework, the distributional deviations (of the training data) can be arbitrarily characterized using, e.g., Wasserstein balls and Kullback-Leibler divergence balls.

2. In stability analyses of learning algorithms, distributional deviations (of the training data) lead to different costs and different hypotheses; cf. Bousquet & Elisseeff (2002, Eq. (5)) where different learned hypotheses are associated with different training data distributions. However, in the proposed distributionally robust learning framework, the distributional deviations (of the training data) result in different costs but different training data distributions share the same robust hypothesis; cf. Definition 1.

## 3 Discussions and Conclusions

This paper studies a concept system of "distributional robustness", and the connections among Bayesian methods, distributionally robust optimization methods, and regularization methods are established; for a detailed and informative summary, see Appendix D.7. In highlights,

1. Under the selection of a Dirichlet-process prior in Bayesian nonparametrics (7), any regularization method (9) is equivalent to a Bayesian method (7) (cf. Proposition 3); in a general setting where the non-parametric prior in (7) is not a Dirichlet process, a regularization method (9) is a special case of a Bayesian method (7) (cf. Remark 3). This finding gives the regularization method (9) a new Bayesian interpretation (cf. Proposition 3 and Remark 4).

2. Bayesian models (7) are shown to be probably approximately distributionally robust (cf. Proposition 2), so are regularization methods (9) (cf. Proposition 3).

3. Min-max DRO models (8) are shown to be realizations of the proposed distributionally robust optimization (cf. Proposition 1 and Corollary 1).

As a result, several practically useful insights in trustworthy machine learning can be obtained, including

1. the robustness-sensitivity trade-off in dealing with distributional uncertainty (cf. Remark 1),

2. the power of bounding generalization error through Bayesian nonparametrics methods (7) (cf. Proposition 2 and Remark 2),

3. the data augmentation effect through Bayesian nonparametrics (7) (cf. Example 5) and regularization (9) (cf. Example 6),

4. the effects of priors on data distribution class and hypothesis class (cf. Remark 4),

5. the bias-variance trade-off in regularization (cf. Remark 5),

6. the benefits of small distributional uncertainty and the Lipschitz continuity of cost functions (cf. Remark 6), and

7. the unified interpretation (through robustness measures) for the Bayesian method, the DRO method, and the regularized method (cf. Figure 1 and Remark 9).

For concrete illustrating examples of these theoretical results, see Appendix Q.

In addition, a new perspective to characterize generalization errors of machine learning models is shown: i.e., generalization errors can be characterized using the distributional uncertainty of the nominal model (cf. Fact 2) or using the robustness measures of robust solutions (cf. Proposition 4 and Corollary 3 in the main body, and Proposition 6 and Corollary 4 in Appendix D.2). Generalization error bounds specified by robustness measures justify the rationale of distributionally robust optimization models (8), (14) and (15): by conducting distributionally robust optimizations that minimize robustness measures, generalization errors are also reduced; see Propositions 4 and 6. For additional closing notes on this point, see Appendix R.

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

# A    Interpretations of (1) in Other Areas

The optimization problem (1) is also popular in several other areas than machine learning where the specific meanings that it conveys vary from one to another. Non-exhaustive examples are as follows.

1. In applied statistics, (1) can be an M-estimation model where $h$ is termed a random criterion function and $\boldsymbol{x}$ is usually the parameter of the distribution $\mathbb{P}_0$ such as the location parameter (Huber, 1964; Huber & Ronchetti, 2009), (Van der Vaart, 2000, Chap. 5). The parameter $\boldsymbol{x}$ is unknown and to be estimated from $n$ i.i.d. observations $\{\xi_i\}_{i\in[n]}$.

2. In operations research and management science, (1) is a stochastic programming model (Shapiro et al., 2009; Anderson & Nguyen, 2020), where $h$ is the cost function such as mean(-variance) objective (Blanchet et al., 2021; Gotoh et al., 2021) and value-at-risk (VaR) objective (Shapiro et al., 2009, p. 16); VaR can be reformulated to the form of (1). Typical examples include the portfolio selection problem and the inventory control problem (Shapiro et al., 2009). Specifically, taking the one-stage inventory control problem as an example, $\xi$ represents the random demand and $\boldsymbol{x}$ is the optimal ordering quantity.

3. In statistical signal processing, (1) can be a state estimation model (Anderson & Moore, 1979, Chap. 2), (Hassibi et al., 1996, Eq. (6)), (Wang, 2022b, pp. 111), in which the unobservable state is to be estimated from the observable measurements. Specifically, $\xi := (\boldsymbol{I}, \boldsymbol{O})$ where $\boldsymbol{I}$ is the state vector and $\boldsymbol{O}$ is the measurement vector; $\boldsymbol{x}$ is the parameter of a state estimator which is a function from $\boldsymbol{O}$ to $\boldsymbol{I}$. State estimation problems are also popular in the machine learning community, for example, inference problems for a hidden Markov process (from observable variables $\boldsymbol{O}$ to hidden variables $\boldsymbol{I}$); see, e.g., Bishop & Nasrabadi (2006, Chap. 13).

When the true underlying distribution $\mathbb{P}_0$ is unknown, it can be estimated from observations $\{\xi_i\}_{i\in[n]}$ (usually i.i.d.) and this class of problems is termed data-driven problems in the literature (Kuhn et al., 2019). Most applied statistics problems, operations research problems, and supervised machine learning problems belong to this category. The distribution $\mathbb{P}_0$ may alternatively be obtained from physics (e.g., from a hidden Morkov process model) and we term this type of problem as ***model-driven*** problems in this paper. Most signal processing problems (e.g., state estimation problems) and engineering automatic control problems (Van Parys et al., 2015), among many others, are members of this class.

In this paper, we collectively refer to the two cases as machine learning problems and distinguish them as data-driven machine learning problems and model-driven machine learning problems. This is because the two kinds of problems share the same philosophy that a proportion of data is used to predict the rest by leveraging the distribution $\mathbb{P}_0$ (or an estimate of $\mathbb{P}_0$). For example, in the data-driven setting, the feature data $\boldsymbol{I}$ can be used to produce the predicted response $\hat{\boldsymbol{O}}$ such that the predicted response $\hat{\boldsymbol{O}}$ is close to the

true response $\boldsymbol{O}$. For another example, in the model-driven setting, the measurement $\boldsymbol{O}$ can be used to produce the estimated state $\hat{\boldsymbol{I}}$ such that the estimated state $\hat{\boldsymbol{I}}$ is close to the true state $\boldsymbol{I}$. However, a model-driven problem can be transformed into a data-driven counterpart if the integral $\mathbb{E}_{\mathbb{P}_0} h(\boldsymbol{x}, \xi)$ is hard to be analytically evaluated, and therefore, a Monte–Carlo sampling technique (e.g., importance sampling) is used to simulate data from $\mathbb{P}_0$ and then approximate the integral by a $n$-sample weighted sum $\sum_{i=1}^{n} \mu_i h(\boldsymbol{x}, \xi_i)$ (Bishop & Nasrabadi, 2006, Chap. 11), where $\mathbb{P}_0$ is approximated by a discrete distribution $\sum_{i=1}^{n} \mu_i \delta_{\xi_i}$ and $\delta_{\xi_i}$ is the Dirac measure at $\xi_i$; $\mu_i$ is the weight of the sample $\xi_i$. An excellent example of using Monte–Carlo sampling to transform a model-driven problem to a data-driven counterpart is the particle filter for state estimation of a hidden Markov model (Bishop & Nasrabadi, 2006, Sec. 13.3.4), (Wang, 2022a). Hence, without loss of practical generality, it is sufficient to investigate only the data-driven case, which is the technical focus of this paper.

## B  Supplementary Literature Review on Data-Driven ERM Model

The generalization performance of ERM is unsatisfactory due to the over-fitting issue on limited data samples.[8] Bayesian methods are potential in reducing the generalization errors (Wu et al., 2018; Anderson & Nguyen, 2020). Regularized SAA methods are also popular to combat over-fitting and reduce the generalization errors (Goodfellow et al., 2016; Shafieezadeh-Abadeh et al., 2019; Germain et al., 2016). The DRO methods can provide a generalization error bound that is independent of the complexity of the hypothesis class (e.g., Vapnik–Chervonenkis dimension, Rademacher complexity) and even applicable for hypothesis classes that have infinite Vapnik–Chervonenkis (VC) dimensions (Shafieezadeh-Abadeh et al., 2019). An exciting property of the DRO method is that, under some conditions, it is equivalent to a regularized empirical risk minimization method (Shafieezadeh-Abadeh et al., 2019; Gao et al., 2022), (Mohajerin Esfahani & Kuhn, 2018, Thm. 6.3), (Kuhn et al., 2019, Thm. 10), (Chen & Paschalidis, 2020, Chap. 4), which explains why the DRO method can generalize well.

## C  Technical Preliminaries

This section summarizes some existing results about ERM and DRO models that are important to motivate and justify the new results in this paper. For those that are less motivational or not frequently referred to, we just provide citations in proper positions.

### C.1  Statistical Similarity Measures and Distributional Balls

#### C.1.1  $\phi$-Divergence Distributional Ball

If for every $\mathbb{P} \in \mathcal{M}(\Xi)$, $\mathbb{P}$ is absolutely continuous with respect to $\bar{\mathbb{P}}$,[9] we can define the $\phi$-divergence (i.e., $f$-divergence) of $\mathbb{P}$ from $\bar{\mathbb{P}}$:

$$F_\phi(\mathbb{P}\|\bar{\mathbb{P}}) = \int_\Xi \phi\left(\frac{\mathrm{d}\mathbb{P}}{\mathrm{d}\bar{\mathbb{P}}}\right) \bar{\mathbb{P}}(\mathrm{d}\xi), \tag{30}$$

where $\phi : [0, \infty) \to (-\infty, \infty]$ is a convex function such that $\phi(1) = 0$ and $0\phi(0/0) = 0$; $\mathrm{d}\mathbb{P}/\mathrm{d}\bar{\mathbb{P}}$ is a Radon-Nikodym derivative. When $\phi(t) := t \ln t$, $\forall t > 0$, the $\phi$-divergence specifies the Kullback–Leibler (KL) divergence. Other choices for $\phi(\cdot)$ may be found in, e.g., Rahimian & Mehrotra (2022, Table 3), Ben-Tal et al. (2013, Table 2).

A $\phi$-divergence distributional ball, induced by the function $\phi$, is a set of distributions on $(\Xi, \mathcal{B}_\Xi)$ that are close to the reference distribution $\bar{\mathbb{P}}$ and defined as

$$B_{\epsilon,\phi}(\bar{\mathbb{P}}) := \{\mathbb{P} \in \mathcal{M}(\Xi) | F_\phi(\mathbb{P}\|\bar{\mathbb{P}}) \le \epsilon\},$$

---

[8] The phenomenon of "over-fitting" in applied statistics and machine learning is also known as "optimizer's curse" in operations research.

[9] Absolute continuity implies that the support set of $\mathbb{P}$ is no larger than that of $\bar{\mathbb{P}}$.

where $\epsilon \geq 0$ is the radius of the ball. When $\epsilon = 0$, the distributional ball $\mathcal{B}_{\epsilon,\phi}(\bar{\mathbb{P}})$ only contains the singleton $\bar{\mathbb{P}}$. Some authors may define a $\phi$-divergence ball as

$$B_{\epsilon,\phi}(\bar{\mathbb{P}}) \coloneqq \{\mathbb{P} \in \mathcal{M}(\Xi) | F_\phi(\bar{\mathbb{P}} \| \mathbb{P}) \leq \epsilon\}$$

which exchanges the order of $\mathbb{P}$ and $\bar{\mathbb{P}}$; see, e.g., Van Parys et al. (2021). Since a $\phi$-divergence is not necessarily a metric, the two definitions are not equivalent.

### C.1.2 Wasserstein Distributional Ball and Its Concentration Property

Let $(\Xi, d)$ be a metric space. Suppose $\mathbb{P}$ and $\bar{\mathbb{P}}$ are two probability measures on $(\Xi, \mathcal{B}_\Xi)$. The order-$p$ Wasserstein distance between $\mathbb{P}$ and $\bar{\mathbb{P}}$, induced by the metric $d$, is defined by

$$W_p(\mathbb{P}, \bar{\mathbb{P}}) = \left[ \inf_{\pi \in \mathcal{M}(\Xi \times \Xi)} \mathbb{E}_\pi d^p(\xi_1, \xi_2) \right]^{\frac{1}{p}} = \left[ \inf_{\pi \in \mathcal{M}(\Xi \times \Xi)} \int_{\Xi \times \Xi} d^p(\xi_1, \xi_2) \pi(\mathrm{d}\xi_1, \mathrm{d}\xi_2) \right]^{\frac{1}{p}}, \tag{31}$$

where $p \geq 1$ and $\pi$ is a coupling of $\mathbb{P}$ and $\bar{\mathbb{P}}$. Usually, the metric $d$ is induced by a norm $\| \cdot \|$ on $\Xi$.

An order-$p$ Wasserstein distributional ball is a set of distributions on $(\Xi, \mathcal{B}_\Xi)$ that are close to the reference distribution $\bar{\mathbb{P}}$ and defined as

$$B_{\epsilon,p}(\bar{\mathbb{P}}) \coloneqq \{\mathbb{P} \in \mathcal{M}(\Xi) | W_p(\mathbb{P}, \bar{\mathbb{P}}) \leq \epsilon\},$$

where $\epsilon \geq 0$ is the radius of the ball. When $\epsilon = 0$, the distributional ball $B_{\epsilon,p}(\bar{\mathbb{P}})$ only contains the singleton $\bar{\mathbb{P}}$.

Wasserstein distributional balls have concentration properties in the data-driven setting. Suppose the true underlying distribution $\mathbb{P}_0$ is light-tailed: i.e., there exist $\alpha > p \geq 1$ (where $p \neq m/2$ and $m$ is the dimension of $\xi$) and $0 < A < \infty$ such that $\mathbb{E}_{\mathbb{P}_0} [\exp(\|\xi\|^\alpha)] \leq A$. Then, there exist constants $c_1, c_2 > 0$ that depend on $\mathbb{P}_0$ only through $\alpha$, $A$, and $m$ such that for any $\eta \in (0, 1]$ the concentration inequality

$$\mathbb{P}_0^n \left[ \mathbb{P}_0 \in B_{\epsilon_n,p}(\hat{\mathbb{P}}_n) \right] \geq 1 - \eta \tag{32}$$

holds if

$$\epsilon_n \geq \begin{cases} \left( \frac{\log(c_1/\eta)}{c_2 n} \right)^{\min\{p/m, 1/2\}} & \text{if } n \geq \frac{\log(c_1/\eta)}{c_2}, \\ \left( \frac{\log(c_1/\eta)}{c_2 n} \right)^{p/\alpha} & \text{if } n < \frac{\log(c_1/\eta)}{c_2}. \end{cases}$$

This result is reported in Kuhn et al. (2019, Thm. 18). However, this concentration bound is more a theoretical than a practical result because for an unknown distribution $\mathbb{P}_0$, we do not know the associated constants $\alpha$ and $A$ in the light-tail assumption (so that $c_1$ and $c_2$ are unknown).

When the support set $\Xi$ is finite and bounded (i.e., $\mathbb{P}_0$ is discrete), there exist concentration properties of $\hat{\mathbb{P}}_n$ with respect to the Wasserstein distance that do not depend on unknown constants; see, e.g., Chen & Paschalidis (2020, pp. 42).

## C.2 Overfitting and Generalization Error

Consider the Sample-Average Approximation (SAA) model with $n$ i.i.d. samples:

$$\min_{\boldsymbol{x}} \mathbb{E}_{\hat{\mathbb{P}}_n} h(\boldsymbol{x}, \xi) = \min_{\boldsymbol{x}} \frac{1}{n} \sum_{i=1}^n h(\boldsymbol{x}, \xi_i).$$

To avoid notational clutter, unless stated otherwise in the following contexts, we implicitly mean that the feasible region of $\boldsymbol{x}$ is $\mathcal{X}$. That is, the minimization is conducted over $\boldsymbol{x} \in \mathcal{X}$. We have

$$\mathbb{E}_{\mathbb{P}_0^n} \left[ \min_{\boldsymbol{x}} \mathbb{E}_{\hat{\mathbb{P}}_n} h(\boldsymbol{x}, \xi) \right] = \mathbb{E}_{\mathbb{P}_0^n} \left[ \min_{\boldsymbol{x}} \frac{1}{n} \sum_{i=1}^n h(\boldsymbol{x}, \xi_i) \right]$$

$$\leq \left[ \min_{\boldsymbol{x}} \frac{1}{n} \sum_{i=1}^n \mathbb{E}_{\mathbb{P}_0} h(\boldsymbol{x}, \xi_i) \right] = \min_{\boldsymbol{x}} \mathbb{E}_{\mathbb{P}_0} h(\boldsymbol{x}, \xi).$$

Suppose $\hat{\boldsymbol{x}}_n \in \operatorname{argmin}_{\boldsymbol{x}} \mathbb{E}_{\hat{\mathbb{P}}_n} h(\boldsymbol{x}, \xi)$ and $\boldsymbol{x}_0 \in \operatorname{argmin}_{\boldsymbol{x}} \mathbb{E}_{\mathbb{P}_0} h(\boldsymbol{x}, \xi)$. We have

$$\mathbb{E}_{\mathbb{P}_0^n} \mathbb{E}_{\hat{\mathbb{P}}_n} h(\hat{\boldsymbol{x}}_n, \xi) \leq \mathbb{E}_{\mathbb{P}_0} h(\boldsymbol{x}_0, \xi) \leq \mathbb{E}_{\mathbb{P}_0} h(\hat{\boldsymbol{x}}_n, \xi).$$

Hence, at the decision $\hat{\boldsymbol{x}}_n$, there is always a performance gap

$$\mathbb{E}_{\mathbb{P}_0^n} \left[ \mathbb{E}_{\mathbb{P}_0} h(\hat{\boldsymbol{x}}_n, \xi) - \mathbb{E}_{\hat{\mathbb{P}}_n} h(\hat{\boldsymbol{x}}_n, \xi) \right] \geq 0$$

between the SAA optimization model $\min_{\boldsymbol{x}} \mathbb{E}_{\hat{\mathbb{P}}_n} h(\boldsymbol{x}, \xi)$ and the true optimization model $\min_{\boldsymbol{x}} \mathbb{E}_{\mathbb{P}_0} h(\boldsymbol{x}, \xi)$. In machine learning, this gap is termed "expected (or average) generalization error gap" from the training error of the SAA model,[10] which is a quantitative measure of "overfitting" of the SAA model that is aligned to the given training data set $\{\xi_i\}_{i \in [n]}$. In operations research, this gap is termed "optimizer's curse"[11] because the "optimal" cost estimated by the SAA model is not reachable in practice; the expected true cost is always larger than the expected SAA-estimated cost. Note that, for example, in business decision making, we allow the predicted budget to be larger than the true overhead. But the situation of the budget crisis is dangerous; this is where the "curse" happens.

This gap monotonically decreases as $n$ gets larger (Shapiro et al., 2009, Prop. 5.6):

$$\mathbb{E}_{\mathbb{P}_0^n} \mathbb{E}_{\hat{\mathbb{P}}_n} h(\hat{\boldsymbol{x}}_n, \xi) \leq \mathbb{E}_{\mathbb{P}_0^{n+1}} \mathbb{E}_{\hat{\mathbb{P}}_{n+1}} h(\hat{\boldsymbol{x}}_{n+1}, \xi), \quad \forall n.$$

When $n$ tends to infinity, the gap disappears. Therefore, the SAA model is asymptotically optimal and is the best choice if $n$ is sufficiently large (Anderson & Nguyen, 2020).

The SAA method has the following type of $(\epsilon, \eta)$-PAC concentration property:

$$\mathbb{P}_0^n [\mathbb{E}_{\mathbb{P}_0} h(\hat{\boldsymbol{x}}_n, \xi) - \mathbb{E}_{\hat{\mathbb{P}}_n} h(\hat{\boldsymbol{x}}_n, \xi) \leq \epsilon] \geq 1 - \eta,$$

which is termed PAC generalization error bound and might be specified by, e.g., Hoeffding's inequality and Bernstein inequality (Wainwright, 2019, Chaps. 2-3), McDiarmid's inequality (Zhang & Chen, 2021), PAC-Bayesian bounds (Germain et al., 2016, Sec. 2), information-theoretical bounds (Xu & Raginsky, 2017; Wang et al., 2019; Rodríguez Gálvez et al., 2021), among many others (Zhang & Chen, 2021). This property is attributed to the weak convergence of the empirical probability measure $\hat{\mathbb{P}}_n$ to the true underlying measure $\mathbb{P}_0$ (Weed & Bach, 2019; Wainwright, 2019; Van der Vaart & Wellner, 1996).

In machine learning, one might be interested in uniform generalization error bound:

$$\mathbb{P}_0^n [\mathbb{E}_{\mathbb{P}_0} h(\boldsymbol{x}, \xi) - \mathbb{E}_{\hat{\mathbb{P}}_n} h(\boldsymbol{x}, \xi) \leq \epsilon] \geq 1 - \eta, \quad \forall \boldsymbol{x},$$

where $\eta$ is independent of $\boldsymbol{x}$ but $\epsilon$ may depend on $\boldsymbol{x}$. The uniform generalization error bound is usually useful in model selection (i.e., model comparison): The hypothesis $\boldsymbol{x}$ at which the value of $\mathbb{E}_{\hat{\mathbb{P}}_n} h(\boldsymbol{x}, \xi) + \epsilon_{\boldsymbol{x}}$ is smaller is better. In applied statistics and operations research, the generalization error gap may also be defined as the difference between the true cost $\mathbb{E}_{\mathbb{P}_0} h(\bar{\boldsymbol{x}}, \xi)$ and the estimated cost $\mathbb{E}_{\bar{\mathbb{P}}} h(\bar{\boldsymbol{x}}, \xi)$, which is an estimate of $\mathbb{E}_{\mathbb{P}_0} h(\bar{\boldsymbol{x}}, \xi)$, at the given optimal decision $\bar{\boldsymbol{x}}$:

$$\Pr[\mathbb{E}_{\mathbb{P}_0} h(\bar{\boldsymbol{x}}, \xi) - \mathbb{E}_{\bar{\mathbb{P}}} h(\bar{\boldsymbol{x}}, \xi) \leq \epsilon] \geq 1 - \eta,$$

where $\bar{\mathbb{P}}$ is any estimate of $\mathbb{P}_0$, not limited to $\hat{\mathbb{P}}_n$, and $\bar{\boldsymbol{x}} \in \operatorname{argmin}_{\boldsymbol{x}} \mathbb{E}_{\bar{\mathbb{P}}} h(\boldsymbol{x}, \xi)$; cf. (2), (5), (7), (8), and (9).

Two-sided versions of generalization error gaps, for example,

$$\mathbb{P}_0^n [|\mathbb{E}_{\mathbb{P}_0} h(\hat{\boldsymbol{x}}_n, \xi) - \mathbb{E}_{\hat{\mathbb{P}}_n} h(\hat{\boldsymbol{x}}_n, \xi)| \leq \epsilon] \geq 1 - \eta,$$

are straightforward to be stated and therefore omitted in this subsection.

Since all these definitions for generalization errors are meaningful and practical in their own rights [see, e.g., Bousquet & Elisseeff (2002)], readers should be careful about the specific meaning of the term "generalization error" whenever it appears in this paper (and other literature in different areas).

---

[10]The random variable $\mathbb{E}_{\mathbb{P}_0} h(\hat{\boldsymbol{x}}_n, \xi) - \mathbb{E}_{\hat{\mathbb{P}}_n} h(\hat{\boldsymbol{x}}_n, \xi)$ is called the generalization error gap of the model aligned to the given training data set $\{\xi_i\}_{i \in [n]}$.

[11]In the operations research literature, some authors may also refer to the performance gap $\mathbb{E}_{\mathbb{P}_0} h(\boldsymbol{x}_0, \xi) - \mathbb{E}_{\mathbb{P}_0^n} \mathbb{E}_{\hat{\mathbb{P}}_n} h(\hat{\boldsymbol{x}}_n, \xi) \geq 0$ as the "optimizer's curse".

### C.3 Connection Between Wasserstein DRO Models and Regularized SAA Models

There exists a close connection between the Wasserstein DRO model (8) and the regularized SAA model (9) under some conditions. For every given $\boldsymbol{x}$, consider the inner maximization sub-problem of the Wasserstein DRO problem (8):

$$
\begin{aligned}
\max_{\mathbb{P}} \quad & \int_{\Xi} h(\boldsymbol{x}, \xi) \mathbb{P}(\mathrm{d}\xi) \\
\text{s.t.} \quad & W_p(\mathbb{P}, \bar{\mathbb{P}}) \leq \epsilon.
\end{aligned}
\tag{33}
$$

The following result is attributed to Mohajerin Esfahani & Kuhn (2018, Thm. 6.3). If $h$ is convex in $\xi \in \Xi \subseteq \mathbb{R}^m$, $\Xi$ is closed and convex, $p = 1$, $d$ is induced by a norm $\|\cdot\|$, and $\bar{\mathbb{P}} \coloneqq \hat{\mathbb{P}}_n$ (i.e., the data-driven setting), then the DRO objective function in (33) is point-wisely upper-bounded by a regularized SAA model:

$$
\max_{\mathbb{P}: W_p(\mathbb{P}, \bar{\mathbb{P}}) \leq \epsilon} \mathbb{E}_{\mathbb{P}} h(\boldsymbol{x}, \xi) \leq \epsilon \cdot f(\boldsymbol{x}) + \frac{1}{n} \sum_{i=1}^{n} h(\boldsymbol{x}, \xi_i), \quad \forall \boldsymbol{x},
\tag{34}
$$

where

$$
f(\boldsymbol{x}) \coloneqq \max_{\boldsymbol{\theta} \in \Xi} \{\|\boldsymbol{\theta}\|_* : h^*(\boldsymbol{x}, \boldsymbol{\theta}) < \infty\},
$$

is a regularizer, $\|\cdot\|_*$ is the dual norm of the norm $\|\cdot\|$, and $h^*(\boldsymbol{x}, \boldsymbol{\theta})$ is the point-wise convex conjugate function of the function $h(\boldsymbol{x}, \xi)$ for every fixed $\boldsymbol{x}$. If $h(\boldsymbol{x}, \xi)$ is further $L_{\boldsymbol{x}}$-Lipschitz continuous in $\xi$ for every $\boldsymbol{x}$, then $f(\boldsymbol{x}) \leq L_{\boldsymbol{x}}$ (Mohajerin Esfahani & Kuhn, 2018, Prop. 6.5). If $\Xi = \mathbb{R}^m$, the equality in (34) holds. Note that this equality connection is valid only if $h$ is convex in $\xi \in \Xi$ and $\Xi = \mathbb{R}^m$, which is restrictive.

Extended discussions can be found in, e.g., Shafieezadeh-Abadeh et al. (2019); Blanchet et al. (2019); Gao (2022); Gao et al. (2022).

**Remark 11** *There also exist similar results between robustness and regularization when the distributional ball is defined using $\phi$-divergence; see, e.g., Duchi et al. (2021).* $\qquad\square$

## D Additional Discussions on The Concept System of Distributional Robustness

### D.1 Extra Concepts of Distributional Robustness

In the main body of the paper, we defined the "distributional robustness" of a given solution; cf. Definition 1. This appendix defines the supplementary concepts of "distributional robustness" from other perspectives. One may skip over this appendix if not interested.

**Philosophy 2 (Robust Optimization in Terms of Solution)** *An optimization model is robust if small perturbations in the model do not lead to large changes in the solution.[12] In other words, the solution is insensitive to (small) model perturbations.* $\qquad\square$

Considering the true model (1) and its induced optimal value functional (10), Philosophy 2 can be mathematically specified in Definition 8.

**Definition 8 (Distributional Robustness in Solution)** *Suppose $T(\mathbb{P})$ has a unique solution for every $\mathbb{P} \in B_\epsilon(\mathbb{P}_0)$. The optimal value functional $T$, or equivalently the model set $\{\mathbb{P} \mapsto T(\mathbb{P}) | \mathbb{P} \in B_\epsilon(\mathbb{P}_0)\}$, is $(\epsilon, L)$-distributionally robust at $\mathbb{P}_0$, in terms of solution, if for every $\mathbb{P} \in B_\epsilon(\mathbb{P}_0)$, we have*

$$
\left\| \operatorname*{argmin}_{\boldsymbol{x} \in \mathcal{X}} \mathbb{E}_{\mathbb{P}} h(\boldsymbol{x}, \xi) - \operatorname*{argmin}_{\boldsymbol{x} \in \mathcal{X}} \mathbb{E}_{\mathbb{P}_0} h(\boldsymbol{x}, \xi) \right\| \leq L,
$$

*where $\|\cdot\|$ denotes any proper norm on $\mathcal{X}$ and the value of $L$, which is the smallest value satisfying the above display, depends on the value of $\epsilon$. Here, given $\epsilon$, $L$ is an **absolute robustness measure** of the optimal value functional $T$ in terms of solution at $\mathbb{P}_0$: Given $\epsilon$, the smaller the value of $L$, the more robust the optimal value functional $T$ at $\mathbb{P}_0$ in terms of solution.* $\qquad\square$

---

[12]If the model is parameterized by some parameters, then the perturbations of the model are reflected in the perturbations of the parameters of this model.

As we can see, mathematically, distributional robustness is a property of a model set (or equivalently a property of a model functional), rather than a property of a single model. However, in practice, a single model is said to be distributionally robust if its induced model functional is distributionally robust. This is the subtle difference between the intuitive (resp. philosophical) and formal (resp. mathematical) concepts of distributional robustness because model perturbations of a single model essentially induce a set of models.

**Philosophy 3 (Robust Optimization in Terms of Objective)** *An optimization model is robust if small perturbations in the model do not lead to large changes in the objective value. In other words, the objective value is insensitive to (small) model perturbations.* □

Considering the true model (1) and its induced optimal value functional (10), Philosophy 3 can be mathematically specified in Definition 9.

**Definition 9 (Distributional Robustness in Objective)** *The optimal value functional $T$, or equivalently the model set $\{\mathbb{P} \mapsto T(\mathbb{P})|\mathbb{P} \in B_\epsilon(\mathbb{P}_0)\}$, is $(\epsilon, L)$-distributionally robust at $\mathbb{P}_0$, in terms of objective value, if for every $\mathbb{P} \in B_\epsilon(\mathbb{P}_0)$, we have*

$$\left| \min_{\boldsymbol{x} \in \mathcal{X}} \mathbb{E}_{\mathbb{P}} h(\boldsymbol{x}, \xi) - \min_{\boldsymbol{x} \in \mathcal{X}} \mathbb{E}_{\mathbb{P}_0} h(\boldsymbol{x}, \xi) \right| \leq L,$$

*where the value of $L$, which is the smallest value satisfying the above display, depends on the value of $\epsilon$. Here, given $\epsilon$, $L$ is an absolute robustness measure of the optimal value functional $T$ in terms of objective value at $\mathbb{P}_0$: Given $\epsilon$, the smaller the value of $L$, the more robust the optimal value functional $T$ at $\mathbb{P}_0$ in terms of objective value.* □

Definition 8 and Definition 9 depict the distributional robustness of the optimal value functional $T$ at $\mathbb{P}_0$, or equivalently, the robustness of the model set $\{\mathbb{P} \mapsto T(\mathbb{P})|\mathbb{P} \in B_\epsilon(\mathbb{P}_0)\}$ from two different perspectives.

Two intuitive examples are given below. Example 8 exhibits the large robustness in terms of solution, whereas Example 9 demonstrates the large robustness in terms of objective value.

**Example 8** *Suppose $x \in \mathcal{X} := \mathbb{R}$, $\xi \in \Xi := \{-\delta, \delta\}$, and $h : \mathcal{X} \times \Xi \to \mathbb{R}$ where $\delta$ is an arbitrarily small positive real number. Consider an one-dimensional optimization problem $\min_x h(x, \xi)$ where*

$$h(x, \xi) := \begin{cases} x^2 & \textit{if } \xi = -\delta, \\ x^2 + 1 & \textit{if } \xi = \delta. \end{cases}$$

*In this example, we assume that $\xi$ follows a degenerate distribution at $-\delta$ in the true case and at $\delta$ in the perturbed case. The optimal solution is $x = 0$ for both two cases, but the optimal objective values are different. Hence, in this example, the robustness measure of the associated model set in terms of solution is $0$, while that in terms of objective value is $1$.* □

**Example 9** *Suppose $x \in \mathcal{X} := \mathbb{R}$, $\xi \in \Xi := \{-\delta, \delta\}$, and $h : \mathcal{X} \times \Xi \to \mathbb{R}$ where $\delta$ is an arbitrarily small positive real number. Consider an one-dimensional optimization problem $\min_x h(x, \xi)$ where*

$$h(x, \xi) := \begin{cases} x^2, & \textit{if } \xi = -\delta \\ (x+1)^2, & \textit{if } \xi = \delta. \end{cases}$$

*In this example, we assume that $\xi$ follows a degenerate distribution at $-\delta$ in the true case and at $\delta$ in the perturbed case. The optimal objective value is zero for both two cases, but the optimal solutions are different. Hence, in this example, the robustness measure of the associated model set in terms of solution is $1$, while that in terms of objective value is $0$.* □

In applied statistics, e.g., M-estimation, we focus on the robustness in terms of solution. This is because, for example, in the robust estimation of the location parameter of a distribution, we need to guarantee that the robust location estimate is sufficiently close to the true location (N.B. the location of a distribution is

usually its mean). However, in many real-life applications in, e.g., operations research, management science, machine learning, and signal processing, we might be concerned with the robustness in terms of objective value because we only need to guarantee that the objective value does not significantly change when the model perturbations exist.

The concept of **robust model functional**, or robust model set, in Definitions 8 and 9 are more ideological than practical because, in real-world problems, we are interested in finding a "robust solution" that is workable for every model in a nominal model set, which gives birth to the concept of **robust solution** in Definition 1, in contrast to the concept of robust model set in Definitions 8 and 9.

## D.2   Distributionally Robust Optimization On the Whole Space $\mathcal{M}(\Xi)$

In this appendix, we discuss the general case where the nominal distributions are not limited to the subspace $\{\mathbb{P}|\Delta(\mathbb{P}, \mathbb{P}_0) \leq \epsilon\}$ for a specified $\epsilon$.

### D.2.1   Formalization of Distributionally Robust Optimization

**Definition 10 (Distributionally Robust Optimization)** *The distributionally robust counterpart for the model* (1) *is defined as*

$$
\begin{aligned}
\min_{\boldsymbol{x}, L} \quad & L \\
s.t. \quad & |\mathbb{E}_{\mathbb{P}} h(\boldsymbol{x}, \xi) - \mathbb{E}_{\mathbb{P}_0} h(\boldsymbol{x}_0, \xi)| \leq L \cdot \Delta(\mathbb{P}, \mathbb{P}_0), \quad \forall \mathbb{P} \in \mathcal{M}(\Xi), \\
& \boldsymbol{x} \in \mathcal{X},
\end{aligned}
\tag{35}
$$

*where $\boldsymbol{x}_0 := \operatorname{argmin}_{\boldsymbol{x}} \mathbb{E}_{\mathbb{P}_0} h(\boldsymbol{x}, \xi)$. If* (35) *has a finite solution $(\boldsymbol{x}^*, L^*)$, then $\boldsymbol{x}^*$ is distributionally robust with the **relative robustness measure** $L^*$.* $\qquad \square$

Compared to Definition 10, Definition 2 is more specific because in practice, sometimes we are only required to consider the smaller distributional space $\{\mathbb{P} \in \mathcal{M}(\Xi)|\Delta(\mathbb{P}, \mathbb{P}_0) \leq \epsilon\}$ rather than the whole space $\mathcal{M}(\Xi)$. This is because additional distributional information $\Delta(\mathbb{P}, \mathbb{P}_0) \leq \epsilon$ is helpful in reducing the conservativeness of a distributionally robust solution.

Interested readers are invited to see more motivations and discussions on Definition 10 in Appendix D.3; one can ignore it without missing the main points of this paper.

**Definition 11 (One-Sided Distributionally Robust Optimization)** *The one-sided distributionally robust counterpart for the model* (1) *is defined as*

$$
\begin{aligned}
\min_{\boldsymbol{x}, L} \quad & L \\
s.t. \quad & \mathbb{E}_{\mathbb{P}} h(\boldsymbol{x}, \xi) - \mathbb{E}_{\mathbb{P}_0} h(\boldsymbol{x}_0, \xi) \leq L \cdot \Delta(\mathbb{P}, \mathbb{P}_0), \quad \forall \mathbb{P} \in \mathcal{M}(\Xi), \\
& L \geq 0, \\
& \boldsymbol{x} \in \mathcal{X},
\end{aligned}
\tag{36}
$$

*where $\boldsymbol{x}_0 := \operatorname{argmin}_{\boldsymbol{x}} \mathbb{E}_{\mathbb{P}_0} h(\boldsymbol{x}, \xi)$. If* (36) *has a finite solution $(\boldsymbol{x}^*, L^*)$, then $\boldsymbol{x}^*$ is distributionally robust with the one-sided relative robustness measure $L^*$.* $\qquad \square$

Other than the concepts of "relative robustness measure" and "absolute robustness measure", we can also define the concept of "local robustness measure", which is placed in Appendix D.4. Interested readers are invited to see it for additional motivation and deeper understanding. However, one can ignore it without missing the main points of this paper.

### D.2.2   Practical Implementations of Distributionally Robust Optimization

We discuss the practical case where $\mathbb{P}_0$ is unknown but an estimate $\bar{\mathbb{P}}$ is available.

**Definition 12 (Surrogate Distributionally Robust Optimization)** *The distributionally robust counterpart for the model* (1) *at the surrogate $\bar{\bar{\mathbb{P}}}$ is defined as*

$$
\begin{aligned}
\min_{\boldsymbol{x}, L} \quad & L \\
s.t. \quad & |\mathbb{E}_{\mathbb{P}} h(\boldsymbol{x}, \xi) - \mathbb{E}_{\bar{\bar{\mathbb{P}}}} h(\bar{\boldsymbol{x}}, \xi)| \le L \cdot \Delta(\mathbb{P}, \bar{\bar{\mathbb{P}}}), \quad \forall \mathbb{P} \in \mathcal{M}(\Xi), \\
& \boldsymbol{x} \in \mathcal{X},
\end{aligned}
\tag{37}
$$

*where $\bar{\boldsymbol{x}} := \operatorname{argmin}_{\boldsymbol{x}} \mathbb{E}_{\bar{\bar{\mathbb{P}}}} h(\boldsymbol{x}, \xi)$. For a data-driven problem, $\bar{\bar{\mathbb{P}}}$ can be chosen as $\hat{\bar{\mathbb{P}}}_n$.* □

Problem (37) is therefore termed the ***distributionally robust counterpart*** to the nominal Problem (2).

**Definition 13 (One-Sided Surrogate Distributionally Robust Optimization)** *The one-sided distributionally robust counterpart for the model* (1) *at the surrogate $\bar{\bar{\mathbb{P}}}$ is defined as*

$$
\begin{aligned}
\min_{\boldsymbol{x}, L} \quad & L \\
s.t. \quad & \mathbb{E}_{\mathbb{P}} h(\boldsymbol{x}, \xi) - \mathbb{E}_{\bar{\bar{\mathbb{P}}}} h(\bar{\boldsymbol{x}}, \xi) \le L \cdot \Delta(\mathbb{P}, \bar{\bar{\mathbb{P}}}), \quad \forall \mathbb{P} \in \mathcal{M}(\Xi), \\
& L \ge 0, \\
& \boldsymbol{x} \in \mathcal{X},
\end{aligned}
\tag{38}
$$

*where $\bar{\boldsymbol{x}} := \operatorname{argmin}_{\boldsymbol{x}} \mathbb{E}_{\bar{\bar{\mathbb{P}}}} h(\boldsymbol{x}, \xi)$.* □

Problem (38) is termed the ***one-sided distributionally robust counterpart*** to the nominal Problem (2).

In operations research, a one-sided surrogate distributionally robust optimization model in Definition 13 is termed a *Robust Satisficing* model (Long et al., 2022). Although the robust satisficing model in Long et al. (2022) is built from other motivations rather than Definition 11 and Definition 13, it works as a specified instance of the proposed distributionally robust optimization framework in Definition 11.

The proposition below provides the rationale for the surrogate distributionally robust optimizations.

**Proposition 5 (Surrogate Distributionally Robust Optimization)** *Let $(\boldsymbol{x}_1, L_1)$ solve the surrogate distributionally robust counterpart* (37) *at $\bar{\bar{\mathbb{P}}}$ for the model* (1). *Then $\boldsymbol{x}_1$ is distributionally robust (in the sense of Definition 10) with robustness measure $\max\{L_1, L_2\}$, if $\Delta$ is a statistical distance on $\mathcal{M}(\Xi)$ and there exists a non-negative scalar $L_2 < \infty$ such that*

$$
|\mathbb{E}_{\bar{\bar{\mathbb{P}}}} h(\bar{\boldsymbol{x}}, \xi) - \mathbb{E}_{\mathbb{P}_0} h(\boldsymbol{x}_0, \xi)| \le L_2 \cdot \Delta(\bar{\bar{\mathbb{P}}}, \mathbb{P}_0).
$$

*Likewise, suppose $(\boldsymbol{x}_1, L_1)$ solves the one-sided surrogate distributionally robust counterpart* (38) *at $\bar{\bar{\mathbb{P}}}$ for the model* (1). *Then $\boldsymbol{x}_1$ is distributionally robust with one-sided robustness measure $\max\{L_1, L_2\}$.*

**Proof.** See Appendix G for the proof based on telescoping and triangle inequalities. □

The condition in Proposition 5 depicts that $\bar{\bar{\mathbb{P}}}$ is a good estimate of $\mathbb{P}_0$: the smaller the value of $L_2$, the better. This condition is not restrictive when $\bar{\bar{\mathbb{P}}}$ is specified by $\hat{\bar{\mathbb{P}}}_n$ in a data-driven setup, due to concentration properties of empirical measures; see, e.g., Weed & Bach (2019); Billingsley (1999); Wainwright (2019); Fournier & Guillin (2015).

### D.2.3 Generalization Error Through Relative Robustness Measure

In analogy to Proposition 4, generalization errors can also be characterized through the relative robustness measure.

**Proposition 6 (Generalization Error Through Relative Robustness Measure)** *Suppose $(\boldsymbol{x}^*, L^*)$ solves the surrogate distributionally robust optimization model* (37), *i.e.,*

$$
\begin{aligned}
\min_{\boldsymbol{x}, L} \quad & L \\
s.t. \quad & |\mathbb{E}_{\mathbb{P}} h(\boldsymbol{x}, \xi) - \mathbb{E}_{\bar{\bar{\mathbb{P}}}} h(\bar{\boldsymbol{x}}, \xi)| \le L \Delta(\mathbb{P}, \bar{\bar{\mathbb{P}}}), \quad \forall \mathbb{P} \in \mathcal{M}(\Xi), \\
& \boldsymbol{x} \in \mathcal{X}.
\end{aligned}
$$

If $h(\boldsymbol{x}, \xi)$ is $L(\xi)$-Lipschitz continuous in $\boldsymbol{x}$ on $\mathcal{X}$, for every $\xi \in \Xi$, then the generalization error gap $|\mathbb{E}_{\mathbb{P}_0} h(\bar{\boldsymbol{x}}, \xi) - \mathbb{E}_{\bar{\mathbb{P}}} h(\bar{\boldsymbol{x}}, \xi)|$ of the nominal model $\min_{\boldsymbol{x}} \mathbb{E}_{\bar{\mathbb{P}}} h(\boldsymbol{x}, \xi)$ is upper bounded by

$$\|\bar{\boldsymbol{x}} - \boldsymbol{x}^*\| \cdot \mathbb{E}_{\mathbb{P}_0} L(\xi) + L^* \Delta(\mathbb{P}_0, \bar{\mathbb{P}}),$$

$\mathbb{P}_0^n$-almost surely. In addition, the generalization error gap $|\mathbb{E}_{\mathbb{P}_0} h(\boldsymbol{x}^*, \xi) - \mathbb{E}_{\mathbb{P}^*} h(\boldsymbol{x}^*, \xi)|$ of the surrogate distributionally robust optimization model (37) is upper bounded by

$$L^* \cdot [\Delta(\mathbb{P}_0, \bar{\mathbb{P}}) + \Delta(\mathbb{P}^*, \bar{\mathbb{P}})],$$

$\mathbb{P}_0^n$-almost surely, where

$$\mathbb{P}^* \in \underset{\mathbb{P} \in \mathcal{M}(\Xi)}{\operatorname{argmax}} \frac{|\mathbb{E}_{\mathbb{P}} h(\boldsymbol{x}^*, \xi) - \mathbb{E}_{\bar{\mathbb{P}}} h(\bar{\boldsymbol{x}}, \xi)|}{\Delta(\mathbb{P}, \bar{\mathbb{P}})}.$$

Note that $|\mathbb{E}_{\mathbb{P}^*} h(\boldsymbol{x}^*, \xi) - \mathbb{E}_{\bar{\mathbb{P}}} h(\bar{\boldsymbol{x}}, \xi)| = L^* \Delta(\mathbb{P}^*, \bar{\mathbb{P}})$.

**Proof.** See Appendix O for the proof based on telescoping and triangle inequalities. $\qquad\square$

Again, note that Proposition 6 holds almost surely, not in the PAC sense, and therefore, generalization error bounds specified by relative robustness measures are stronger than those specified in the PAC sense.

**Corollary 4 (Generalization Error Through Relative Robustness Measure)** *Under the settings of Proposition 6, the expected generalization errors are given by*

$$\mathbb{E}_{\mathbb{P}_0^{n+1}} L(\xi) \|\bar{\boldsymbol{x}} - \boldsymbol{x}^*\| + \mathbb{E}_{\mathbb{P}_0^n} L^* \Delta(\mathbb{P}_0, \bar{\mathbb{P}}),$$

*and $\mathbb{E}_{\mathbb{P}_0^n} \{L^* \cdot [\Delta(\mathbb{P}_0, \bar{\mathbb{P}}) + \Delta(\mathbb{P}^*, \bar{\mathbb{P}})]\}$, respectively. Note that $\bar{\mathbb{P}}$, $\mathbb{P}^*$, $\bar{\boldsymbol{x}}$, $\boldsymbol{x}^*$, and $L^*$ depend on the specific choice of the training data.*

**Proof.** See Appendix P. $\qquad\square$

Proposition 6 and Corollary 4 mean that whenever the DRO model (37) is solved, the generalization errors of the nominal model $\min_{\boldsymbol{x}} \mathbb{E}_{\bar{\mathbb{P}}} h(\boldsymbol{x}, \xi)$ and the surrogate distributionally robust optimization model (37) are also tightly controlled. The one-sided versions of Proposition 6 and Corollary 4 are straightforward to be claimed and therefore omitted in this paper.

### D.3 Fractional Distributionally Robust Optimization

To achieve distributional robustness (i.e., the objective value is as still as possible at a given solution), we are first motivated to give the definition of the fractional distributionally robust counterpart for a model set centered at the true model (1), or simply, the fractional distributionally robust counterpart for the true model (1).

**Definition 14 (Fractional Distributionally Robust Optimization)** *The fractional distributionally robust counterpart for the model (1) is defined as*

$$\min_{\boldsymbol{x} \in \mathcal{X}} \max_{\mathbb{P} \in \mathcal{M}(\Xi)} \frac{|\mathbb{E}_{\mathbb{P}} h(\boldsymbol{x}, \xi) - \mathbb{E}_{\mathbb{P}_0} h(\boldsymbol{x}_0, \xi)|}{\Delta(\mathbb{P}, \mathbb{P}_0)}, \tag{39}$$

*where $\boldsymbol{x}_0 := \operatorname{argmin}_{\boldsymbol{x}} \mathbb{E}_{\mathbb{P}_0} h(\boldsymbol{x}, \xi)$. If (39) has a finite solution $(\boldsymbol{x}^*, \mathbb{P}^*)$, then $\boldsymbol{x}^*$ is a distributionally robust solution with the* **relative robustness measure**

$$L^* := \frac{|\mathbb{E}_{\mathbb{P}^*} h(\boldsymbol{x}^*, \xi) - \mathbb{E}_{\mathbb{P}_0} h(\boldsymbol{x}_0, \xi)|}{\Delta(\mathbb{P}^*, \mathbb{P}_0)}.$$

The proposition below states the relationship between the fractional distributionally robust counterpart (39) and the distributionally robust counterpart (35).

**Proposition 7** *The two distributionally robust optimization models* (39) *and* (35) *are equivalent in the sense that if* $(\boldsymbol{x}^*, \mathbb{P}^*)$ *solves* (39) *then* $(\boldsymbol{x}^*, L^*)$ *solves* (35) *where*

$$L^* = \frac{|\mathbb{E}_{\mathbb{P}^*} h(\boldsymbol{x}^*, \xi) - \mathbb{E}_{\mathbb{P}_0} h(\boldsymbol{x}_0, \xi)|}{\Delta(\mathbb{P}^*, \mathbb{P}_0)};$$

*If* $(\boldsymbol{x}^*, L^*)$ *solves* (35) *then* $(\boldsymbol{x}^*, \mathbb{P}^*)$ *solves* (39) *where* $\mathbb{P}^*$ *satisfies*

$$|\mathbb{E}_{\mathbb{P}^*} h(\boldsymbol{x}^*, \xi) - \mathbb{E}_{\mathbb{P}_0} h(\boldsymbol{x}_0, \xi)| = L^* \cdot \Delta(\mathbb{P}^*, \mathbb{P}_0).$$

**Proof.** See Appendix F. □

Due to the equivalence between (39) and (35), one may use any one of them in practice whichever is easier to solve for specific problems.

The one-sided version is defined below.

**Definition 15 (One-Sided Fractional Distributionally Robust Optimization)** *The one-sided fractional distributionally robust counterpart for the model* (1) *is defined as*

$$\min_{\boldsymbol{x} \in \mathcal{X}} \max_{\mathbb{P} \in \mathcal{M}(\Xi)} \frac{\mathbb{E}_{\mathbb{P}} h(\boldsymbol{x}, \xi) - \mathbb{E}_{\mathbb{P}_0} h(\boldsymbol{x}_0, \xi)}{\Delta(\mathbb{P}, \mathbb{P}_0)}, \tag{40}$$

*where* $\boldsymbol{x}_0 := \operatorname{argmin}_{\boldsymbol{x}} \mathbb{E}_{\mathbb{P}_0} h(\boldsymbol{x}, \xi)$. *If* (40) *has a finite solution* $(\boldsymbol{x}^*, \mathbb{P}^*)$, *then* $\boldsymbol{x}^*$ *is a distributionally robust solution with the one-sided relative robustness measure*

$$L^* := \max \left\{ 0, \ \frac{\mathbb{E}_{\mathbb{P}^*} h(\boldsymbol{x}^*, \xi) - \mathbb{E}_{\mathbb{P}_0} h(\boldsymbol{x}_0, \xi)}{\Delta(\mathbb{P}^*, \mathbb{P}_0)} \right\}.$$

### D.4 Local Robustness Measure

The robustness measures given by (39), (35), and (12) are global properties of $\boldsymbol{x}^*$ for the functional $M : (\boldsymbol{x}^*, \mathbb{P}) \mapsto \mathbb{E}_{\mathbb{P}} h(\boldsymbol{x}^*, \xi)$ at $\mathbb{P}_0$. We can also define a local robustness measure for $\boldsymbol{x}^*$ for the functional $M : (\boldsymbol{x}^*, \mathbb{P}) \mapsto \mathbb{E}_{\mathbb{P}} h(\boldsymbol{x}^*, \xi)$ at $\mathbb{P}_0$. One can skip over this appendix without missing the main points of this paper if not interested.

**Definition 16** *The **local robustness measure** $L_0$ of a distributionally robust solution $\boldsymbol{x}^*$, in terms of objective value, for the functional* $M : (\boldsymbol{x}^*, \mathbb{P}) \mapsto \mathbb{E}_{\mathbb{P}} h(\boldsymbol{x}^*, \xi)$ *at* $\mathbb{P}_0$ *is defined as*

$$L_0 \quad := \lim_{\epsilon \downarrow 0} \frac{\min\limits_{\boldsymbol{x} \in \mathcal{X}} \max\limits_{\mathbb{P} \in B_\epsilon(\mathbb{P}_0)} \left| \mathbb{E}_{\mathbb{P}} h(\boldsymbol{x}, \xi) - \mathbb{E}_{\mathbb{P}_0} h(\boldsymbol{x}_0, \xi) \right|}{\epsilon}, \tag{41}$$

*where* $\boldsymbol{x}_0 := \operatorname{argmin}_{\boldsymbol{x}} \mathbb{E}_{\mathbb{P}_0} h(\boldsymbol{x}, \xi)$ *and* $\boldsymbol{x}^*$ *solves* (41). □

The robustness measure in Definition 16 includes Gotoh et al. (2021, cf. Eq. (8)), Gotoh et al. (2018) as a special case.

**Definition 17** *The local robustness measure $L_0$ of a distributionally robust solution $\boldsymbol{x}^*$, in terms of solution, for the functional* $M : (\boldsymbol{x}^*, \mathbb{P}) \mapsto \mathbb{E}_{\mathbb{P}} h(\boldsymbol{x}^*, \xi)$ *at* $\mathbb{P}_0$ *is defined as*

$$L_0 \quad := \lim_{\epsilon \downarrow 0} \frac{\left\| \operatorname{argmin}\limits_{\boldsymbol{x} \in \mathcal{X}} \max\limits_{\mathbb{P} \in B_\epsilon(\mathbb{P}_0)} \mathbb{E}_{\mathbb{P}} h(\boldsymbol{x}, \xi) - \boldsymbol{x}_0 \right\|}{\epsilon}, \tag{42}$$

*where* $\boldsymbol{x}_0 := \operatorname{argmin}_{\boldsymbol{x}} \mathbb{E}_{\mathbb{P}_0} h(\boldsymbol{x}, \xi)$ *and* $\boldsymbol{x}^*$ *solves* (42). □

The robustness measure in Definition 17 is highly related to the concept of *Influence Function* in applied statistics (Hampel, 1974, cf. Sec. 2.2) for outlier-robust M-estimators. To be specific, the influence function of an estimator is a special case of (42) when the statistical distance $\Delta$ used to define $B_\epsilon(\mathbb{P}_0)$ is specified by the Huber's $\epsilon$-contamination set, which is used for outlier-robust M-statistics. Note again that in applied statistics, one is concerned more with the solution itself rather than the objective value.

## D.5   Min-Max Distributionally Robust Optimization

The min-max distributionally robust optimization models are only able to provide one-sided distributional robustness. Note that the two-sided version (12) is equivalent to

$$
\begin{aligned}
\min_{\boldsymbol{x}, L} \quad & L \\
\text{s.t.} \quad & \mathbb{E}_{\mathbb{P}_0} h(\boldsymbol{x}_0, \xi) - \mathbb{E}_{\mathbb{P}} h(\boldsymbol{x}, \xi) \leq L, \quad \forall \mathbb{P} : \Delta(\mathbb{P}, \mathbb{P}_0) \leq \epsilon, \\
& \mathbb{E}_{\mathbb{P}} h(\boldsymbol{x}, \xi) - \mathbb{E}_{\mathbb{P}_0} h(\boldsymbol{x}_0, \xi) \leq L, \quad \forall \mathbb{P} : \Delta(\mathbb{P}, \mathbb{P}_0) \leq \epsilon, \\
& \boldsymbol{x} \in \mathcal{X},
\end{aligned}
$$

which is further equivalent to

$$
\begin{aligned}
\min_{\boldsymbol{x}} \quad & \max_{\mathbb{P}} \left\{ \max_{\mathbb{P}} \left[ \mathbb{E}_{\mathbb{P}_0} h(\boldsymbol{x}_0, \xi) - \mathbb{E}_{\mathbb{P}} h(\boldsymbol{x}, \xi) \right], \quad \max_{\mathbb{P}} \left[ \mathbb{E}_{\mathbb{P}} h(\boldsymbol{x}, \xi) - \mathbb{E}_{\mathbb{P}_0} h(\boldsymbol{x}_0, \xi) \right] \right\} \\
\text{s.t.} \quad & \Delta(\mathbb{P}, \mathbb{P}_0) \leq \epsilon, \\
& \boldsymbol{x} \in \mathcal{X}.
\end{aligned}
$$

Hence, the min-max distributionally robust optimization model (18) is a one-sided relaxed version of the above display; the other-side relaxation is

$$
\begin{aligned}
\min_{\boldsymbol{x}} \max_{\mathbb{P}} \quad & \mathbb{E}_{\mathbb{P}_0} h(\boldsymbol{x}_0, \xi) - \mathbb{E}_{\mathbb{P}} h(\boldsymbol{x}, \xi) \\
\text{s.t.} \quad & \Delta(\mathbb{P}, \mathbb{P}_0) \leq \epsilon, \\
& \boldsymbol{x} \in \mathcal{X},
\end{aligned}
$$

that is

$$
\begin{aligned}
\max_{\boldsymbol{x}} \min_{\mathbb{P}} \quad & \mathbb{E}_{\mathbb{P}} h(\boldsymbol{x}, \xi) \\
\text{s.t.} \quad & \Delta(\mathbb{P}, \mathbb{P}_0) \leq \epsilon, \\
& \boldsymbol{x} \in \mathcal{X}.
\end{aligned}
$$

## D.6   Unbounded Support Set

When $\Xi$ is unbounded, it is plausible to set

$$
\frac{\mathrm{d}\hat{\mathbb{P}}}{\mathrm{d}\mathcal{L}} := a(\boldsymbol{x}) \exp^{-b(\boldsymbol{x}) \|\xi - \xi_0\|^r}, \quad \forall \boldsymbol{x},
$$

among many other possibilities, such that

$$
1 = \int_{\Xi} a(\boldsymbol{x}) \exp^{-b(\boldsymbol{x}) \|\xi - \xi_0\|^r} \mathrm{d}\xi, \quad \forall \boldsymbol{x},
$$

and

$$
f(\boldsymbol{x}) = \int_{\Xi} h(\boldsymbol{x}, \xi) a(\boldsymbol{x}) \exp^{-b(\boldsymbol{x}) \|\xi - \xi_0\|^r} \mathrm{d}\xi, \quad \forall \boldsymbol{x},
$$

where $a, b : \mathcal{X} \to \mathbb{R}_+$ are two design functions and $r > 0$ is a design parameter (if they exist), and $\xi_0$ is the design location (possibly the mean) of $\hat{\mathbb{P}}$.

## D.7   Summary of Relations Among Concepts

The relations among all concepts of distributional robustness, distributionally robust optimization method, Bayesian method, and regularization method, defined in this paper, can be summarized as follows.

1. The concept of distributional robustness is given in Definition 1.

2. Two-sided distributionally robust optimizations [i.e., (12) and (35)] and one-sided distributionally robust optimizations [i.e., (13) and (36)] can provide distributional robustness.

3. The difference between (12) and (35) is whether there is a condition $\Delta(\mathbb{P}, \mathbb{P}_0) \leq \epsilon$, so is between (13) and (36).

4. The one-sided distributionally robust optimization (13) is equivalent to a min-max distributionally robust optimization (18); see Proposition 1.

5. The Bayesian method (7) is probably approximately distributionally robust; see Proposition 2.

6. Under Dirichlet-process priors, the regularization method (9) is equivalent to a Bayesian method (7); see Proposition 3.

7. By changing the center of the distributional family from $\bar{\bar{\mathbb{P}}}$ to $\mathbb{P}_0$, the two-sided surrogate distributionally robust optimizations (14) and (37) can be transformed to the two-sided distributionally robust optimizations (12) and (35), respectively. The distributional robustness of the two-sided surrogate distributionally robust optimization (14) and (37) can be guaranteed by Fact 1 and Proposition 5, respectively.

8. By changing the center of the distributional family from $\bar{\bar{\mathbb{P}}}$ to $\mathbb{P}_0$, the one-sided surrogate distributionally robust optimizations (15) and (38) can be transformed to the one-sided distributionally robust optimizations (13) and (36), respectively. The distributional robustness of the one-sided surrogate distributionally robust optimization (15) and (38) can be guaranteed by Fact 1 and Proposition 5, respectively.

9. The one-sided surrogate distributionally robust optimizations (15) and (38) are the one-sided relaxations of the two-sided surrogate distributionally robust optimizations (14) and (37), respectively.

10. The difference between (14) and (37) is whether there is a condition $\Delta(\mathbb{P}, \bar{\mathbb{P}}) \leq \epsilon$, so is between (15) and (38).

11. The one-sided surrogate distributionally robust optimization (15) is equivalent to a surrogate min-max distributionally robust optimization (19); see Corollary 1. Model (19) is the formal definition of (and hence the same as) min-max DRO (8).

# E   Proof of Fact 1

**Proof.**  We have

$$\begin{aligned}
|\mathbb{E}_{\mathbb{P}}h(\boldsymbol{x}_1,\xi) - \mathbb{E}_{\mathbb{P}_0}h(\boldsymbol{x}_0,\xi)| \quad &= |\mathbb{E}_{\mathbb{P}}h(\boldsymbol{x}_1,\xi) - \mathbb{E}_{\bar{\mathbb{P}}}h(\bar{\boldsymbol{x}},\xi) + \mathbb{E}_{\bar{\mathbb{P}}}h(\bar{\boldsymbol{x}},\xi) - \mathbb{E}_{\mathbb{P}_0}h(\boldsymbol{x}_0,\xi)| \\
&\leq |\mathbb{E}_{\mathbb{P}}h(\boldsymbol{x}_1,\xi) - \mathbb{E}_{\bar{\mathbb{P}}}h(\bar{\boldsymbol{x}},\xi)| + |\mathbb{E}_{\bar{\mathbb{P}}}h(\bar{\boldsymbol{x}},\xi) - \mathbb{E}_{\mathbb{P}_0}h(\boldsymbol{x}_0,\xi)| \\
&\leq L_1 + L_2.
\end{aligned}$$

Likewise, we have

$$\begin{aligned}
\mathbb{E}_{\mathbb{P}}h(\boldsymbol{x}_1,\xi) - \mathbb{E}_{\mathbb{P}_0}h(\boldsymbol{x}_0,\xi) \quad &= \mathbb{E}_{\mathbb{P}}h(\boldsymbol{x}_1,\xi) - \mathbb{E}_{\bar{\mathbb{P}}}h(\bar{\boldsymbol{x}},\xi) + \mathbb{E}_{\bar{\mathbb{P}}}h(\bar{\boldsymbol{x}},\xi) - \mathbb{E}_{\mathbb{P}_0}h(\boldsymbol{x}_0,\xi) \\
&\leq \mathbb{E}_{\mathbb{P}}h(\boldsymbol{x}_1,\xi) - \mathbb{E}_{\bar{\mathbb{P}}}h(\bar{\boldsymbol{x}},\xi) + |\mathbb{E}_{\bar{\mathbb{P}}}h(\bar{\boldsymbol{x}},\xi) - \mathbb{E}_{\mathbb{P}_0}h(\boldsymbol{x}_0,\xi)| \\
&\leq L_1 + L_2.
\end{aligned}$$

This completes the proof.                                                      $\square$

# F    Proof of Proposition 7

**Proof.** Suppose $(\boldsymbol{x}^*, \mathbb{P}^*)$ solves (39). By Definition 14, we have

$$L^* := \frac{|\mathbb{E}_{\mathbb{P}^*} h(\boldsymbol{x}^*, \xi) - \mathbb{E}_{\mathbb{P}_0} h(\boldsymbol{x}_0, \xi)|}{\Delta(\mathbb{P}^*, \mathbb{P}_0)} \geq \frac{|\mathbb{E}_{\mathbb{P}} h(\boldsymbol{x}^*, \xi) - \mathbb{E}_{\mathbb{P}_0} h(\boldsymbol{x}_0, \xi)|}{\Delta(\mathbb{P}, \mathbb{P}_0)}, \quad \forall \mathbb{P} \in \mathcal{M}(\Xi),$$

that is,

$$|\mathbb{E}_{\mathbb{P}} h(\boldsymbol{x}^*, \xi) - \mathbb{E}_{\mathbb{P}_0} h(\boldsymbol{x}_0, \xi)| \leq L^* \cdot \Delta(\mathbb{P}, \mathbb{P}_0), \quad \forall \mathbb{P} \in \mathcal{M}(\Xi).$$

Suppose $(\boldsymbol{x}^*, L^*)$ solves (35). We have

$$|\mathbb{E}_{\mathbb{P}} h(\boldsymbol{x}^*, \xi) - \mathbb{E}_{\mathbb{P}_0} h(\boldsymbol{x}_0, \xi)| \leq L^* \cdot \Delta(\mathbb{P}, \mathbb{P}_0), \quad \forall \mathbb{P} \in \mathcal{M}(\Xi).$$

As a result, by assuming that $\mathbb{P}^*$ satisfies the equality of the above display, we have

$$L^* = \frac{|\mathbb{E}_{\mathbb{P}^*} h(\boldsymbol{x}^*, \xi) - \mathbb{E}_{\mathbb{P}_0} h(\boldsymbol{x}_0, \xi)|}{\Delta(\mathbb{P}^*, \mathbb{P}_0)} \geq \frac{|\mathbb{E}_{\mathbb{P}} h(\boldsymbol{x}^*, \xi) - \mathbb{E}_{\mathbb{P}_0} h(\boldsymbol{x}_0, \xi)|}{\Delta(\mathbb{P}, \mathbb{P}_0)}, \quad \forall \mathbb{P} \in \mathcal{M}(\Xi).$$

This completes the proof. Note that two optimization problems are equivalent if an optimal solution of one optimization problem is a feasible solution of the other optimization problem. Note also that $\boldsymbol{x}^*$ and $\mathbb{P}^*$ may not be unique; however, this does not change the statement. $\qquad\square$

# G    Proof of Proposition 5

**Proof.** We have, $\forall \mathbb{P} \in \mathcal{M}(\Xi)$,

$$
\begin{aligned}
|\mathbb{E}_{\mathbb{P}} h(\boldsymbol{x}_1, \xi) - \mathbb{E}_{\mathbb{P}_0} h(\boldsymbol{x}_0, \xi)| \ &= |\mathbb{E}_{\mathbb{P}} h(\boldsymbol{x}_1, \xi) - \mathbb{E}_{\bar{\mathbb{P}}} h(\bar{\boldsymbol{x}}, \xi) + \mathbb{E}_{\bar{\mathbb{P}}} h(\bar{\boldsymbol{x}}, \xi) - \mathbb{E}_{\mathbb{P}_0} h(\boldsymbol{x}_0, \xi)| \\
&\leq |\mathbb{E}_{\mathbb{P}} h(\boldsymbol{x}_1, \xi) - \mathbb{E}_{\bar{\mathbb{P}}} h(\bar{\boldsymbol{x}}, \xi)| + |\mathbb{E}_{\bar{\mathbb{P}}} h(\bar{\boldsymbol{x}}, \xi) - \mathbb{E}_{\mathbb{P}_0} h(\boldsymbol{x}_0, \xi)| \\
&\leq L_1 \cdot \Delta(\mathbb{P}, \bar{\mathbb{P}}) + L_2 \cdot \Delta(\bar{\mathbb{P}}, \mathbb{P}_0) \\
&\leq L \cdot \Delta(\mathbb{P}, \mathbb{P}_0),
\end{aligned}
$$

where $L := \max\{L_1, L_2\}$.

Likewise, we have, $\forall \mathbb{P} \in \mathcal{M}(\Xi)$,

$$
\begin{aligned}
\mathbb{E}_{\mathbb{P}} h(\boldsymbol{x}_1, \xi) - \mathbb{E}_{\mathbb{P}_0} h(\boldsymbol{x}_0, \xi) \ &= \mathbb{E}_{\mathbb{P}} h(\boldsymbol{x}_1, \xi) - \mathbb{E}_{\bar{\mathbb{P}}} h(\bar{\boldsymbol{x}}, \xi) + \mathbb{E}_{\bar{\mathbb{P}}} h(\bar{\boldsymbol{x}}, \xi) - \mathbb{E}_{\mathbb{P}_0} h(\boldsymbol{x}_0, \xi) \\
&\leq \mathbb{E}_{\mathbb{P}} h(\boldsymbol{x}_1, \xi) - \mathbb{E}_{\bar{\mathbb{P}}} h(\bar{\boldsymbol{x}}, \xi) + |\mathbb{E}_{\bar{\mathbb{P}}} h(\bar{\boldsymbol{x}}, \xi) - \mathbb{E}_{\mathbb{P}_0} h(\boldsymbol{x}_0, \xi)| \\
&\leq L_1 \cdot \Delta(\mathbb{P}, \bar{\mathbb{P}}) + L_2 \cdot \Delta(\bar{\mathbb{P}}, \mathbb{P}_0) \\
&\leq L \cdot \Delta(\mathbb{P}, \mathbb{P}_0).
\end{aligned}
$$

This completes the proof. $\qquad\square$

# H    Proof of Proposition 1

**Proof.** Problem (13) is equivalent to

$$
\begin{aligned}
\min_{\boldsymbol{x}, L} \quad & L \\
\text{s.t.} \quad & \max_{\mathbb{P}: \ \Delta(\mathbb{P}, \mathbb{P}_0) \leq \epsilon} \mathbb{E}_{\mathbb{P}} h(\boldsymbol{x}, \xi) - \mathbb{E}_{\mathbb{P}_0} h(\boldsymbol{x}_0, \xi) \leq L, \\
& L \geq 0, \\
& \boldsymbol{x} \in \mathcal{X}.
\end{aligned}
\tag{43}
$$

By eliminating $L$, the above display is equivalent to

$$
\begin{aligned}
\min_{\boldsymbol{x}} \max_{\mathbb{P}} \quad & \mathbb{E}_{\mathbb{P}} h(\boldsymbol{x}, \xi) - \mathbb{E}_{\mathbb{P}_0} h(\boldsymbol{x}_0, \xi) \\
\text{s.t.} \quad & \Delta(\mathbb{P}, \mathbb{P}_0) \leq \epsilon, \\
& \boldsymbol{x} \in \mathcal{X},
\end{aligned}
\tag{44}
$$

which is further equivalent to (18) because $\mathbb{E}_{\mathbb{P}_0} h(\boldsymbol{x}_0, \xi)$ is a constant. Note that

$$
\max_{\mathbb{P}:\ \Delta(\mathbb{P},\mathbb{P}_0)\leq\epsilon} \mathbb{E}_{\mathbb{P}} h(\boldsymbol{x}, \xi) \geq \mathbb{E}_{\mathbb{P}_0} h(\boldsymbol{x}, \xi) \geq \min_{\boldsymbol{x}\in\mathcal{X}} \mathbb{E}_{\mathbb{P}_0} h(\boldsymbol{x}, \xi) = \mathbb{E}_{\mathbb{P}_0} h(\boldsymbol{x}_0, \xi), \quad \forall \boldsymbol{x} \in \mathcal{X}.
\tag{45}
$$

Note also that the above steps are all reversible. This completes the proof. $\qquad\square$

# I  Proof of Theorem 1

**Proof.** We have

$$
\begin{aligned}
\mathbb{E}_{\mathbb{Q}} \mathbb{E}_{\mathbb{P}} h(\boldsymbol{x}, \xi) \ &= \int_{\mathbb{R}} \cdot \int_{\Xi} h(\boldsymbol{x}, \xi) \mathbb{P}(\mathrm{d}\xi) \cdot \mathbb{Q}(\mathrm{d}\mathbb{P}(\mathrm{d}\xi)), \\
&= \int_{\Xi} h(\boldsymbol{x}, \xi) \cdot \int_{\mathbb{R}} \mathbb{P}(\mathrm{d}\xi) \mathbb{Q}(\mathrm{d}\mathbb{P}(\mathrm{d}\xi)), \\
&= \int_{\Xi} h(\boldsymbol{x}, \xi) \mathbb{P}'(\mathrm{d}\xi), \\
&= \mathbb{E}_{\mathbb{P}'} h(\boldsymbol{x}, \xi).
\end{aligned}
$$

The second equality is due to the measure-theoretic version of Fubini's theorem (Cohn, 2013, p. 148) by noting that probability measures are $\sigma$-finite and that $\mathbb{P}(\mathrm{d}\xi)$ is a random variable on $\mathbb{R}$ for every Borel set $\mathrm{d}\xi$. The third equality is due to the definition of a mean distribution by noting that the mean distribution of $\mathbb{Q}$ is essentially a mixture distribution of the elements in the set $\mathcal{M}(\Xi)$ with weights determined by $\mathbb{Q}$; cf. Gaudard & Hadwin (1989), Ghosal & Van der Vaart (2017, p. 27). $\qquad\square$

# J  Proof of Proposition 2

**Proof.** For every $\boldsymbol{x}$, since $\mathbb{P}$ is a random measure, $\mathbb{E}_{\mathbb{P}} h(\boldsymbol{x}, \xi)$ is a random variable and $\mathbb{E}_{\mathbb{Q}} \mathbb{E}_{\mathbb{P}} h(\boldsymbol{x}, \xi)$ is its mean. According to Markov's inequality, we have

$$
\begin{aligned}
\mathbb{Q}[|\mathbb{E}_{\mathbb{P}} h(\boldsymbol{x}, \xi) - \mathbb{E}_{\mathbb{P}_0} h(\boldsymbol{x}_0, \xi)| \leq L] \ &\geq 1 - \frac{\mathbb{E}_{\mathbb{Q}} |\mathbb{E}_{\mathbb{P}} h(\boldsymbol{x}, \xi) - \mathbb{E}_{\mathbb{P}_0} h(\boldsymbol{x}_0, \xi)|}{L}, && \forall \boldsymbol{x}, \\
&\geq 1 - \frac{\mathbb{E}_{\mathbb{Q}} |\mathbb{E}_{\mathbb{P}} h(\boldsymbol{x}, \xi)| + |\mathbb{E}_{\mathbb{P}_0} h(\boldsymbol{x}_0, \xi)|}{L}, && \forall \boldsymbol{x}, \\
&= 1 - \frac{\mathbb{E}_{\mathbb{Q}} \mathbb{E}_{\mathbb{P}} h(\boldsymbol{x}, \xi) + \mathbb{E}_{\mathbb{P}_0} h(\boldsymbol{x}_0, \xi)}{L}, && \forall \boldsymbol{x}.
\end{aligned}
\tag{46}
$$

Eq. (46) implies that whenever $\mathbb{E}_{\mathbb{Q}} \mathbb{E}_{\mathbb{P}} h(\boldsymbol{x}, \xi)$ is minimized by $\boldsymbol{x}^*$ through (7), the solution $\boldsymbol{x}^*$ is distributionally robust with absolute robustness measure $L$ with probability at least

$$
\max\left\{0, \quad 1 - \frac{\mathbb{E}_{\mathbb{Q}} \mathbb{E}_{\mathbb{P}} h(\boldsymbol{x}^*, \xi) + \mathbb{E}_{\mathbb{P}_0} h(\boldsymbol{x}_0, \xi)}{L}\right\}.
$$

This completes the proof. $\qquad\square$

# K  Proof of Proposition 3

**Proof.** If (26) holds, the associated regularized SAA optimization problem (25) is equivalent to a Bayesian method (23). By constructing the posterior mean of $\mathbb{Q}$ to be $\mathbb{E}\mathbb{Q} := \beta_n \hat{\mathbb{P}} + (1 - \beta_n) \hat{\mathbb{P}}_n$, we have that (25)

is equivalent to (7). The rest of the statement follows from Markov's inequality; cf. Proposition 2 and its proof. □

## L   Proof of Fact 2

**Proof.**   We have

$$
\begin{aligned}
|\mathbb{E}_{\mathbb{P}_0} h(\boldsymbol{x}, \xi) - \mathbb{E}_{\bar{\mathbb{P}}} h(\boldsymbol{x}, \xi)| &= \left| \int \int h(\boldsymbol{x}, \xi) - h(\boldsymbol{x}, \bar{\xi}) \mathrm{d}\pi(\xi, \bar{\xi}) \right| \\
&\le \int \int \left| h(\boldsymbol{x}, \xi) - h(\boldsymbol{x}, \bar{\xi}) \right| \mathrm{d}\pi(\xi, \bar{\xi}) \\
&\le L_{\boldsymbol{x}} \int \int \|\xi - \bar{\xi}\| \mathrm{d}\pi(\xi, \bar{\xi}),
\end{aligned}
$$

where $\pi$ is a coupling of $\mathbb{P}_0$ and $\bar{\mathbb{P}}$. Since $\pi$ is arbitrary, it holds that

$$
|\mathbb{E}_{\mathbb{P}_0} h(\boldsymbol{x}, \xi) - \mathbb{E}_{\bar{\mathbb{P}}} h(\boldsymbol{x}, \xi)| \le L_{\boldsymbol{x}} \inf_{\pi \in \mathcal{M}(\Xi \times \Xi)} \int \int \|\xi - \bar{\xi}\| \mathrm{d}\pi(\xi, \bar{\xi}).
$$

The right-hand side is the order-1 Wasserstein distance. This completes the proof. □

## M   Proof of Proposition 4

**Proof.**   Due to the feasibility of $(\boldsymbol{x}^*, L^*, \mathbb{P}_0)$, we have

$$
|\mathbb{E}_{\mathbb{P}_0} h(\boldsymbol{x}^*, \xi) - \mathbb{E}_{\bar{\mathbb{P}}} h(\bar{\boldsymbol{x}}, \xi)| \le L^*.
$$

Hence, if $h(\boldsymbol{x}, \xi)$ is $L(\xi)$-Lipschitz continuous in $\boldsymbol{x}$ on $\mathcal{X}$, the generalization error gap $|\mathbb{E}_{\mathbb{P}_0} h(\bar{\boldsymbol{x}}, \xi) - \mathbb{E}_{\bar{\mathbb{P}}} h(\bar{\boldsymbol{x}}, \xi)|$ of the nominal model $\min_{\boldsymbol{x}} \mathbb{E}_{\bar{\mathbb{P}}} h(\boldsymbol{x}, \xi)$ can be given as

$$
\begin{aligned}
|\mathbb{E}_{\mathbb{P}_0} h(\bar{\boldsymbol{x}}, \xi) - \mathbb{E}_{\bar{\mathbb{P}}} h(\bar{\boldsymbol{x}}, \xi)| &= |\mathbb{E}_{\mathbb{P}_0} h(\bar{\boldsymbol{x}}, \xi) - \mathbb{E}_{\mathbb{P}_0} h(\boldsymbol{x}^*, \xi) + \mathbb{E}_{\mathbb{P}_0} h(\boldsymbol{x}^*, \xi) - \mathbb{E}_{\bar{\mathbb{P}}} h(\bar{\boldsymbol{x}}, \xi)| \\
&\le |\mathbb{E}_{\mathbb{P}_0} h(\bar{\boldsymbol{x}}, \xi) - \mathbb{E}_{\mathbb{P}_0} h(\boldsymbol{x}^*, \xi)| + |\mathbb{E}_{\mathbb{P}_0} h(\boldsymbol{x}^*, \xi) - \mathbb{E}_{\bar{\mathbb{P}}} h(\bar{\boldsymbol{x}}, \xi)| \\
&\le \mathbb{E}_{\mathbb{P}_0} |h(\bar{\boldsymbol{x}}, \xi) - h(\boldsymbol{x}^*, \xi)| + L^* \\
&\le \|\bar{\boldsymbol{x}} - \boldsymbol{x}^*\| \cdot \mathbb{E}_{\mathbb{P}_0} L(\xi) + L^*.
\end{aligned}
$$

The above display holds $\mathbb{P}_0^n$-almost surely. Similarly, since $|\mathbb{E}_{\mathbb{P}^*} h(\boldsymbol{x}^*, \xi) - \mathbb{E}_{\bar{\mathbb{P}}} h(\bar{\boldsymbol{x}}, \xi)| = L^*$, we have

$$
\begin{aligned}
|\mathbb{E}_{\mathbb{P}_0} h(\boldsymbol{x}^*, \xi) - \mathbb{E}_{\mathbb{P}^*} h(\boldsymbol{x}^*, \xi)| &\le |\mathbb{E}_{\mathbb{P}_0} h(\boldsymbol{x}^*, \xi) - \mathbb{E}_{\bar{\mathbb{P}}} h(\bar{\boldsymbol{x}}, \xi)| + |\mathbb{E}_{\mathbb{P}^*} h(\boldsymbol{x}^*, \xi) - \mathbb{E}_{\bar{\mathbb{P}}} h(\bar{\boldsymbol{x}}, \xi)| \\
&\le L^* + L^* = 2L^*.
\end{aligned}
$$

The above display holds $\mathbb{P}_0^n$-almost surely. This completes the proof. □

## N   Proof of Corollary 3

**Proof.**   We have

$$
\begin{aligned}
\mathbb{E}_{\mathbb{P}_0^n} |\mathbb{E}_{\mathbb{P}_0} h(\bar{\boldsymbol{x}}, \xi) - \mathbb{E}_{\bar{\mathbb{P}}} h(\bar{\boldsymbol{x}}, \xi)| &= \mathbb{E}_{\mathbb{P}_0^n} |\mathbb{E}_{\mathbb{P}_0} h(\bar{\boldsymbol{x}}, \xi) - \mathbb{E}_{\mathbb{P}_0} h(\boldsymbol{x}^*, \xi) + \mathbb{E}_{\mathbb{P}_0} h(\boldsymbol{x}^*, \xi) - \mathbb{E}_{\bar{\mathbb{P}}} h(\bar{\boldsymbol{x}}, \xi)| \\
&\le \mathbb{E}_{\mathbb{P}_0^n} |\mathbb{E}_{\mathbb{P}_0} h(\bar{\boldsymbol{x}}, \xi) - \mathbb{E}_{\mathbb{P}_0} h(\boldsymbol{x}^*, \xi)| + \mathbb{E}_{\mathbb{P}_0^n} |\mathbb{E}_{\mathbb{P}_0} h(\boldsymbol{x}^*, \xi) - \mathbb{E}_{\bar{\mathbb{P}}} h(\bar{\boldsymbol{x}}, \xi)| \\
&\le \mathbb{E}_{\mathbb{P}_0^{n+1}} |h(\bar{\boldsymbol{x}}, \xi) - h(\boldsymbol{x}^*, \xi)| + \mathbb{E}_{\mathbb{P}_0^n} L^* \\
&\le \mathbb{E}_{\mathbb{P}_0^{n+1}} L(\xi) \|\bar{\boldsymbol{x}} - \boldsymbol{x}^*\| + \mathbb{E}_{\mathbb{P}_0^n} L^*.
\end{aligned}
$$

The other statement is obvious. This completes the proof. □

## O    Proof of Proposition 6

**Proof.**  The relative robustness measure $L^*$ of the solution $\boldsymbol{x}^*$ satisfies

$$|\mathbb{E}_{\mathbb{P}_0} h(\boldsymbol{x}^*, \xi) - \mathbb{E}_{\bar{\mathbb{P}}} h(\bar{\boldsymbol{x}}, \xi)| \le L^* \Delta(\mathbb{P}_0, \bar{\mathbb{P}}).$$

Hence, if $h(\boldsymbol{x}, \xi)$ is $L(\xi)$-Lipschitz continuous in $\boldsymbol{x}$ on $\mathcal{X}$, the generalization error gap $|\mathbb{E}_{\mathbb{P}_0} h(\bar{\boldsymbol{x}}, \xi) - \mathbb{E}_{\bar{\mathbb{P}}} h(\bar{\boldsymbol{x}}, \xi)|$ of the nominal model $\min_{\boldsymbol{x}} \mathbb{E}_{\bar{\mathbb{P}}} h(\boldsymbol{x}, \xi)$ can be given as

$$
\begin{aligned}
|\mathbb{E}_{\mathbb{P}_0} h(\bar{\boldsymbol{x}}, \xi) - \mathbb{E}_{\bar{\mathbb{P}}} h(\bar{\boldsymbol{x}}, \xi)| \quad &= |\mathbb{E}_{\mathbb{P}_0} h(\bar{\boldsymbol{x}}, \xi) - \mathbb{E}_{\mathbb{P}_0} h(\boldsymbol{x}^*, \xi) + \mathbb{E}_{\mathbb{P}_0} h(\boldsymbol{x}^*, \xi) - \mathbb{E}_{\bar{\mathbb{P}}} h(\bar{\boldsymbol{x}}, \xi)| \\
&\le |\mathbb{E}_{\mathbb{P}_0} h(\bar{\boldsymbol{x}}, \xi) - \mathbb{E}_{\mathbb{P}_0} h(\boldsymbol{x}^*, \xi)| + |\mathbb{E}_{\mathbb{P}_0} h(\boldsymbol{x}^*, \xi) - \mathbb{E}_{\bar{\mathbb{P}}} h(\bar{\boldsymbol{x}}, \xi)| \\
&\le \mathbb{E}_{\mathbb{P}_0} |h(\bar{\boldsymbol{x}}, \xi) - h(\boldsymbol{x}^*, \xi)| + L^* \Delta(\mathbb{P}_0, \bar{\mathbb{P}}) \\
&\le \mathbb{E}_{\mathbb{P}_0} L(\xi) \|\bar{\boldsymbol{x}} - \boldsymbol{x}^*\| + L^* \Delta(\mathbb{P}_0, \bar{\mathbb{P}}).
\end{aligned}
$$

The above display holds $\mathbb{P}_0^n$-almost surely. Similarly, since $|\mathbb{E}_{\mathbb{P}^*} h(\boldsymbol{x}^*, \xi) - \mathbb{E}_{\bar{\mathbb{P}}} h(\bar{\boldsymbol{x}}, \xi)| = L^* \Delta(\bar{\mathbb{P}}, \mathbb{P}^*)$, we have

$$
\begin{aligned}
|\mathbb{E}_{\mathbb{P}_0} h(\boldsymbol{x}^*, \xi) - \mathbb{E}_{\mathbb{P}^*} h(\boldsymbol{x}^*, \xi)| \quad &\le |\mathbb{E}_{\mathbb{P}_0} h(\boldsymbol{x}^*, \xi) - \mathbb{E}_{\bar{\mathbb{P}}} h(\bar{\boldsymbol{x}}, \xi)| + |\mathbb{E}_{\bar{\mathbb{P}}} h(\bar{\boldsymbol{x}}, \xi) - \mathbb{E}_{\mathbb{P}^*} h(\boldsymbol{x}^*, \xi)| \\
&= |\mathbb{E}_{\mathbb{P}_0} h(\boldsymbol{x}^*, \xi) - \mathbb{E}_{\bar{\mathbb{P}}} h(\bar{\boldsymbol{x}}, \xi)| + |\mathbb{E}_{\mathbb{P}^*} h(\boldsymbol{x}^*, \xi) - \mathbb{E}_{\bar{\mathbb{P}}} h(\bar{\boldsymbol{x}}, \xi)| \\
&\le L^* \Delta(\mathbb{P}_0, \bar{\mathbb{P}}) + L^* \Delta(\mathbb{P}^*, \bar{\mathbb{P}}) \\
&= L^* \cdot [\Delta(\mathbb{P}_0, \bar{\mathbb{P}}) + \Delta(\mathbb{P}^*, \bar{\mathbb{P}})].
\end{aligned}
$$

The above display holds $\mathbb{P}_0^n$-almost surely. This completes the proof. $\qquad\square$

## P    Proof of Corollary 4

**Proof.**  We have

$$
\begin{aligned}
\mathbb{E}_{\mathbb{P}_0^n} |\mathbb{E}_{\mathbb{P}_0} h(\bar{\boldsymbol{x}}, \xi) - \mathbb{E}_{\bar{\mathbb{P}}} h(\bar{\boldsymbol{x}}, \xi)| \quad &= \mathbb{E}_{\mathbb{P}_0^n} |\mathbb{E}_{\mathbb{P}_0} h(\bar{\boldsymbol{x}}, \xi) - \mathbb{E}_{\mathbb{P}_0} h(\boldsymbol{x}^*, \xi) + \mathbb{E}_{\mathbb{P}_0} h(\boldsymbol{x}^*, \xi) - \mathbb{E}_{\bar{\mathbb{P}}} h(\bar{\boldsymbol{x}}, \xi)| \\
&\le \mathbb{E}_{\mathbb{P}_0^n} |\mathbb{E}_{\mathbb{P}_0} h(\bar{\boldsymbol{x}}, \xi) - \mathbb{E}_{\mathbb{P}_0} h(\boldsymbol{x}^*, \xi)| + \mathbb{E}_{\mathbb{P}_0^n} |\mathbb{E}_{\mathbb{P}_0} h(\boldsymbol{x}^*, \xi) - \mathbb{E}_{\bar{\mathbb{P}}} h(\bar{\boldsymbol{x}}, \xi)| \\
&\le \mathbb{E}_{\mathbb{P}_0^{n+1}} |h(\bar{\boldsymbol{x}}, \xi) - h(\boldsymbol{x}^*, \xi)| + \mathbb{E}_{\mathbb{P}_0^n} L^* \Delta(\mathbb{P}_0, \bar{\mathbb{P}}) \\
&\le \mathbb{E}_{\mathbb{P}_0^{n+1}} L(\xi) \|\bar{\boldsymbol{x}} - \boldsymbol{x}^*\| + \mathbb{E}_{\mathbb{P}_0^n} L^* \Delta(\mathbb{P}_0, \bar{\mathbb{P}}).
\end{aligned}
$$

The other statement is obvious. This completes the proof. $\qquad\square$

## Q    Concrete Examples

In this section, we give concrete examples to provide an intuitive understanding of the relations among the existing frameworks: the true model (1), the nominal model (2), the empirical ERM method (3), the Bayesian method (7), the min-max DRO method (8), and the regularization method (9). Specifically, covariance estimation and linear regression are discussed in Subsections Q.1 and Q.2, respectively.

### Q.1    Covariance Estimation

Suppose we have $n$ samples $\{\xi_i\}_{i \in [n]}$ from the true $m$-dimensional Gaussian distribution $\mathbb{P}_0$ with zero mean and covariance $\boldsymbol{\Sigma}_0$. In many engineering problems, we need to estimate the covariance matrix $\boldsymbol{\Sigma}_0$ and the precision matrix $\boldsymbol{\Sigma}_0^{-1}$ (Mohan et al., 2012; Chen et al., 2010; Nguyen et al., 2022). The sample covariance matrix can be obtained as

$$\hat{\boldsymbol{\Sigma}}_n = \frac{1}{n} \sum_{i=1}^n \xi_i \xi_i^\top,$$

which can serve as an estimator of $\boldsymbol{\Sigma}_0$. However, when the dimension $m$ of $\xi$ is larger than the sample size $n$, it is well believed that $\hat{\boldsymbol{\Sigma}}_n$ is not a good estimator (Ledoit & Wolf, 2012). For the covariance estimation case, the true optimization formula, i.e., (1), is particularized into

$$\min_{\boldsymbol{X} \succ \boldsymbol{0}} -2\mathbb{E}_{\mathbb{P}_0} \langle \boldsymbol{X}, \xi\xi^\top \rangle + \operatorname{Tr}\left[\boldsymbol{X}^2\right], \tag{47}$$

which leads to

$$\min_{\boldsymbol{X} \succ \boldsymbol{0}} -2 \langle \boldsymbol{X}, \boldsymbol{\Sigma}_0 \rangle + \mathrm{Tr}\left[ \boldsymbol{X}^2 \right]. \tag{48}$$

When $\mathbb{P}_0$ is unknown but we have an estimate $\bar{\mathbb{P}}$ of $\mathbb{P}_0$, the nominal optimization formula, i.e., (2), is

$$\min_{\boldsymbol{X} \succ \boldsymbol{0}} -2\mathbb{E}_{\bar{\mathbb{P}}} \langle \boldsymbol{X}, \xi\xi^\top \rangle + \mathrm{Tr}\left[ \boldsymbol{X}^2 \right], \tag{49}$$

whose explicit form, after denoting $\bar{\boldsymbol{\Sigma}} := \mathbb{E}_{\bar{\mathbb{P}}}[\xi\xi^\top]$, is

$$\min_{\boldsymbol{X} \succ \boldsymbol{0}} -2 \langle \boldsymbol{X}, \bar{\boldsymbol{\Sigma}} \rangle + \mathrm{Tr}\left[ \boldsymbol{X}^2 \right]. \tag{50}$$

When $\bar{\mathbb{P}}$ is specified by the empirical distribution $\hat{\mathbb{P}}_n$, we have the empirical risk minimization formula, i.e., (3),

$$\min_{\boldsymbol{X} \succ \boldsymbol{0}} -2\mathbb{E}_{\hat{\mathbb{P}}_n} \langle \boldsymbol{X}, \xi\xi^\top \rangle + \mathrm{Tr}\left[ \boldsymbol{X}^2 \right], \tag{51}$$

which leads to

$$\min_{\boldsymbol{X} \succ \boldsymbol{0}} -2 \langle \boldsymbol{X}, \hat{\boldsymbol{\Sigma}}_n \rangle + \mathrm{Tr}\left[ \boldsymbol{X}^2 \right]. \tag{52}$$

Directly employing the nominal distribution $\bar{\mathbb{P}}$ and conducting the nominal optimization (49) may cause significant performance degradation. We assume that $\mathbb{P}_0$ is included in the distributional ball $B_\epsilon(\bar{\mathbb{P}})$.

The Bayesian method (7) assumes that there exists a second-order probability measure $\mathbb{Q}$ on $B_\epsilon(\bar{\mathbb{P}})$. Therefore, the Bayesian counterpart for the nominal model (49) is

$$\min_{\boldsymbol{X} \succ \boldsymbol{0}} -2 \langle \boldsymbol{X}, \mathbb{E}_{\mathbb{Q}}\boldsymbol{\Sigma} \rangle + \mathrm{Tr}\left[ \boldsymbol{X}^2 \right], \tag{53}$$

where $\boldsymbol{\Sigma} = \mathbb{E}_{\mathbb{P}}[\xi\xi^\top]$ for every $\mathbb{P}$ in $B_\epsilon(\bar{\mathbb{P}})$. According to Theorem 1, if $\mathbb{P}'$ is the mean of $\mathbb{P}$ under $\mathbb{Q}$, then (53) becomes

$$\min_{\boldsymbol{X} \succ \boldsymbol{0}} -2 \langle \boldsymbol{X}, \boldsymbol{\Sigma}' \rangle + \mathrm{Tr}\left[ \boldsymbol{X}^2 \right], \tag{54}$$

where $\boldsymbol{\Sigma}' = \mathbb{E}_{\mathbb{P}'}[\xi\xi^\top]$. Hence, if $\mathbb{P}'$ is closer to $\mathbb{P}_0$ than $\bar{\mathbb{P}}$ (i.e., $\boldsymbol{\Sigma}'$ is closer to $\boldsymbol{\Sigma}_0$ than $\bar{\boldsymbol{\Sigma}}$), then the Bayesian counterpart (54) has smaller generalization errors than the nominal model (49), due to Fact 2. In addition, if $\mathbb{Q}$ is specified as a Dirichlet-process-like prior, then due to (23), we have $\mathbb{E}_{\mathbb{Q}}\boldsymbol{\Sigma} = \beta_n\hat{\boldsymbol{\Sigma}} + (1 - \beta_n)\hat{\boldsymbol{\Sigma}}_n$, where $\hat{\boldsymbol{\Sigma}} = \mathbb{E}_{\hat{\mathbb{P}}}[\xi\xi^\top]$ and $\hat{\mathbb{P}}$ is a prior estimate of $\mathbb{P}_0$ before seeing samples $\{\xi_i\}_{i\in[n]}$. As a result, the Bayesian counterpart (53) becomes

$$\min_{\boldsymbol{X} \succ \boldsymbol{0}} -2 \langle \boldsymbol{X}, \beta_n\hat{\boldsymbol{\Sigma}} + (1 - \beta_n)\hat{\boldsymbol{\Sigma}}_n \rangle + \mathrm{Tr}\left[ \boldsymbol{X}^2 \right], \tag{55}$$

which is known as a shrinkage covariance estimator if $\hat{\boldsymbol{\Sigma}} := \boldsymbol{I}$ (Chen et al., 2010). Model (55) is equivalent (in the sense of having the same minimizer) to

$$\min_{\boldsymbol{X} \succ \boldsymbol{0}} -2 \langle \boldsymbol{X}, \hat{\boldsymbol{\Sigma}}_n \rangle + \mathrm{Tr}\left[ \boldsymbol{X}^2 \right] + \frac{\beta_n}{1 - \beta_n}\left\{ -2 \langle \boldsymbol{X}, \hat{\boldsymbol{\Sigma}} \rangle + \mathrm{Tr}\left[ \boldsymbol{X}^2 \right] \right\}. \tag{56}$$

By letting $\lambda_n := \frac{\beta_n}{1 - \beta_n}$ and $f_1(\boldsymbol{X}) := -2 \langle \boldsymbol{X}, \hat{\boldsymbol{\Sigma}} \rangle + \mathrm{Tr}\left[ \boldsymbol{X}^2 \right]$, (56) becomes

$$\min_{\boldsymbol{X} \succ \boldsymbol{0}} -2 \langle \boldsymbol{X}, \hat{\boldsymbol{\Sigma}}_n \rangle + \mathrm{Tr}\left[ \boldsymbol{X}^2 \right] + \lambda_n f_1(\boldsymbol{X}), \tag{57}$$

which is a regularized empirical risk minimization model; cf. (9). For theoretical details, recall Proposition 3 and Remark 3.

The min-max distributionally robust optimization (DRO) method (8), unlike the Bayesian method (7), does not assume the existence of $\mathbb{Q}$ on $B_\epsilon(\bar{\mathbb{P}})$. Instead, the DRO method works against the worst case in $B_\epsilon(\bar{\mathbb{P}})$. Therefore, the DRO counterpart for the nominal model (49) is

$$\min_{\boldsymbol{X} \succ \boldsymbol{0}} \max_{\mathbb{P} \in B_\epsilon(\bar{\mathbb{P}})} -2\mathbb{E}_{\mathbb{P}} \langle \boldsymbol{X}, \xi\xi^\top \rangle + \mathrm{Tr}\left[ \boldsymbol{X}^2 \right]. \tag{58}$$

If $B_\epsilon(\bar{\mathbb{P}})$ is specified by the Wasserstein distance, then (58) is particularized to

$$\min_{\boldsymbol{X} \succ \boldsymbol{0}} \max_{\boldsymbol{\Sigma} \succ \boldsymbol{0}} \quad -2 \langle \boldsymbol{X}, \boldsymbol{\Sigma} \rangle + \mathrm{Tr}\left[\boldsymbol{X}^2\right]$$
$$\text{s.t.} \qquad \mathrm{Tr}\left[\boldsymbol{\Sigma} + \bar{\boldsymbol{\Sigma}} - 2\left(\boldsymbol{\Sigma}^{\frac{1}{2}} \bar{\boldsymbol{\Sigma}} \boldsymbol{\Sigma}^{\frac{1}{2}}\right)^{\frac{1}{2}}\right] \leq \epsilon^2. \tag{59}$$

Note that by replacing $\bar{\boldsymbol{\Sigma}}$ with $\hat{\boldsymbol{\Sigma}}_n$, (59) is a data-driven DRO model at the empirical distribution $\hat{\mathbb{P}}_n$. Supposing $\boldsymbol{\Sigma}^*$ solves (59), we have the $\boldsymbol{X}$-point-wise upper bound for the true objective function as

$$-2 \langle \boldsymbol{X}, \boldsymbol{\Sigma}_0 \rangle + \mathrm{Tr}\left[\boldsymbol{X}^2\right] \leq -2 \langle \boldsymbol{X}, \boldsymbol{\Sigma}^* \rangle + \mathrm{Tr}\left[\boldsymbol{X}^2\right], \quad \forall \boldsymbol{X}.$$

Hence, by conducting the min-max DRO method (8), the true cost at $\boldsymbol{\Sigma}_0$ can also be reduced. Existing literature has established the relation between the min-max DRO method (8) and the regularized method (9). To be specific, according to Appendix C.3, the objective function of the min-max DRO formulation (59) is $\boldsymbol{X}$-point-wise upper bounded by a regularized objective function:

$$\max_{\boldsymbol{\Sigma} \succ \boldsymbol{0}} -2 \langle \boldsymbol{X}, \boldsymbol{\Sigma} \rangle + \mathrm{Tr}\left[\boldsymbol{X}^2\right] \leq -2 \langle \boldsymbol{X}, \bar{\boldsymbol{\Sigma}} \rangle + \mathrm{Tr}\left[\boldsymbol{X}^2\right] + \epsilon \cdot f_2(\boldsymbol{X}), \quad \forall \boldsymbol{X},$$

where the regularizer $f_2(\boldsymbol{X})$ is induced by the cost function $h(\boldsymbol{X}, \xi) := -2\xi^\top \boldsymbol{X} \xi + \mathrm{Tr}\left[\boldsymbol{X}^2\right]$; the strict equivalence, however, cannot be guaranteed to hold because the cost function $h(\boldsymbol{X}, \xi)$ is not convex in $\xi$.

The regularization counterpart (9) for the nominal model (49) is

$$\min_{\boldsymbol{X} \succ \boldsymbol{0}} -2 \langle \boldsymbol{X}, \bar{\boldsymbol{\Sigma}} \rangle + \mathrm{Tr}\left[\boldsymbol{X}^2\right] + \lambda f(\boldsymbol{X}). \tag{60}$$

When we consider the nominal distribution to be $\hat{\mathbb{P}}_n$, we have

$$\min_{\boldsymbol{X} \succ \boldsymbol{0}} -2 \left\langle \boldsymbol{X}, \hat{\boldsymbol{\Sigma}}_n \right\rangle + \mathrm{Tr}\left[\boldsymbol{X}^2\right] + \lambda_n f(\boldsymbol{X}). \tag{61}$$

In the practice of machine learning, typical choices for $f(\boldsymbol{X})$ include $\|\boldsymbol{X}\|_2$ (i.e., Ridge) and $\|\boldsymbol{X}\|_1$ (i.e., LASSO). On the other hand, the randomized learning counterpart for the empirically nominal model (51), i.e., PAC-Bayesian model (28), is

$$\min_{\mathbb{Q}_{\boldsymbol{X}}: \boldsymbol{X} \succ \boldsymbol{0}} \mathbb{E}_{\mathbb{Q}_{\boldsymbol{X}}} \left\{ -2 \left\langle \boldsymbol{X}, \hat{\boldsymbol{\Sigma}}_n \right\rangle + \mathrm{Tr}\left[\boldsymbol{X}^2\right] \right\} + \lambda_n \cdot \mathrm{KL}(\mathbb{Q}_{\boldsymbol{X}} \| \Pi_{\boldsymbol{X}}), \tag{62}$$

where $\mathbb{Q}_{\boldsymbol{X}}$ is a distribution of $\boldsymbol{X}$. When we let $\mathbb{Q}_{\boldsymbol{X}}$ be a point-mass distribution, (62) degenerates to

$$\min_{\boldsymbol{X} \succ \boldsymbol{0}} -2 \left\langle \boldsymbol{X}, \hat{\boldsymbol{\Sigma}}_n \right\rangle + \mathrm{Tr}\left[\boldsymbol{X}^2\right] + \lambda_n \cdot f_3(\boldsymbol{X}), \tag{63}$$

where $f_3(\boldsymbol{X}) := -\log \pi(\boldsymbol{X})$ and $\pi(\boldsymbol{X})$ is the density of $\Pi_{\boldsymbol{X}}$ with respect to the Lebesgue measure. This is the first Bayesian interpretation of the regularized empirical risk minimization model (61), which is widely used in the existing literature. In this paper, however, we propose another new Bayesian interpretation (7) for (61). Let $\beta_n$ be such that $\lambda_n = \frac{\beta_n}{1-\beta_n}$ and $\hat{\mathbb{P}}$ be a uniform distribution on $B_\epsilon(\hat{\mathbb{P}}_n)$ with density function, with respect to the Lebesgue measure $\mathcal{L}$,

$$\frac{\mathrm{d}\hat{\mathbb{P}}}{\mathrm{d}\mathcal{L}} := \frac{f(\boldsymbol{X})}{\displaystyle\int_\Xi -2\xi^\top \boldsymbol{X} \xi + \mathrm{Tr}\left[\boldsymbol{X}^2\right] \mathrm{d}\xi}, \quad \forall \boldsymbol{X}.$$

As a result, (61) becomes a Bayesian method (55), or (53), or (7):

$$\min_{\boldsymbol{X} \succ \boldsymbol{0}} \mathbb{E}_{\mathbb{Q}} \mathbb{E}_{\mathbb{P}} \left\{ -2 \left\langle \boldsymbol{X}, \xi\xi^\top \right\rangle + \mathrm{Tr}\left[\boldsymbol{X}^2\right] \right\} = \min_{\boldsymbol{X} \succ \boldsymbol{0}} -2 \left\langle \boldsymbol{X}, \mathbb{E}_{\mathbb{Q}} \boldsymbol{\Sigma} \right\rangle + \mathrm{Tr}\left[\boldsymbol{X}^2\right];$$

for theoretical details on this point, recall Proposition 3 and Remark 3. Note that the difference between the two Bayesian interpretations is over which space the prior distribution $\mathbb{Q}$ is used: the hypothesis space for $\boldsymbol{X}$ (i.e., the first interpretation) or the data space for $\mathbb{P} \in B_\epsilon(\hat{\mathbb{P}}_n)$ (i.e., the second interpretation); see also Remark 4.

### Q.2 Linear Regression

Given the results for covariance estimation in Subsection Q.1, the discussion on the case of linear regression is technically straightforward. We just highlight some main points.

Consider the linear regression model $\xi_{\text{out}} = \boldsymbol{x}^\top \xi_{\text{in}} + e$ where $e \in \mathbb{R}$ denotes the regression error. Suppose that the data are sampled from a zero-mean Gaussian distribution, that is,

$$\xi := \begin{bmatrix} \xi_{\text{in}} \\ \xi_{\text{out}} \end{bmatrix} \sim \mathcal{N}(\boldsymbol{0}, \boldsymbol{\Sigma}_0).$$

The true optimization problem (1) is particularized into

$$\min_{\boldsymbol{x} \in \mathcal{X}} \begin{bmatrix} \boldsymbol{x} \\ -1 \end{bmatrix}^\top \boldsymbol{\Sigma}_0 \begin{bmatrix} \boldsymbol{x} \\ -1 \end{bmatrix}. \tag{64}$$

The nominal model (2) becomes

$$\min_{\boldsymbol{x} \in \mathcal{X}} \begin{bmatrix} \boldsymbol{x} \\ -1 \end{bmatrix}^\top \bar{\boldsymbol{\Sigma}} \begin{bmatrix} \boldsymbol{x} \\ -1 \end{bmatrix}. \tag{65}$$

The Bayesian counterpart for the nominal model (65) is

$$\min_{\boldsymbol{x} \in \mathcal{X}} \begin{bmatrix} \boldsymbol{x} \\ -1 \end{bmatrix}^\top [\mathbb{E}_{\mathbb{Q}} \boldsymbol{\Sigma}] \begin{bmatrix} \boldsymbol{x} \\ -1 \end{bmatrix}. \tag{66}$$

and under a Dirichlet-process-like prior $\mathbb{Q}$ with prior $\hat{\mathbb{P}}$, (66) further becomes

$$\min_{\boldsymbol{x} \in \mathcal{X}} \begin{bmatrix} \boldsymbol{x} \\ -1 \end{bmatrix}^\top \left[ \beta_n \hat{\boldsymbol{\Sigma}} + (1 - \beta_n) \hat{\boldsymbol{\Sigma}}_n \right] \begin{bmatrix} \boldsymbol{x} \\ -1 \end{bmatrix}, \tag{67}$$

which is equivalent (in the sense of having the same optimizer) to

$$\min_{\boldsymbol{x} \in \mathcal{X}} \begin{bmatrix} \boldsymbol{x} \\ -1 \end{bmatrix}^\top \hat{\boldsymbol{\Sigma}}_n \begin{bmatrix} \boldsymbol{x} \\ -1 \end{bmatrix} + \frac{\beta_n}{1 - \beta_n} \begin{bmatrix} \boldsymbol{x} \\ -1 \end{bmatrix}^\top \hat{\boldsymbol{\Sigma}} \begin{bmatrix} \boldsymbol{x} \\ -1 \end{bmatrix}. \tag{68}$$

By letting $\lambda_n := \frac{\beta_n}{1 - \beta_n}$ and $f(\boldsymbol{x}) := \begin{bmatrix} \boldsymbol{x} \\ -1 \end{bmatrix}^\top \hat{\boldsymbol{\Sigma}} \begin{bmatrix} \boldsymbol{x} \\ -1 \end{bmatrix}$, (68) becomes a $\hat{\boldsymbol{\Sigma}}$-Tikhonov regularized ERM model (9), i.e.,

$$\min_{\boldsymbol{x} \in \mathcal{X}} \begin{bmatrix} \boldsymbol{x} \\ -1 \end{bmatrix}^\top \hat{\boldsymbol{\Sigma}}_n \begin{bmatrix} \boldsymbol{x} \\ -1 \end{bmatrix} + \lambda_n f(\boldsymbol{x}). \tag{69}$$

When $\hat{\boldsymbol{\Sigma}}$ is an identity matrix, (69) further becomes a Ridge regression model.

In addition, the min-max DRO counterpart for the nominal model (65) under the Wasserstein distance is particularized into

$$\begin{aligned} \min_{\boldsymbol{x} \in \mathcal{X}} \max_{\boldsymbol{\Sigma}} \quad & \begin{bmatrix} \boldsymbol{x} \\ -1 \end{bmatrix}^\top \boldsymbol{\Sigma} \begin{bmatrix} \boldsymbol{x} \\ -1 \end{bmatrix} \\ \text{s.t.} \quad & \text{Tr} \left[ \boldsymbol{\Sigma} + \bar{\boldsymbol{\Sigma}} - 2 \left( \boldsymbol{\Sigma}^{\frac{1}{2}} \bar{\boldsymbol{\Sigma}} \boldsymbol{\Sigma}^{\frac{1}{2}} \right)^{\frac{1}{2}} \right] \leq \epsilon^2. \end{aligned} \tag{70}$$

## R Closing Notes on Generalization Error Bounds

The main points regarding generalization error bounds can be recapped as follows, by considering the nominal distribution $\bar{\mathbb{P}}$ and its induced distributional ball $B_\epsilon(\bar{\mathbb{P}})$ such that $\mathbb{P}_0 \in B_\epsilon(\bar{\mathbb{P}})$.

1. When the Bayesian method (7) is solved by $\boldsymbol{x}^*$, according to Proposition 2 and (46), we have, with probability at least $1 - \eta$,

$$|\mathbb{E}_{\mathbb{P}} h(\boldsymbol{x}^*, \xi) - \mathbb{E}_{\mathbb{P}_0} h(\boldsymbol{x}_0, \xi)| \leq L^*, \quad \forall \mathbb{P} \in B_\epsilon(\bar{\mathbb{P}})$$

where the robustness measure $L^*$ for the robust solution $\boldsymbol{x}^*$ is

$$L^* = \frac{\mathbb{E}_{\mathbb{Q}} \mathbb{E}_{\mathbb{P}} h(\boldsymbol{x}^*, \xi) + \mathbb{E}_{\mathbb{P}_0} h(\boldsymbol{x}_0, \xi)}{\eta},$$

provided that $h$ is non-negative. In addition, due to the arbitrariness of $\mathbb{P}$, with probability at least $1 - \eta$, the absolute excess risk satisfies

$$|\mathbb{E}_{\mathbb{P}_0} h(\boldsymbol{x}^*, \xi) - \mathbb{E}_{\mathbb{P}_0} h(\boldsymbol{x}_0, \xi)| \leq L^*.$$

This explains why Bayesian methods have the potential to generalize well; a similar statement can be obtained using Remark 2.[13] Note that minimizing $L$ is equivalent to minimizing $\mathbb{E}_{\mathbb{Q}} \mathbb{E}_{\mathbb{P}} h(\boldsymbol{x}, \xi)$ over $\boldsymbol{x}$, i.e., conducting (7); note also that the better the $\mathbb{Q}$, the smaller the $L^*$. (The better the $\mathbb{Q}$, the more $\mathbb{P}$ concentrate around $\mathbb{P}_0$; the best $\mathbb{Q}$'s are such that $\mathbb{E}_{\mathbb{Q}} \mathbb{E}_{\mathbb{P}} h(\boldsymbol{x}, \xi) = \mathbb{E}_{\mathbb{P}_0} h(\boldsymbol{x}, \xi)$; cf. Theorem 1.)

2. When the surrogate min-max DRO (8) or (19) is solved by $(\boldsymbol{x}^*, \mathbb{P}^*)$, according to Definition 5 and Corollary 1, we have the robustness measure $L^* = \mathbb{E}_{\mathbb{P}^*} h(\boldsymbol{x}^*, \xi) - \mathbb{E}_{\bar{\mathbb{P}}} h(\bar{\boldsymbol{x}}, \xi)$ for the solution $\boldsymbol{x}^*$; note that $(\boldsymbol{x}^*, L^*)$ solves (15). Therefore, the true cost (i.e., generalization error) $\mathbb{E}_{\mathbb{P}_0} h(\boldsymbol{x}^*, \xi)$ is upper bounded by

$$\mathbb{E}_{\mathbb{P}_0} h(\boldsymbol{x}^*, \xi) \leq \mathbb{E}_{\mathbb{P}^*} h(\boldsymbol{x}^*, \xi) = \mathbb{E}_{\bar{\mathbb{P}}} h(\bar{\boldsymbol{x}}, \xi) + L^*.$$

In addition, the (one-sided) excess risk satisfies

$$\mathbb{E}_{\mathbb{P}_0} h(\boldsymbol{x}^*, \xi) - \mathbb{E}_{\mathbb{P}_0} h(\boldsymbol{x}_0, \xi) = \mathbb{E}_{\mathbb{P}_0} [h(\boldsymbol{x}^*, \xi) - h(\boldsymbol{x}_0, \xi)] \leq \|\boldsymbol{x}^* - \boldsymbol{x}_0\| \cdot \mathbb{E}_{\mathbb{P}_0} L(\xi),$$

where $L(\xi)$ is the Lipschitz constant of $h(\cdot, \xi)$. This explains why min-max DRO methods have the potential to generalize well; note that minimizing $L$ is equivalent to minimizing the upper bound of the true cost. Therefore, min-max DRO methods serve as computational solution methods for (15) and find one-sided robustness measures $L$. (For computational methods to find two-sided robustness measures, see Appendix D.5.)

3. When the regularized method (9) at $\bar{\mathbb{P}} := \hat{\mathbb{P}}_n$ is solved by $\boldsymbol{x}^*$, due to Proposition 3, we have, with probability at least $1 - \eta$,

$$|\mathbb{E}_{\mathbb{P}} h(\boldsymbol{x}^*, \xi) - \mathbb{E}_{\mathbb{P}_0} h(\boldsymbol{x}_0, \xi)| \leq L^*, \quad \forall \mathbb{P} \in B_\epsilon(\hat{\mathbb{P}}_n)$$

where the robustness measure $L^*$ for the robust solution $\boldsymbol{x}^*$ is

$$L^* = \frac{(1 - \beta_n) \left[ \mathbb{E}_{\hat{\mathbb{P}}_n} h(\boldsymbol{x}^*, \xi) + \lambda_n f(\boldsymbol{x}^*) \right] + \mathbb{E}_{\mathbb{P}_0} h(\boldsymbol{x}_0, \xi)}{\eta},$$

provided that $h$ is non-negative and $\beta_n$ is defined through $\lambda_n = \frac{\beta_n}{1 - \beta_n}$. Note that

$$\begin{aligned} \mathbb{E}_{\hat{\mathbb{P}}_n} h(\boldsymbol{x}^*, \xi) + \lambda_n f(\boldsymbol{x}^*) &:= \mathbb{E}_{\hat{\mathbb{P}}_n} h(\boldsymbol{x}^*, \xi) + \frac{\beta_n}{1 - \beta_n} \mathbb{E}_{\hat{\mathbb{P}}} h(\boldsymbol{x}^*, \xi) \\ &= \frac{1}{1 - \beta_n} \mathbb{E}_{\beta_n \hat{\mathbb{P}} + (1 - \beta_n) \hat{\mathbb{P}}_n} h(\boldsymbol{x}^*, \xi) \\ &= \frac{1}{1 - \beta_n} \mathbb{E}_{\mathbb{Q}} \mathbb{E}_{\mathbb{P}} h(\boldsymbol{x}^*, \xi). \end{aligned}$$

In addition, due to the arbitrariness of $\mathbb{P}$, with probability at least $1 - \eta$, the absolute excess risk satisfies

$$|\mathbb{E}_{\mathbb{P}_0} h(\boldsymbol{x}^*, \xi) - \mathbb{E}_{\mathbb{P}_0} h(\boldsymbol{x}_0, \xi)| \leq L^*.$$

---

[13] With probability at least $1 - \eta$, $\mathbb{E}_{\mathbb{P}} h(\boldsymbol{x}^*, \xi) \leq L^*$ for every $\mathbb{P}$ (hence the generalization error at $\boldsymbol{x}^*$ satisfies $\mathbb{E}_{\mathbb{P}_0} h(\boldsymbol{x}^*, \xi) \leq L^*$), where $L^* = \mathbb{E}_{\mathbb{Q}} \mathbb{E}_{\mathbb{P}} h(\boldsymbol{x}^*, \xi) / \eta$. Note that $L^*$ here is not a robustness measure.

This explains why regularized methods have the potential to generalize well. Note that minimizing $L$ is equivalent to minimizing $\mathbb{E}_{\hat{\mathbb{P}}_n} h(\boldsymbol{x}, \xi) + \lambda_n f(\boldsymbol{x})$ over $\boldsymbol{x}$, i.e., solving (9); note also that the better the $(\lambda_n, f(\boldsymbol{x}))$, the smaller the $L^*$. (The best $(\lambda_n, f(\boldsymbol{x}))$'s are such that $\mathbb{E}_{\hat{\mathbb{P}}_n} h(\boldsymbol{x}, \xi) + \lambda_n f(\boldsymbol{x}) = \mathbb{E}_{\mathbb{P}_0} h(\boldsymbol{x}, \xi)$; see also Remark 5.)

To conclude, the benefits of bounding generalization errors by robustness measures are summarized in Figure 1, Subsection 2.4.3, and Remark 9.

