# OpenReview forum: "Distributional Robustness Bounds Generalization Errors"
_TMLR — Withdrawn by Authors_

### Review · Reviewer_2Dgt · 2025-03-07

**Summary Of Contributions:**

There is a lot going on in this paper, but in broad terms the authors are interested in looking at distinct categories of learning problems, and the relationships between these problems, both in terms of the objective functions used in their formulation and the solutions to those objectives. The main categories of interest, roughly speaking, are "DRO", "Bayesian", and "additive-regularizer". Much of the paper's essence is contained in its formal definitions. In particular, the authors express a strong interest in putting the notion of distributionally-robust optimization on more solid theoretical footing.

Section 2, titled "Main Results", spans 15 pages and is broken into numerous sub-sections. They start by introducing their own definition of distributional robustness as a property of a solution, and at the same time introduce a quantity to be interpreted as measuring the degree of such robustness (the smallest upper bound on the expected loss deviations seen in Defn 1). They proceed by showing how this relates closely to traditional DRO objective functions, and jumping to sub-section 2.4 they emphasize how for smooth-enough cost functions, this quantity appears in upper bounds on the traditional "generalization error" incurred by what one would typically call an ERM solution (see Theorem 6).

The authors also show how one can essentially "plug in" the Bayesian objective function to their new distributionally robust (DR) formalization to get DR guarantees with "high probability", where the probability depends on the degree of distributional robustness. They then leverage a connection between Bayesian methods and regularized ERM to extend the DR-based connection even further.

**Audience:**

No

**Broader Impact Concerns:**

No particular concerns.

**Claims And Evidence:**

No

**Requested Changes:**

Please see the comments raised above. I do not expect the authors to respond to the questions I raised above, as they were meant to be instructional. I think the paper needs a major overhaul in order to be considered acceptable.

**Strengths And Weaknesses:**

On the positive side, establishing connections between learning tasks which appear distinct at face value is a natural and important direction for machine learning research. The authors' distributional robustness quantity of interest is natural, and their main results meticulously draw connections between different realms of learning task formulations and learning strategies. Aside from issues of clarity and some imprecise statements, I believe the results are valid, with many of them being what one would intuitively expect given the definitions and core assumptions used.

On the other hand, I feel that the clarity of writing and overall presentation of this paper is quite far below the acceptance threshold that one would expect for any respectable machine learning publication. I will give several concrete examples to back this statement up below. In addition to clarity issues, I have some concerns regarding the core insights of this work; I feel like almost all the connections established in this paper can safely be considered "known" by the community. TMLR is strict about clarity and correctness, but lenient on novelty and significance, so I will not position these concerns as being too critical, but in any case I will provide more details as feedback for the authors below.

Putting it rather bluntly, the main issue with this paper is that it is poorly written and poorly organized, especially when considered as a potential publication in TMLR.

Starting with the poor organization first, of the 19 pages of main content in the paper, 15 of those pages fall into section 2, titled "Main Results". TMLR is supposed to be a venue for clearly communicating results which might not otherwise be accepted at the usual big international ML conferences. It does not take 15 pages to communicate the main results of this paper. There is a lot of obscure prose that is in no way a "main result" (e.g., Example 1, all of section 2.1.4, etc.). Also, there are numerous definitions/results which are essentially copy-pasted with minor alterations; take for example Definitions 2, 3, and 4. Do we really need to spend a full page of the critical main results section on these definitions? After re-considering if there are indeed some "main results" here worth disseminating, I recommend the authors significantly re-structure the paper so that the main insights are communicated effectively; a "main results" section is only meaningful if it is concise and actually centered around main results.

Before getting into concrete examples of poor writing, I would like to highlight that the main motivations and claims in this paper are themselves not clear nor convincing. In 1.4 the authors establish their main motivations and try to link this to their list of contributions in 1.5, but I find this all quite sub-standard. The authors say (page 3) that *"there does not exist a quantity to measure the distributional robustness"* of what they call a *"robust solution"*. I completely disagree with this, and feel that anyone familiar with the massive literature on distributionally robust optimization would also disagree. The risk (be it empirical or not) maximized over all distributions in the uncertainty set is perfectly natural as quantifying the degree of robustness. The authors' "DR" notion just looks at the difference between this DRO risk and some natural "good" reference point. Their motivation here is like saying that the expected value of the loss at test time (what they call *"generalization error"* on page 2) does not measure off-sample generalization, but the *difference* (e.g., their (5)) does. On existing links between Bayesian, regularization, and DRO, the authors admit that connections have been established in the literature, but emphasize (capitals theirs) *"NONE of the existing literature (profoundly) discusses the three methods together in a unified framework..."*. This sentence is off-putting and really quite meaningless. If the existing connections in the literature have not been distilled down and described sufficiently clearly, then why not write a survey paper that does this?

Regarding clarity issues with the main claims in 1.5, consider for example point 4, which says that *"generalization errors can be characterized from the perspective of the distributional uncertainty of the nominal distribution"*. This claim goes way beyond what is actually shown in Theorem 6. Yes, the authors derive some upper bounds based on their DR quantities $L$ and $L^{\\ast}$. Does this in any way *characterize* the generalization error gap to be incurred by the solution of interest? Consider when $\\epsilon$ is large; this will presumably lead to increasingly large $L$ and $L^{\\ast}$, and optimality in the DRO problem (14) need not imply anything about how good the DRO solution is in terms of the true underlying risk. Similarly, an ERM solution might be great in terms of the generalization error gap but terrible in terms of the DR quantities. Upper bounds are obtained, but I would venture to say that Theorem 6 offers very little in terms of insights and nothing is "characterized". I could go on and make similar statements for the other claims as well.

Below, let me provide a bullet-point list of issues with exposition which I found in this paper. It is by no means exhaustive, but I think it provides sufficient evidence for my claim of poor writing.

- All the business related to the upper bound in section 1.2 is not really convincing. Why should minimizing any upper bound imply that the generalization error is *"also controlled"*? Consider an upper bound that bottoms out at a value which is larger than the maximum value the expected loss/cost takes; I can always create such an upper bound, but nothing is "controlled".

- The footnote on page 2: why use both SAA and ERM interchangeably? This is meaningless. If two terms refer to the same thing, choose one and be consistent. TMLR is a journal aimed at the machine learning community, and ERM is the established term. It is perfectly fine to mention that different fields have different terms for the same concepts, but I cannot understand why the authors want to (and indeed do) use both terms within one paper.

- Page 5: the authors say that distributional robustness is *"philosophically popular"* but it remains to be *"mathematically defined"*. This, as I mentioned earlier, is just not true. The authors may not like the existing definitions, which is fine, but there are numerous formal definitions which can readily be interpreted as quantifying robustness that aligns with their Philosophy 1 (page 5).

- The use of the term "model" in this paper is extremely troublesome. Following the authors' notation, I would venture to say that the vast majority of people in the machine learning community use the term "model" colloquially to refer to either the predictor/classifier/etc that is characterized by $\\boldsymbol{x}$, or the the set of those predictors/classifers/etc that is characterized by $\\mathcal{X}$. I can tell that the authors use the term "model" to refer to a learning problem characterized by a minimization task, but I think they have the wrong audience in mind.

- The authors use the term *"model set"* from page 6 onwards, but the mathematical formulation is really unclear. In addition to the issue I raised earlier about what the term "model" means to most machine learning researchers, the set notation used by the authors is in my opinion impossible to parse. They say that their function $T$ induces a model set $\\{ \\mathbb{P} \\mapsto T(\\mathbb{P}) \\,\\vert\\, \\mathbb{P} \\in B\_{\\epsilon}(\\mathbb{P}\_{0})\\}$, but I read $\\mathbb{P} \\mapsto T(\\mathbb{P})$ as representing a *function*, not the value returned by a function. Are the authors equating the function *value* $T(\\mathbb{P})$ for each $\\mathbb{P}$ with a learning problem (what they call a "model")? If so, then the usual way of writing this would be $\\{ T(\\mathbb{P}) \\,\\vert\\, \\mathbb{P} \\in B\_{\\epsilon}(\\mathbb{P}\_{0})\\}$. For a research paper aimed at professionals, whose formalism is supposed to be its main strength, this is quite sub-standard in my opinion.

- There are lots of unclear concepts peppered throughout the paper. Take for example *"distributionally robust counterpart"* which appears in bold on page 6. The authors attempt to explain it in Example 2, but I was confused. First, the "If$\\ldots$. Then$\\ldots$." structure is very unclear. Second, what does this "counterpart" even mean? $\\mathbb{P}^{\\prime}$ can be arbitrarily far from $\\mathbb{P}\_{0}$, correct? So it is a distribution which may be totally unrelated to $\\mathbb{P}\_{0}$ and data distributions similar to (close to) $\\mathbb{P}\_{0}$, but it called the DR counterpart for the ball at $\\mathbb{P}\_{0}$ if it admits a solution which is DR near $\\mathbb{P}\_{0}$... having read through it several times I was able to get the notion that the authors have in mind, but I'm left wondering whether this notion needed to be in the "main results" section, or even in the paper at all.

- Theorem 4 and Remark 2 following it: I get what the authors are trying to say, but things are quite unclear; for example, wasn't the authors' DR notion defined in terms of the difference taken over all $\\mathbb{P}$ in a ball around $\\mathbb{P}\_{0}$? What happened to that set here? Everything falls apart if the DR definition uses a $\\epsilon$ ball around $\\mathbb{P}\_{0}$ but $\\mathbb{Q}$ is a distribution over a totally different set of data distributions. Also, the authors use the term "counterpart" here; as I mentioned, the pseudo-definition for counterpart prior to Example 2 was very unclear, and it is yet another possible point of confusion if the authors are using this term strictly in Theorem 4 or not. Again, if the reader struggles and reads between the lines or ignores inconsistencies, they can probably get an idea of what the authors are trying to communicate, but it should not be such a struggle.

I could go on and on with this list, but I think the above points are sufficient.

Finally, regarding the value of the insights in this paper, are all the Theorems in section 2 supposed to be of equal value as "main results"? I feel like the majority of the results here are exactly what one would expect given the definitions and assumptions, with Theorem 6 being a prime example with smoothness assumptions for the cost function added on. As I mentioned earlier, if this were a survey paper, and the authors wanted to provide a clearer explanation of links between different objective functions, plus highlighting a few new connections which do not appear in the literature explicitly, I think that would be a very natural approach, but as it stands, the key motivations regarding the shortcomings of the existing literature are extremely weak (as discussed earlier), and the main results are mostly positioned as completely new and original. There are lots of papers which explore this kind of material. Here are a few examples below.

- [The Bayesian Learning Rule](https://arxiv.org/abs/2107.04562), Khan and Rue (JMLR 2023)
- [A Survey of Learning Criteria Going Beyond the Usual Risk](https://arxiv.org/abs/2110.04996), Holland and Tanabe (JAIR 2023)
- [Rank-based Decomposable Losses in Machine Learning: A Survey](https://arxiv.org/abs/2207.08768), Hu et al. (TPAMI 2023)
- [Risk-Adaptive Approaches to Stochastic Optimization: A Survey](https://arxiv.org/abs/2212.00856), Royset (SIAM Review 2025)

I know the above surveys do not exactly do what the authors are trying to do here, but I think they convey a bit about how much is relatively well-known by the community.

__Issue with anonymity__

On page 20, the authors include acknowledgements which should have been anonymized. Strictly speaking this is actually a pretty serious anonymity violation.

---

### Review · Reviewer_g8SC · 2025-03-22

**Summary Of Contributions:**

The manuscript explores an important topic, distributional uncertainty in machine learning, by formally defining the conceptual framework of distributional robustness and establishing connections between distributionally robust optimization methods, regularization methods, and Bayesian approaches. Notably, the authors demonstrate that any regularization method can be viewed as a special case of a Bayesian method when a non-parametric prior is selected, with an equivalence observed under a Dirichlet-process prior. Furthermore, they show that Bayesian models are probably approximately distributionally robust. Finally, the manuscript offers a new perspective for characterizing the generalization errors of machine learning models, either through the newly defined distributional uncertainty of nominal models or via robustness measures of robust solutions. Overall, the paper is well-written, with results presented using clear notation and well-defined assumptions. The inclusion of explanatory discussions following the results significantly enhances the reader’s understanding. However, several notable shortcomings should be addressed to further strengthen the manuscript.

**Audience:**

Yes

**Broader Impact Concerns:**

I do not identify any ethical concerns in this study.

**Claims And Evidence:**

Yes

**Requested Changes:**

1. In Section "2.5 Comparisons With Existing Literature", the authors should include a comparison with relevant related works, such as [1], to provide a more comprehensive discussion

References:

[1] Wang, Shixiong, Haowei Wang, and Jean Honorio. "Learning Against Distributional Uncertainty: On the Trade-off Between Robustness and Specificity." arXiv preprint arXiv:2301.13565 (2023).

2. In Theorem 2 (Min-max distributionally robust optimization), if $(\boldsymbol{x}^*,L^*)$ solves (13), it must be shown that $(\boldsymbol{x}^*,\mathbb{P}^*)$ solves (18), where
$$\mathbb{E}\_{\mathbb{P}^*}h(\boldsymbol{x}^*,\xi^*)-\mathbb{E}\_{\mathbb{P}\_0}h(\boldsymbol{x}\_0,\xi^*) = L^*.$$
This requires proving that for any $\mathbb{P}$ satisfying $\Delta(\mathbb{P},\mathbb{P}_0) \le \epsilon$ and any $\boldsymbol{x} \in \mathcal{X}$, the following holds:
$$\mathbb{E}\_{\mathbb{P}}h(\boldsymbol{x}^*,\xi) \le \mathbb{E}\_{\mathbb{P}^*}h(\boldsymbol{x}^*,\xi) \le \mathbb{E}\_{\mathbb{P}^*}h(\boldsymbol{x},\xi).$$
However, it is unclear how the current brief argument, particularly the step involving the elimination of $L$, in Appendix H. Proof of Theorem 2 establishes this result. The authors should provide a more detailed explanation to clarify the proof. Similar issues arise in the proofs of Theorem 5 and Corollary 1.

3. The authors should clearly specify and highlight the technical contributions of all the main theoretical results in the manuscript, particularly the theorems. In particular, they should elaborate on the key challenges involved in establishing these new theoretical insights. The current presentation gives the impression that these theorems are trivial results.

4. The authors should ensure consistency in their citation format. For example, the sentence: "The optimization problem (1) is also popular in several other areas beyond statistical machine learning, e.g., applied statistics Wang et al. (2022b), operations research Shapiro et al. (2009), system simulations Kouri et al. (2022), and statistical signal processing (Wang, 2022b), where the specific meanings that it conveys vary from one to another." should be revised to: "The optimization problem (1) is also popular in several other areas beyond statistical machine learning, e.g., applied statistics (Wang et al., 2022b), operations research (Shapiro et al., 2009), system simulations (Kouri et al., 2022), and statistical signal processing (Wang, 2022b), where the specific meanings it conveys vary across domains."
Maintaining a uniform citation style enhances readability and ensures clarity in referencing sources.

5.  The reference should be corrected as follows: From:
"*Ruidi Chen, Ioannis Ch Paschalidis, et al. Distributionally robust learning. Foundations and Trends® in Optimization, 4(1-2):1–243, 2020.*
To:
"*Ruidi Chen, Ioannis Ch Paschalidis. Distributionally robust learning. Foundations and Trends® in Optimization, 4(1-2):1–243, 2020.*"

**Strengths And Weaknesses:**

**Strengths:** The authors provide several practically useful insights into uncertainty-aware machine learning, including the robustness-sensitivity trade-off in handling distributional uncertainty, the effectiveness of Bayesian methods in bounding generalization error, the benefits of small distributional uncertainty, the bias-variance trade-off in regularization methods, and the min-max formulation in distributionally robust optimization. Additionally, they offer concrete examples to intuitively illustrate the relationships among existing frameworks, such as covariance estimation, inverse covariance estimation, and linear regression. These contributions highlight the practical implications of the proposed distributional robustness framework.

**Weaknesses:**

1. The authors do not provide any simulation studies, either on synthetic or real datasets, to empirically demonstrate the new insights into uncertainty-aware machine learning models discussed in the manuscript. While the primary contribution is theoretical, some empirical studies would have been expected to illustrate the advantages of the the proposed concept system of “distributional robustness” in practical data settings.

2. Most of the theorems in the manuscript appear to be straightforward extensions of existing results from the literature rather than novel contributions. As such, they would be more appropriately stated as lemmas or propositions. From a practical and computational perspective, the authors do not provide any empirical evidence to support the benefits of bounding generalization errors using robustness measures. To strengthen the manuscript, the authors should include numerical experiments similar to those presented in the related work [1].

References:

[1] Wang, Shixiong, Haowei Wang, and Jean Honorio. "Learning Against Distributional Uncertainty: On the Trade-off Between Robustness and Specificity." arXiv preprint arXiv:2301.13565 (2023).

---

### Review · Reviewer_tDaL · 2025-04-02

**Summary Of Contributions:**

The paper provides a unifying framework to investigate the connections between Bayesian stochastic optimization, distributional robust optimization (DRO), and regularization. In particular, this connection is explored by defining a "distributional robustness" measure, which, in turn, helps derive generalization error bounds.

**Audience:**

Yes

**Claims And Evidence:**

No

**Requested Changes:**

Major comments:
- Comments written in the weakness should be seen as the major comments.


Minor comments:
- In Section 1.4, it's claimed that there doesn't exist a quantity to measure the distributional robustness of a robust solution. I disagree with this and suggest re-wording. The OR/ML/Stochastic programming/Operations management literature has a long history of looking at this under terms such as regret, price of robustness, the expected value of additional information, etc., which all aim to measure distributional robustness to an uncertain probability distribution. In other words, distributional ambiguity, or as coined in this paper, "distributional uncertainty."
- Example 2: Should be "suppose $x^{\prime}$.." instead of "if $x^{\prime}$.." in the first line. Also, define $P^{\prime}$.
- Theorem 1: Should be model (2) instead of model (1).
- Appendix D.2.2, comments after Definition 13: It must be "a one-sided .." instead of "an one-sided.."

**Strengths And Weaknesses:**

Strengths:

- The paper is mostly well-written. I found it easy to follow the main components, at least individually.
- The paper should be seen as "survey/review" paper that could be helpful to beginners as well.
- The paper clarifies many concepts in ML/OR, especially defining various generalization errors concepts.
- The paper introduces a "distributional robustness" measure, which helps unify Bayesian stochastic optimization, distributional robust optimization (DRO), and regularization.

Weaknesses:
- The main message is lost and/or the paper overpromises: The paper claims that it introduces a distributional robustness measure to justify why Bayesian stochastic optimization and regularization generalize as well as DRO, given their connection. However, this message is lost, and I disagree that it's well-presented (Thm 6). In fact, the one-sided version of Thm 1, is a direct result of the equivalence of DRO and Definition 3 (DRO with one-sided absolute robustness) in addition to Lipschitz continuity. So why is this interesting? Section 2.4.3 is devoted to the benefits of bounding generalization errors by distributional robustness/uncertainty measure, but I'm not convinced we have learned anything new about DRO and regularization.

- I'm wondering if the main contribution of this paper is to investigate generalization error bounds for Bayesian SO.  If yes, I believe the contribution of this paper is only marginal in terms of novelty, and I continue to see this paper as a survey paper/tutorial, at best, which is fine for some audiences.

- Wrong proof in "Appendix D.2.3. Generalization error through relative robustness measure": Thm 8 claims a wrong statement in the second part of the theorem. Indeed, in the last part of the proof, the triangle inequality is used in the wrong direction! This theorem aims to reflect the insight from Thm 6 but uses the relative robustness measure as opposed to the absolute robustness measure.

---

### Note · Authors · 2025-04-15

I have read and agree with the venue's withdrawal policy on behalf of myself and my co-authors.